# Performative Policy Gradient:
# Optimality in Performative Reinforcement Learning

**Debabrota Basu** [* 1]   **Udvas Das** [* 1]   **Brahim Driss** [1]   **Uddalak Mukherjee** [* 2]

## Abstract

Post-deployment machine learning algorithms often influence the environments that they act in, and thus, *performatively shift* the underlying dynamics that the standard Reinforcement Learning (RL) ignores. While designing optimal algorithms in this *performative* setting has been studied in supervised learning, the RL counterpart remains under-explored. In this paper, we prove the performative counterparts of the performance difference lemma and the policy gradient theorem in RL, and introduce the **Performative Policy Gradient** algorithm (PePG). PePG is the first policy gradient algorithm designed to account for performativity in RL. Under softmax parametrisation, and also with and without entropy regularisation, we prove that PePG converges to *performatively optimal policies*, i.e. policies that remain optimal under the distribution shifts induced by themselves. Thus, PePG significantly extends the prior works in Performative RL that achieves *performative stability* but not optimality. Our empirical analysis on standard performative RL environments validate that PePG outperforms the existing performative RL algorithms aiming for stability[1].

## 1. Introduction

Reinforcement Learning (RL) studies the dynamic decision making problems under incomplete information (Sutton & Barto, 1998). Since an RL algorithm tries and optimises a utility function over a sequence of interactions with an unknown environment, RL has emerged as a powerful tool for

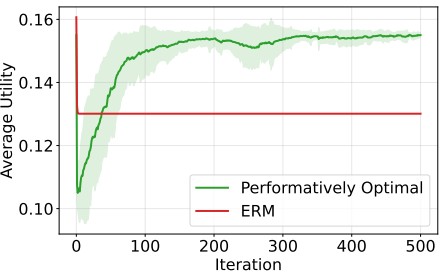

*Figure 1.* Average utility (over 20 runs) obtained by ERM and Performatively Optimal policies for $\beta = 0.5$.

algorithmic decision making. Specially, in the last decade, RL has underpinned some of the celebrated successes of AI, such as championing Go with AlphaGo (Silver et al., 2014), aligning Large Language Models (LLMs) (Bai et al., 2022), reasoning (Havrilla et al., 2024) etc. The classical paradigm of RL assumes the underlying environment to be *static*, and the goal of RL algorithms is to find a utility-maximising, aka *optimal policy*, for choosing actions over time. But *the static environment assumption does not hold universally.*

*In this digital age, algorithms are not passive.* Their decisions also shape the environment that they interact with, inducing distribution shifts. This phenomenon in which predictive AI models trigger actions that influence their own outcomes is known as *performativity*. In supervised learning, the study of *performative prediction* is pioneered by Perdomo et al. (2020), and followed by an extensive literature across optimisation, control, multi-agent RL, and games (Izzo et al., 2021; 2022; Miller et al., 2021; Li & Wai, 2022; Narang et al., 2023; Piliouras & Yu, 2023; Góis et al., 2024). There has been attempts to achieve performative optimality or stability for real-life tasks– recommender systems (Eilat & Rosenfeld, 2023), measuring the power of firms (Hardt et al., 2022; Mofakhami et al., 2023), healthcare (Zhang et al., 2022) etc. *Performativity is also omnipresent in deployed RL systems.* For example, an RL algorithm deployed in a recommender system does not only maximise the user satisfaction but also shifts the preferences of the users in long-term (Chaney et al., 2018; Mansoury et al., 2020).

**Example 1** (Performative RL in loan approval)**.** *A loan approval algorithm predicts whether an applicant should (or should not) obtain a loan according to their credit*

---

[*]Equal contribution [1]Univ. Lille, Inria, CNRS, Centrale Lille, UMR 9189 – CRIStAL F-59000 Lille, France [2]ACMU, Indian Statistical Institute, Kolkata - 700108, India. Correspondence to: Udvas Das <udvas.das@inria.fr>.

*Proceedings of the $43^{rd}$ International Conference on Machine Learning*, Seoul, South Korea. PMLR 306, 2026. Copyright 2026 by the author(s).

[1]Code is available at https://github.com/brahimdriss/PePG.

*score $x$ that depends on the capitals of the applicant and the population. At each time $t$, a loan applicant arrives with a credit score $x_t \sim \mathcal{N}(\mu_t, \sigma^2)$. The bank chooses to give a loan by applying a softmax binary classifier $\boldsymbol{\pi_\theta} : \mathbb{R} \to \{0, 1\}$ on $x_t$ with threshold parameter $\theta$. This decision has two effects. (a) The bank receives a positive payoff $R$, if the loan applicant repays their loan, or else, loses by $L$. Thus, the bank's expected utility for policy $\boldsymbol{\pi_\theta}$ is $U(\theta, \mu) \triangleq \mathbb{E}_x\big[\boldsymbol{\pi_\theta}(x)(\mathbb{P}(repay|x)R - (1 - \mathbb{P}(repay|x))L)\big]$. (b) Since the capitals of both the applicant and the population influence the credit score, we model the change in population mean $\mu_{t+1}$ by the bank's policy, i.e. approval rate $\mathbb{E}_x\big[\pi_\theta(x)\big]$. Specifically, $\mu_{t+1} \triangleq (1-\beta)\mu_t + \beta f\big(\mathbb{E}_{x_t}\big[\pi_\theta(x_t)\big]\big)$, where $\beta \in [0, 1]$ is the performative strength and $f : \mathbb{R} \to [-M, M]$. Now, if one ignores the performative nature of this problem, and tries to find the optimal with respect to a static credit distribution, it obtains $\theta^{\mathrm{ERM}} \triangleq \arg\max_\theta U(\theta, \mu_0)$. In contrast, if it considers performativity, it obtains $\theta^{\mathrm{Perf}} \triangleq \arg\max_\theta U(\theta, \mu^*(\theta))$. Figure 1 shows that the average utility obtained by $\theta^{\mathrm{ERM}}$ and $\theta^{\mathrm{Perf}}$ are different– demonstrating performativity as a common phenomenon across decision making problems, and its effect on the desired optimal solution (Appendix B).*

These practical problems have motivated the study of *performative RL*. Though Bell et al. (2021) were the first to propose a setting where the transition and reward of an underlying MDP non-deterministically depend on the deployed policy, (Mandal et al., 2023) formally introduced *Performative RL*, and its solution concepts, i.e., performatively stable and optimal policies. *Performatively stable policies* do not change due to distribution shifts after deployment. *Performatively optimal policies* yield the highest expected return once deployed in a performative RL environment. Mandal et al. (2023) proposed direct optimization and ascent based techniques to attain performative stability upon repeated retraining. Extending this, (Rank et al., 2024; Mandal & Radanovic, 2024) solved the same problem with delayed retraining for gradually shifting and linear MDPs. However, *there exists no algorithm yet in performative RL that provably converges to the performative optimal policy*.

In classical RL, Policy Gradient (PG) algorithms treat policy as a parametric function and update the parameters through gradient ascent algorithms (Williams, 1992; Sutton et al., 1999; Kakade, 2001). PGs are efficient and scalable. TRPO (Schulman et al., 2015), PPO (Schulman et al., 2017), NPG (Kakade, 2001), DDPG (Silver et al., 2014) are some of the PG algorithms widely used across modern RL. Recent theoretical advances also establish finite-sample convergence guarantees (Agarwal et al., 2021; Yuan et al., 2022) for different PG algorithms. Motivated by the simplicity and efficiency of PG algorithms, we ask two questions.

---

1. *How to design PG-type algorithms for performative RL environments to achieve optimality?*
2. *What are the minimal conditions for PG-type algorithms to converge to the performatively optimal policy?*

---

**Our contributions** address both the questions affirmatively, and showcase the difference of optimality-seeking and stability-seeking algorithms in performative RL.

**I. Algorithm Design:** *We propose the first Performative Policy Gradient algorithm,* PePG, *both with and without entropy-regularisation for performative RL environments (Section 3.2).* We derive the performative policy gradient theorem showing that the gradient of performative value function involves the classical policy gradient term and two novel gradient terms for environment shifts– (a) the expected gradient of reward, and (b) the expected gradient of log-transition probabilities times its impact on the expected cumulative return. We deploy this result to estimate the performative policy gradient for *any differentiable parametrisations of the policy and the environment.*

**II. Convergence to Performative Optimality.** We show that for any Performative Markov Decision Process (PeMDP) with smooth transition functions and rewards, PePG converges close to the optimal policy (Section 3.1). We provide a novel and generic recipe to prove convergence of PePG via (a) smoothness of the performative value function, and (b) a performative gradient domination lemma capturing the per-step improvements due to performative policy gradients. As a concrete example, we show that, for PeMDPs with softmax policies, linear rewards and exponential family transitions, PePG converges to an $\left(\epsilon + \frac{1}{1-\gamma}\right)$-ball around a performative optimal policy in $\mathcal{O}\left(\frac{|\mathcal{S}||\mathcal{A}|^2}{\epsilon^2(1-\gamma)^3}\right)$ iterations, where $|\mathcal{S}|$ and $|\mathcal{A}|$ are the number of states and actions, respectively, and $\gamma$ is discount factor.

**III. Stability- vs. Optimality-seeking Performative RL.** We further theoretically and numerically contrast the performances of stability-seeking and optimality-seeking algorithms. Theoretically, we derive the performative performance difference lemma that distinguishes the effect of policy update on these two objectives. Experimentally, we compare the performances of PePG with the state-of-the-art stability-seeking algorithms, MDRR (Mixed Delayed Repeated Retraining, Rank et al. (2024)) and RPO-FS (Mandal et al., 2023). The results validate that PePG yields significantly higher average return than the baselines.

---

(Mandal et al., 2023) poses the question of developing performative policy gradient algorithms as an open problem. *We* affirmatively *solve an extension of this open problem to compute the performatively optimal policies for PeMDPs with discrete state-actions.*

---

## 2. Preliminaries: From RL to Performative RL

Now, we formalise the RL and performative RL problems, and provide the basics of policy gradient algorithms in RL.

### 2.1. Infinite-horizon Discounted MDP

In RL, we commonly study Markov Decision Processes (MDPs) defined as the tuple $(\mathcal{S}, \mathcal{A}, \mathbf{P}, r, \gamma)$. Here, $\mathcal{S} \subseteq \mathbb{R}^d$ is the state space and $\mathcal{A} \subseteq \mathbb{R}^d$ is the action space. Both the spaces are assumed to be compact. At any time $t \in \mathbb{N}$, an agent plays an action $a_t \in \mathcal{A}$ at a state $s_t \in \mathcal{S}$. It transits the MDP environment to a state $s_{t+1}$ according to a transition function $\mathbf{P}(\cdot \mid s_t, a_t) \in \Delta(\mathcal{S})$. The agent further receives a reward $r(s_t, a_t) \in \mathbb{R}$ quantifying the goodness of taking the action $a_t$ at $s_t$. The strategy to take an action is represented by a stochastic map, called *policy*, i.e., $\boldsymbol{\pi}$ : $\mathcal{S} \to \Delta(\mathcal{A})$. Given an initial state distribution $\boldsymbol{\rho} \in \Delta(\mathcal{S})$, *the goal is to find the optimal policy $\boldsymbol{\pi}^\star$ that maximises* the expected discounted sum of rewards, i.e., the *value function*: $V^{\boldsymbol{\pi}}(\boldsymbol{\rho}) \triangleq \mathbb{E}_{s_0 \sim \boldsymbol{\rho}, s_{t+1} \sim \mathbf{P}(\cdot \mid s_t, \boldsymbol{\pi}(s_t))} [\sum_{t=0}^{\infty} \gamma^t r(s_t, \boldsymbol{\pi}(s_t))]$, where $\gamma \in (0, 1)$ is called the *discount factor*. $\gamma$ indicates how much a previous reward matters in the next step, and bounds the effective horizon of a policy to $(1 - \gamma)^{-1}$.

**Policy Gradient (PG) Algorithms.** PG algorithms maximise the value function by updating the policy with a gradient of value function (Williams, 1992). To compute the gradient, we choose a parametric family of policies $\boldsymbol{\pi_\theta}$ for some $\boldsymbol{\theta} \in \mathbb{R}^d$ (e.g. direct (Agarwal et al., 2021; Wang & Zou, 2022), softmax (Agarwal et al., 2021; Mei et al., 2020), Gaussian (Ciosek & Whiteson, 2020)). Specifically, vanilla PG (Algorithm 1) performs a gradient ascent on the policy parameter at each step $t$. As the goal is to maximise $V^{\boldsymbol{\pi}}(\boldsymbol{\rho})$, we update $\boldsymbol{\theta}$ towards $\nabla_{\boldsymbol{\theta}} V^{\boldsymbol{\pi}}(\boldsymbol{\rho})$, i.e., the direction improving the value $V^{\boldsymbol{\pi}}(\boldsymbol{\rho})$ with a fixed learning rate $\eta > 0$. For vanilla PG, the policy gradient takes the convenient form leading to estimators computable only with policy rollouts.

---

**Algorithm 1** Vanilla Policy Gradient

---
1: **Input:** Learning rate $\eta > 0$.
2: **Initialize:** Policy parameter $\boldsymbol{\theta}_0(s, a) \forall s \in \mathcal{S}, a \in \mathcal{A}$.
3: **for** $t = 1$ to T **do**
4:     Estimate the gradient $\nabla_{\boldsymbol{\theta}} V^{\boldsymbol{\pi}}(\boldsymbol{\rho}) \mid_{\boldsymbol{\theta} = \boldsymbol{\theta}_t}$
5:     **Gradient ascent:** $\boldsymbol{\theta}_{t+1} \leftarrow \boldsymbol{\theta}_t + \eta \nabla_{\boldsymbol{\theta}} V^{\boldsymbol{\pi}}(\boldsymbol{\rho}) \mid_{\boldsymbol{\theta} = \boldsymbol{\theta}_t}$
6: **end for**

---

**Theorem 1** (Policy Gradient Theorem (Sutton et al., 1999)). *Given a differentiable parametrisation $\boldsymbol{\theta} \mapsto \boldsymbol{\pi_\theta}$, the Q-value and advantage functions are $Q^{\boldsymbol{\pi_\theta}}(s, a) \triangleq \mathbb{E}_{s_{t+1} \sim \mathbf{P}(\cdot \mid s_t, \boldsymbol{\pi}(s_t))} [\sum_{t=0}^{\infty} \gamma^t r(s_t, \boldsymbol{\pi}(s_t)) \mid s_0 = s, a_0 = a]$ and $A^{\boldsymbol{\pi_\theta}}(s, a) \triangleq Q^{\boldsymbol{\pi_\theta}}(s, a) - V^{\boldsymbol{\pi_\theta}}(s)$. Then, the gradient of value function is $(1 - \gamma) \nabla_{\boldsymbol{\theta}} V^{\boldsymbol{\pi_\theta}}(\boldsymbol{\rho}) = \mathbb{E}_{\tau \sim \mathbb{P}^{\boldsymbol{\pi_\theta}}} [\sum_{t=0}^{\infty} \gamma^t A^{\boldsymbol{\pi_\theta}}(s, a) \nabla_{\boldsymbol{\theta}} \log \boldsymbol{\pi_\theta}(a \mid s)]$.*

Since the value function is not concave in the policy param-

eters, achieving optimality with PG becomes a challenge. But scalability and efficiency of PG motivated a rich line of research deriving minimum conditions and parametric forms of policies for convergence to optimal policy (Agarwal et al., 2021; Mei et al., 2020; Wang & Zou, 2022; Yuan et al., 2022). *Our work extends these algorithmic techniques and theoretical insights to performative RL.*

### 2.2. Infinite-horizon Discounted Performative MDP

Given a set of policies $\Pi$, the Performative Markov Decision Process (PeMDP) is defined as the set of MDPs $\mathcal{M}(\Pi) \triangleq \{\mathcal{M}(\boldsymbol{\pi}) \mid \boldsymbol{\pi} \in \Pi\}$, where each MDP is a tuple $\mathcal{M}(\boldsymbol{\pi}) \triangleq (\mathcal{S}, \mathcal{A}, \mathbf{P}_{\boldsymbol{\pi}}, r_{\boldsymbol{\pi}}, \gamma)$. Note that the transition function and rewards are now functions of the deployed policy $\boldsymbol{\pi} \in \Delta(\mathcal{A})$ (Mandal et al., 2023). In this setting, the probability of generating a trajectory $\tau \triangleq (s_t, a_t)_{t=0}^{\infty}$ with a policy $\boldsymbol{\pi}$ and underlying MDP $\mathcal{M}(\boldsymbol{\pi}')$ is given by[2] $\mathbb{P}_{\boldsymbol{\pi}'}^{\boldsymbol{\pi}}(\tau \mid \boldsymbol{\rho}) \triangleq \boldsymbol{\rho}(s_0) \prod_{t=0}^{\infty} \boldsymbol{\pi}(a_t \mid s_t) \mathbf{P}_{\boldsymbol{\pi}'}(s_{t+1} \mid s_t, a_t)$. The state-action occupancy measure for the deployed policy $\boldsymbol{\pi}$ and the environment-inducing policy $\boldsymbol{\pi}'$ is $d_{\boldsymbol{\pi}', \boldsymbol{\rho}}^{\boldsymbol{\pi}} \triangleq (1 - \gamma) \quad \mathbb{E}_{\tau \sim \mathbb{P}_{\boldsymbol{\pi}'}^{\boldsymbol{\pi}}} [\sum_{t=0}^{\infty} \gamma^t \mathbb{1}(s_t = s, a_t = a) \mid s_0 \sim \boldsymbol{\rho}]$. Now, we define the total expected return in PeMDPs, i.e., the *performative value function*, that we aim to maximise.

**Definition 1** (Performative Value Function). *Given a policy $\boldsymbol{\pi} \in \Pi$ and an initial state distribution $\boldsymbol{\rho} \in \Delta(\mathcal{S})$, the performative value function $V_{\boldsymbol{\pi}}^{\boldsymbol{\pi}}(\boldsymbol{\rho})$ is*

$$V_{\boldsymbol{\pi}}^{\boldsymbol{\pi}}(\boldsymbol{\rho}) \triangleq \mathbb{E}_{\tau \sim \mathbb{P}_{\boldsymbol{\pi}}^{\boldsymbol{\pi}}} \left[ \sum_{t=0}^{\infty} \gamma^t r_{\boldsymbol{\pi}}(s_t, \boldsymbol{\pi}(s_t)) \mid s_0 \sim \boldsymbol{\rho} \right]. \quad (1)$$

Equation (2) captures the performativity aspect in PeMDPs as the dynamics changes with a deployed policy $\boldsymbol{\pi}(\cdot \mid s)$. On a similar note, we define the performative Q-value function (or action-value function) of a policy $\boldsymbol{\pi}$.

**Definition 2** (Performative Q-value). *Given a policy $\boldsymbol{\pi} \in \Pi$ and an initial state-action pair $(s, a) \in (\mathcal{S}, \mathcal{A})$, the performative Q-value function $Q_{\boldsymbol{\pi}}^{\boldsymbol{\pi}}(s, a)$ is*

$$Q_{\boldsymbol{\pi}}^{\boldsymbol{\pi}}(s, a) \triangleq \mathbb{E}_{\tau \sim \mathbb{P}_{\boldsymbol{\pi}}^{\boldsymbol{\pi}}} \left[ \sum_{t=0}^{\infty} \gamma^t r_{\boldsymbol{\pi}}(s_t, a_t) \Big| s_0 = s, a_0 = a \right] \quad (2)$$

Performative Q-value satisfies $Q_{\boldsymbol{\pi}}^{\boldsymbol{\pi}}(s, a) = r_{\boldsymbol{\pi}}(s, a) + \gamma \mathbb{E}_{s' \sim \mathbf{P}_{\boldsymbol{\pi}}(\cdot \mid s, a)} [V_{\boldsymbol{\pi}}^{\boldsymbol{\pi}}(s')]$. We can maximise performative value function in two ways: (i) considering $\boldsymbol{\pi}$ as both the environment-inducing policy and the policy of the RL agent, or (ii) agent plays another policy $\boldsymbol{\pi}'$, while fixing $\boldsymbol{\pi}$ as the environment-inducing policy. At this point, we introduce the notions of optimal and stable policies in PeMDPs.

**Definition 3** (Performative Optimality). *A policy $\boldsymbol{\pi}_o^\star$ is performatively optimal if it maximizes the performative value function, i.e., $\boldsymbol{\pi}_o^\star \in \arg\max_{\boldsymbol{\pi} \in \Delta(\mathcal{A})} V_{\boldsymbol{\pi}}^{\boldsymbol{\pi}}(\boldsymbol{\rho})$.*

---
[2]Hereafter, for relevant quantities, $\boldsymbol{\pi}$ in superscript denotes the deployed policy, and $\boldsymbol{\pi}'$ in subscript denotes the transition and reward functions shifting policy that the algorithm interacts with.

This implies that if we play the policy $\boldsymbol{\pi}$ in the environment induced by policy $\boldsymbol{\pi}$ to maximise the expected return, we land on the performatively optimal policy.

**Definition 4** (Performative Stability). *A policy $\boldsymbol{\pi}_s^\star$ is performatively stable if there is no gain in performative value function due to deploying any other policy than $\boldsymbol{\pi}_s^\star$ in the environment induced by $\boldsymbol{\pi}_s^\star$ i.e., $\boldsymbol{\pi}_s^\star \in \arg\max_{\boldsymbol{\pi}\in\Delta(\mathcal{A})} V_{\boldsymbol{\pi}_s^\star}^{\boldsymbol{\pi}}(\boldsymbol{\rho})$.*

A performatively optimal policy may not be stable, i.e., $\boldsymbol{\pi}_o^\star$ may not be optimal for an environment $\mathcal{M}(\boldsymbol{\pi}_o^\star)$, when it is deployed (Mandal et al., 2023). In general, the performative value function of $\boldsymbol{\pi}_o^\star$ might be equal to or higher than that of $\boldsymbol{\pi}_s^\star$. In this work, *we design PG algorithms computing a performatively optimal policy for a given PeMDP*, and reinstate their differences with performatively stable policies.

The existing literature on PeMDPs (Mandal et al., 2023; Mandal & Radanovic, 2024; Rank et al., 2024; Chen et al., 2024; Pollatos et al., 2025) focused primarily on finding a performatively stable policy (Definition 4). In practice, while the stable policies matter for certain applications, they might show very sub-optimal performance, which are not desired in many real-life tasks. *We bridge this gap by proposing the first provably converging and computationally-efficient PG algorithm for PeMDPs.* We also empirically show the deficiency of the existing stability-seeking algorithms if we aim for optimality (Section 5).

**Entropy Regularised PeMDPs.** Entropy regularisation has emerged as a simple but powerful technique in classical RL to design smooth and efficient RL algorithms with sufficient exploration. Thus, we study a variant of the performative value function that is regularised using discounted entropy (Neu et al., 2017; Liu et al., 2019; Zhao et al., 2019). In this approach, the original value function in Definition 1 is regularised with the discounted entropy $H_{\boldsymbol{\pi}}(\boldsymbol{\rho}) \triangleq \mathbb{E}_{\tau\sim\mathbb{P}_{\boldsymbol{\pi}}^{\boldsymbol{\pi}}}[-\sum_{t=0}^\infty \gamma^t \log\boldsymbol{\pi}(a_t \mid s_t)]$. This is equivalent to maximising the expected reward with a shifted reward function $\tilde{r}_{\boldsymbol{\pi}}(\boldsymbol{\pi}(s_t),s_t) \triangleq r_{\boldsymbol{\pi}}(\boldsymbol{\pi}(s_t),s_t) - \lambda\log(\boldsymbol{\pi}(a_t \mid s_t))$ for some $\lambda \geq 0$. $\tilde{r}_{\boldsymbol{\pi}}$ is referred as the "soft-reward" in literature (Wang & Uchibe, 2024; Herman et al., 2016; Shi et al., 2019). Now, we define the *soft performative value function*.

**Definition 5** (Entropy Regularised (or *Soft*) Performative Value Function). *Given a policy $\boldsymbol{\pi} \in \Pi$, a starting state distribution $\boldsymbol{\rho} \in \Delta(S)$, and a regularisation parameter $\lambda \geq 0$, the soft performative value function*

$$\tilde{V}_{\boldsymbol{\pi}}^{\boldsymbol{\pi}}(\boldsymbol{\rho}) \triangleq \mathbb{E}_{\substack{\tau\sim\mathbb{P}_{\boldsymbol{\pi}}^{\boldsymbol{\pi}} \\ s_0\sim\boldsymbol{\rho}}}\left[\sum_{t=0}^\infty \gamma^t \tilde{r}_{\boldsymbol{\pi}}(s_t,\boldsymbol{\pi}(s_t))\right] \quad (3)$$

Since policies belong to the probability simplex, the entropy regularisation naturally lends to smoother and stable PG algorithms. Later, we show that the discounted entropy is a smooth function of the policy parameters for PeMDPs

extending the optimization-wise benefits of entropy regularisation to PeMDPs. Additionally, using the notion of soft rewards, we can similarly define soft performatively optimal and stable policies for entropy regularised PeMDPs. Here, *we unifiedly design PG algorithms for both the unregularised and the entropy regularised PeMDPs*.

# 3. Policy Gradient in Performative RL

We first study the impact of policy updates in PeMDPs. Then, we leverage it to derive the performative policy gradient theorem, and design Performative PG (PePG) algorithm for any differentiable parametric policy class.

## 3.1. Impact of Policy Updates on PeMDPs

In RL, performance difference lemma quantifies the impact of changing policies on the value functions (Kakade & Langford, 2002a). It has been central to analysing and developing PG algorithms (Agarwal et al., 2021; Silver et al., 2014; Kallel et al., 2024). Here, we derive the performative version of the performance difference lemma quantifying the shift in the performative value function due to change in the deployed and environment-inducing policies.

---

**Lemma 1** (Performative Performance Difference Lemma). *The difference in performative value functions induced by $\boldsymbol{\pi}$ and $\boldsymbol{\pi}' \in \Pi$ while starting from the initial state distribution $\boldsymbol{\rho}$ is $V_{\boldsymbol{\pi}}^{\boldsymbol{\pi}}(\boldsymbol{\rho}) - V_{\boldsymbol{\pi}'}^{\boldsymbol{\pi}'}(\boldsymbol{\rho}) =$*

$$\frac{1}{1-\gamma}\Big(\mathbb{E}_{(s,a)\sim d_{\boldsymbol{\pi}',\boldsymbol{\rho}}^{\boldsymbol{\pi}}}\big[A_{\boldsymbol{\pi}'}^{\boldsymbol{\pi}'}(s,a) + (r_{\boldsymbol{\pi}}(s,a) - r_{\boldsymbol{\pi}'}(s,a))$$
$$+\gamma(\mathbf{P}_{\boldsymbol{\pi}}(\cdot|s,a) - \mathbf{P}_{\boldsymbol{\pi}'}(\cdot|s,a))^\top V_{\boldsymbol{\pi}}^{\boldsymbol{\pi}}(\cdot)\big]\Big) \quad (4)$$

*where $A_{\boldsymbol{\pi}'}^{\boldsymbol{\pi}'}(s,a) \triangleq Q_{\boldsymbol{\pi}'}^{\boldsymbol{\pi}'}(s,a) - V_{\boldsymbol{\pi}'}^{\boldsymbol{\pi}'}(s)$ is the performative advantage function for any $s \in \mathcal{S}$ and $a \in \mathcal{A}$.*

---

The crux is to decompose the performative value among environment-inducing and deployed policies. Specifically, we observe that the suboptimality gap

$$\text{SubOpt}(\boldsymbol{\pi}') \triangleq V_{\boldsymbol{\pi}^\star}^{\boldsymbol{\pi}^\star}(\boldsymbol{\rho}) - V_{\boldsymbol{\pi}'}^{\boldsymbol{\pi}'}(\boldsymbol{\rho})$$
$$= \underbrace{V_{\boldsymbol{\pi}^\star}^{\boldsymbol{\pi}^\star}(\boldsymbol{\rho}) - V_{\boldsymbol{\pi}'}^{\boldsymbol{\pi}^\star}(\boldsymbol{\rho})}_{\text{performative shift term}} + \underbrace{V_{\boldsymbol{\pi}'}^{\boldsymbol{\pi}^\star}(\boldsymbol{\rho}) - V_{\boldsymbol{\pi}'}^{\boldsymbol{\pi}'}(\boldsymbol{\rho})}_{\text{performance difference term}}$$

(1) *Connection to Classical RL.* In classical RL, the performance difference lemma yields $V^{\boldsymbol{\pi}}(\boldsymbol{\rho}) - V^{\boldsymbol{\pi}'}(\boldsymbol{\rho}) = \frac{1}{1-\gamma}\mathbb{E}_{(s,a)\sim d_\rho^{\boldsymbol{\pi}}}[A^{\boldsymbol{\pi}'}(s,a)]$. The first term in Lemma 1 is equivalent to the classical result in the environment induced by $\boldsymbol{\pi}'$. But due to environment shift, two more terms appear in the performative performance difference incorporating the impacts of reward shifts and transition shifts.

(2) *Connection to Performative Stability.* The performance difference term, i.e., $V_{\boldsymbol{\pi}'}^{\boldsymbol{\pi}}(\boldsymbol{\rho}) - V_{\boldsymbol{\pi}'}^{\boldsymbol{\pi}'}(\boldsymbol{\rho})$, quantifies the impact

**Algorithm 2** PePG: **Pe**rformative **P**olicy **G**radient

1: **Input:** Reward bound $R_{\max} > 0$, discount factor $\gamma \in (0, 1)$, initial state distribution $\rho$, learning rate $\eta > 0$
2: **Initialize:** Initial policy parameters $\theta_0$, initial value function parameters $\phi_0$.
3: **for** $k = 1, 2, \ldots$ **do**
4:     **Collect trajectories:** $\mathcal{D}_k = \{\tau_i\}_{i=1}^I$, where each $\tau_i \triangleq \{(s_{i,t}, a_{i,t}, s_{i,t+1}, r_{i,t})\}_{t=0}^{T-1}$ of length $T$ by playing $\pi_{\theta_k}$
5:     Compute returns $R_k \triangleq \{R_{k,i}\}_{i=1}^I$, where $R_{k,i} = \{R_{k,i,t}\}_{t=0}^{T-1}$
6:     Compute advantage estimates $\hat{A}_k(\tau_i)$ using value function $\hat{V}_{\phi_k}(\tau_i)$ for each $\tau_i \in \mathcal{D}_k$, i.e., estimate of $V_{\pi_{\theta_k}}^{\pi_{\theta_k}}(\tau_i)$ obtained from fitted value network with parameters $\phi_k$
7:     **Gradient estimation:** Estimate gradient using (8)
8:     **Ascent step:** Use parameter update Equation (5)
9:     Fit value function $V_{\phi_{k+1}}$

$$\phi_{k+1} \leftarrow \arg\min_{\phi} \frac{1}{IT} \sum_{i=1}^I \sum_{t=0}^{T-1} \left( \hat{V}_{\phi_k}(s_t \in \tau_i) - R_{k,i,t} \right)^2$$

10: **end for**

of changing the deployed policy from $\pi'$ to $\pi$ in an environment induced by $\pi'$. Thus, a stability-seeking algorithm tries to minimise this term, while an optimality-seeking algorithm incorporates both the terms.

Hence, we ask:

> *How much does the performative shift term influence the performative performance difference?*

Hereafter, we focus on PeMDPs with bounded rewards, which is a common assumption in RL (Agarwal et al., 2019).

**Assumption 1** (Bounded Rewards). *We assume that the rewards of PeMDPs are bounded in* $[-R_{\max}, R_{\max}]$.

Now, we bound the effect of performative shift for gradually shifting PeMDPs with Lipschitz transitions and rewards with respect to the deployed policies (Rank et al., 2024).

**Lemma 2** (Bounding the Performative Performance Difference for Gradually Shifting Environments). *Let us assume that the rewards and transitions are Lipschitz functions of policy, i.e.,* $\|r_{\pi} - r_{\pi'}\|_1 \leq L_r \|\pi - \pi'\|_{\infty}$ *and* $\|\mathbf{P}_{\pi} - \mathbf{P}_{\pi'}\|_1 \leq L_{\mathbf{P}} \|\pi - \pi'\|_{\infty}$, *for* $L_r, L_{\mathbf{P}} \geq 0$. *Under Assumption 1, the performative shift in the sub-optimality gap of a policy* $\pi_{\theta}$ *satisfies*

$$\left| \mathrm{SubOpt}(\pi_{\theta}) - \frac{1}{1-\gamma} \mathbb{E}_{(s,a) \sim d_{\pi_{\theta}, \rho}^{\pi_o^\star}} [A_{\pi_{\theta}}^{\pi_{\theta}}(s, a)] \right| \leq$$

$$\frac{2\sqrt{2}}{1-\gamma} \left( L_r + \frac{\gamma L_{\mathbf{P}} R_{\max}}{1-\gamma} \right) \mathbb{E}_{s_0 \sim \rho} D_{\mathrm{H}} \left( \pi_o^\star(\cdot|s_0) \| \pi_{\theta}(\cdot|s_0) \right)$$

$D_{\mathrm{H}}(\cdot\|\cdot)$ *is the Hellinger distance between distributions.*

**Implications.** In classical RL, the RHS of Lemma 2 is 0 (Agarwal et al., 2019, Lemma 2), i.e. advantage solely characterises the suboptimality gap in the usual non-performative MDPs. Lemma 2 characterises the *extra cost* to adapt to performativity of the environment in terms of Hellinger distance between the true performatively optimal policy $\pi_o^\star$ and any other policy $\pi_{\theta}$. This implies that the gap between the optimal performative value function and that of any stability-seeking algorithm is $\mathcal{O}\left((1-\gamma)^{-1}\right)$, if we set $R_{\max} = \mathcal{O}(1-\gamma)$. This gap is significantly less than the sub-optimality gap achieved by the existing algorithms. For example, repeated policy optimisation shows sub-optimality gap $\mathcal{O}\left(\max\{\frac{|\mathcal{S}|^{5/3}|\mathcal{A}|^{1/3}\epsilon^{2/3}}{(1-\gamma)^{14/3}}, \frac{\epsilon|\mathcal{S}|}{(1-\gamma)^4}\}\right)$ (Mandal et al., 2023). Thus, we aim to design algorithms achieving sub-optimality gap $\mathcal{O}\left((1-\gamma)^{-1}\right)$ (Theorem 4).

We note that an optimality-seeking algorithm tries to minimise both the advantage function and the performative shifts in the environment quantified by $D_{\mathrm{H}}\left(\pi_o^\star(\cdot|s_0)\|\pi_{\theta}(\cdot|s_0)\right)$. A stability-seeking algorithm tries to minimise the advantage function, and thus, RHS of Lemma 2 cannot go lower than $\mathcal{O}((1-\gamma)^{-2})$. Thus, the optimality-seeking algorithms can yield a lower sub-optimality gap than the stability-seeking algorithms if they also incorporate the performative shifts.

**3.2. Algorithm: Performative Policy Gradient (PePG)**

For optimality-seeking algorithms, the goal is to maximise the performative value function. Gradient ascent is a standard first-order optimisation method to find maxima of a function. The ascent step of performative policy gradient is

$$\theta_{t+1} \leftarrow \begin{cases} \theta_t + \eta_t \nabla_{\theta} V_{\pi_{\theta}}^{\pi_{\theta}}(\tau) \mid_{\theta=\theta_t} & \text{(unregularised)} \\ \theta_t + \eta_t \nabla_{\theta} \tilde{V}_{\pi_{\theta}}^{\pi_{\theta}}(\tau) \mid_{\theta=\theta_t} & \text{(entropy reg.)} \end{cases} \quad (5)$$

Given this ascent step, we evaluate the gradient at each time from the roll-outs of the present policy. In classical PG, the policy gradient theorem supports this computation (Williams, 1992; Sutton et al., 1999; Silver et al., 2014). Now, we derive the *performative policy gradient theorem*.

**Theorem 2** (Performative Policy Gradient Theorem). *(a) For the unregularised case, the gradient of the performative value function with respect to* $\theta$ *is* $\nabla_{\theta} V_{\pi_{\theta}}^{\pi_{\theta}}(\tau) =$

$$\mathbb{E}_{\tau \sim \mathbb{P}_{\pi_{\theta}}^{\pi_{\theta}}} \left[ \sum_{t=0}^{\infty} \gamma^t A_{\pi_{\theta}}^{\pi_{\theta}}(s_t, a_t) \left( \nabla_{\theta} \log \pi_{\theta}(a_t \mid s_t) \right) \right.$$

$$+ \nabla_{\boldsymbol{\theta}} \log P_{\boldsymbol{\pi_\theta}}(s_{t+1}|s_t, a_t) \Big) + \gamma^t \nabla_{\boldsymbol{\theta}} r_{\boldsymbol{\pi_\theta}}(s_t, a_t) \Big]. \quad (6)$$

*(b) For the entropy-regularised case, we define the soft advantage, soft Q-value, and soft value functions with respect to the soft rewards $\tilde{r}_{\boldsymbol{\pi_\theta}}$ satisfying $\tilde{A}_{\boldsymbol{\pi_\theta}}^{\boldsymbol{\pi_\theta}}(s,a) = \tilde{Q}_{\boldsymbol{\pi_\theta}}^{\boldsymbol{\pi_\theta}}(s,a) - \tilde{V}_{\boldsymbol{\pi_\theta}}^{\boldsymbol{\pi_\theta}}(s)$ that further yields $\nabla_{\boldsymbol{\theta}} \tilde{V}_{\boldsymbol{\pi_\theta}}^{\boldsymbol{\pi_\theta}}(\tau) =$*

$$\mathop{\mathbb{E}}_{\tau \sim \mathbb{P}_{\boldsymbol{\pi_\theta}}^{\boldsymbol{\pi_\theta}}} \Big[ \sum_{t=0}^{\infty} \gamma^t \tilde{A}_{\boldsymbol{\pi_\theta}}^{\boldsymbol{\pi_\theta}}(s_t, a_t) \Big( \nabla_{\boldsymbol{\theta}} \log \boldsymbol{\pi_\theta}(a_t \mid s_t)$$
$$+ \nabla_{\boldsymbol{\theta}} \log P_{\boldsymbol{\pi_\theta}}(s_{t+1}|s_t, a_t) \Big) + \gamma^t \nabla_{\boldsymbol{\theta}} \tilde{r}_{\boldsymbol{\pi_\theta}}(s_t, a_t) \Big]. \quad (7)$$

PePG: With the appropriate parameter choices, and initialisation of the policy parameter $\boldsymbol{\theta}$ and value function parameter $\phi$, for each episode $k$, PePG first collects $I$ trajectories to calculate the return $R^i$ and estimates the advantage function $\hat{A}_k$ (Line 4-6). For a particular trajectory $\tau_i$, the estimated advantage for a given state-action is $\widehat{A_{\boldsymbol{\pi_{\theta_k}}}^{\boldsymbol{\pi_{\theta_k}}}}(s_t^i, a_t^i) = R_{t,k}^i - V_{\phi_k}(s_t^i)$, where $R^i = \sum_{t=0}^{T-1} \gamma^t r_{\boldsymbol{\theta_k}}(s_t^i, a_t^i)$.

**Gradient Estimation (Line 7).** Using all the $I$ trajectories, PePG computes a gradient estimate as $\widehat{\nabla_{\boldsymbol{\theta_k}} V_{\boldsymbol{\pi_{\theta_k}}}^{\boldsymbol{\pi_{\theta_k}}}}(\tau) =$

$$\frac{1}{I} \sum_{i=1}^{I} \sum_{t=0}^{T} \gamma^t \Big( \widehat{A_{\boldsymbol{\pi_{\theta_k}}}^{\boldsymbol{\pi_{\theta_k}}}}(s_t^i, a_t^i) \Big( \nabla_{\boldsymbol{\theta_k}} \log \boldsymbol{\pi_{\theta_k}}(a_t^i \mid s_t^i) +$$
$$\nabla_{\boldsymbol{\theta_k}} \log P_{\boldsymbol{\pi_{\theta_k}}}(s_{t+1}^i|s_t^i, a_t^i) \Big) + \nabla_{\boldsymbol{\theta_k}} r_{\boldsymbol{\pi_{\theta_k}}}(s_t^i, a_t^i|\boldsymbol{\theta_k}) \Big). \quad (8)$$

The performative gradient (Equation (8)) requires two terms: $\nabla_{\boldsymbol{\theta}} r_{\boldsymbol{\pi_\theta}}(s,a)$ and $\nabla_{\boldsymbol{\theta}} \log \mathbf{P}_{\boldsymbol{\pi_\theta}}(s'|s,a)$ in addition to gradient of log-policy. The computation of the first two gradients depend on what we know about the environment's response to $\boldsymbol{\theta}$. For a known parametric form like exponential family PeMDPs (Section 4), the transition has an explicit form $\mathbf{P}_{\boldsymbol{\pi_\theta}}(s'|s,a) = \exp(\boldsymbol{\theta}_{s,a} \psi(s') - \log Z(\boldsymbol{\theta}))$ and rewards are linear in $\boldsymbol{\theta}$. Both gradients can be computed in closed form, and the performative gradient in Equation (8) is exact. For unknown parametric general PeMDPs, when the closed form is unavailable, we learn differentiable approximations: a reward model $f_r : \boldsymbol{\theta} \mapsto \hat{r}_{\boldsymbol{\pi_\theta}}$ and a transition model $f_p : \boldsymbol{\theta} \mapsto \hat{\mathbf{P}}_{\boldsymbol{\pi_\theta}}$ via neural networks. The gradient terms are obtained by back-propagating through $f_r$ and $f_p$.

Finally, in Line 8, PePG updates the policy parameter for the next episode using a gradient ascent step leveraging the estimated average gradient over all $I$ trajectories. Specifically, we plug in $\widehat{\nabla_{\boldsymbol{\theta_k}} V_{\boldsymbol{\pi_{\theta_k}}}^{\boldsymbol{\pi_{\theta_k}}}}$ to both the unregularised and entropy-regularised update rules as given in Equation (5).

## 4. Convergence Analysis of PePG

In this section, we first derive the minimal condition on the PeMDPs and derive the convergnce analysis of PePG

for them. Then, we specialise the analysis for exponential family PeMDPs. Hereafter, we use softmax parametrisation of policies (Mei et al., 2020; Agarwal et al., 2019; 2021), defined as $\pi_{\boldsymbol{\theta}}(a|s) = \frac{e^{\boldsymbol{\theta}_{s,a}}}{\sum_{a'} e^{\boldsymbol{\theta}_{s,a'}}}$ for all $a \in \mathcal{A}$ and $s \in \mathcal{S}$.

**Generic Convergence Analysis.** To maximise a given objective, any first-order ascent method uses gradients to compute the direction for improvement. Following that, smoothness of the objective plays a critical role to stitch the ascent steps over iterations (Equation (5)), and thus, influences the convergence rate, step size selection, and overall efficiency.

Specifically, for PeMDPs the value function depends on reward and transition functions. Thus, we start the convergence analysis of PePG with the necessary smoothness assumption on reward and transitions, and their gradients.

**Assumption 2** (Smooth/Bounded Sensitivity PeMDPs). *The transition and reward functions of the PeMDP are (a) Lipschitz functions of policy, i.e. $\|r_{\boldsymbol{\pi}} - r_{\boldsymbol{\pi}'}\|_1 \leq L_r \|\boldsymbol{\pi} - \boldsymbol{\pi}'\|_\infty$ and $\|\mathbf{P}_{\boldsymbol{\pi}} - \mathbf{P}_{\boldsymbol{\pi}'}\|_1 \leq L_{\mathbf{P}} \|\boldsymbol{\pi} - \boldsymbol{\pi}'\|_\infty$, with $L_r, L_{\mathbf{P}} \geq 0$, and (b) smooth functions of policy, i.e. $\|\nabla_{\boldsymbol{\pi}} r_{\boldsymbol{\pi}} - \nabla_{\boldsymbol{\pi}'} r_{\boldsymbol{\pi}'}\|_1 \leq R_2 \|\boldsymbol{\pi} - \boldsymbol{\pi}'\|_\infty$ and $\|\nabla_{\boldsymbol{\pi}} \mathbf{P}_{\boldsymbol{\pi}} - \nabla_{\boldsymbol{\pi}'} \mathbf{P}_{\boldsymbol{\pi}'}\|_1 \leq T_2 \|\boldsymbol{\pi} - \boldsymbol{\pi}'\|_\infty$ with $R_2, T_2 \geq 0$.*

Assumption 2-(a) is essential to ensure bounded gradients for a first-order optimiser. Assumption 2-(b) is equivalent to the bounded sensitivity condition of the environmental dynamics in control theory (He et al., 2025) and smoothness in optimisation (Nesterov, 2018; Mahdavi et al., 2013).

Classical RL requires the value function to be smooth in the parameterisation of $\pi$ to ensure existence of gradients. Similarly in PRL, we require smoothness of log-policy, log-transitions, and rewards to prove convergence of PePG. This assumption resonates with the smoothness of objective functions with respect to the policy/decision parameters in performative learning (Perdomo et al., 2020; Izzo et al., 2021; Sahitaj et al., 2025) and decision-dependent optimization literature (He et al., 2025) that require smoothness of the objective function. Furthermore, we demonstrate that this assumption hold for PeMDPs with exponential family of transitions and linear rewards.

**Challenges and Three Step Analysis.** The main challenge to prove convergence of PePG arises due to non-concavity of the performative value function in the policy parametrisation $\boldsymbol{\theta}$. A similar issue occurs while proving convergence of PG-type algorithms in classical RL, which has been overcome by leveraging smoothness properties of the value functions and by deriving the local Polyak-Łojasiewicz (PL)-type conditions, known as *gradient domination* (Agarwal et al., 2019; Yuan et al., 2022). Extending these insights, we devise a three step convergence analysis for PePG.

**Step 1: Performative Gradient Domination.** First, we connect the performative performance difference (Lemma 2)

| Algorithms | Regulariser $\lambda$ | Min. #samples | Environment |
|---|---|---|---|
| RPO-FS (Mandal et al., 2023) | $\mathcal{O}\left(\frac{|\mathcal{S}|+\gamma|\mathcal{S}|^{5/2}}{(1-\omega)(1-\gamma)^4}\right)$ | $\mathcal{O}\left(\frac{|\mathcal{A}|^2|\mathcal{S}|^3}{\epsilon^4(1-\gamma)^6\lambda^2}\ln(\#\text{iter})\right)$ | Direct PeMDPs + quadratic-regul. on occupancy $\omega$-dependence between two envs. |
| MDRR (Rank et al., 2024) | $\mathcal{O}\left(\frac{|\mathcal{S}|+\gamma|\mathcal{S}|^{5/2}}{(1-\omega)(1-\gamma)^4}\right)$ | $\mathcal{O}\left(\frac{|\mathcal{A}|^2|\mathcal{S}|^3}{\epsilon^4(1-\gamma)^6\lambda^2}\ln(\#\text{iter})\right)$ | Direct PeMDPs + quadratic-regul. on occupancy $\omega$-dependence between two envs. |
| PePG (This work, Theorem 4 (b)) | $\frac{R_{\max}(1-\gamma)}{1+2\log(|\mathcal{A}|)}$ | $\mathcal{O}\left(\frac{|\mathcal{S}||\mathcal{A}|^2}{\epsilon^2(1-\gamma)^3}\right)$ | Exponential family PeMDPs + entropy regul. on policy |
| PePG (This work, Theorem 4 (a)) | $0$ | $\mathcal{O}\left(\frac{|\mathcal{S}||\mathcal{A}|}{\epsilon^2}\max\left\{\frac{\gamma R_{\max}|\mathcal{A}|}{(1-\gamma)^3},\frac{\gamma^2}{(1-\gamma)^4}\right\}\right)$ | Exponential family PeMDPs + no regularisation |

*Table 1.* Comparison of theoretical performance of SOTA stability-seeking algorithms and PePG.

with the norm of the performative policy gradient (Theorem 2). This allows us to connect the per-iteration improvement in the performative value function in PePG with the performative gradient ascent at that step.

> **Lemma 3** (Performative Gradient Domination Lemma).
> *Let* $\mathsf{Cov} \triangleq \max_{\boldsymbol{\theta},\boldsymbol{\nu}}\left\|\frac{d^{\pi_{\boldsymbol{\theta}}^*}_{\pi_{\boldsymbol{\theta}},\rho}}{d^{\pi_{\boldsymbol{\theta}}}_{\pi_{\boldsymbol{\theta}},\boldsymbol{\nu}}}\right\|_{\infty}$. *Then, under Assumption 1 and 2-(a), we get*
> *(a)* $\mathrm{SubOpt}(\boldsymbol{\pi_\theta}) \leq \sqrt{|\mathcal{S}||\mathcal{A}|}\mathsf{Cov}\|\nabla_{\boldsymbol{\theta}}V^{\boldsymbol{\pi_\theta}}_{\boldsymbol{\pi_\theta}}(\boldsymbol{\nu})\|_2 + \frac{1+\mathsf{Cov}}{(1-\gamma)^2}\left(L_r + L_{\mathbf{P}}R_{\max}\right)$ *for unregularised values, and*
> *(b)* $\mathrm{SubOpt}(\boldsymbol{\pi_\theta} \mid \boldsymbol{\lambda}) \leq \sqrt{|\mathcal{S}||\mathcal{A}|}\mathsf{Cov}\|\nabla_{\boldsymbol{\theta}}\tilde{V}^{\boldsymbol{\pi_\theta}}_{\boldsymbol{\pi_\theta}}(\boldsymbol{\nu})\|_2 + \frac{2+\mathsf{Cov}}{(1-\gamma)^2}\left(L_r + L_{\mathbf{P}}(R_{\max} + \lambda\log|\mathcal{A}|)\right)$ *for entropy-regularisation.*

**Step 2: Smoothness of Performative Value Functions.** Now, to properly stitch the gradient ascent steps over iterations, we prove that the unregularised performative value function is $\mathcal{O}(\frac{|\mathcal{A}|}{(1-\gamma)^2})$-smooth (Appendix E, Lemma 5). Further, we show that the entropy regularised performative value function is also $\mathcal{O}(\frac{|\mathcal{A}|}{(1-\gamma)^2})$-smooth as entropy is a $\mathcal{O}\left(\frac{\log|\mathcal{A}|}{(1-\gamma)^3}\right)$-smooth function for PeMDPs (Lemma 6).

**Step 3: Iterative Application of Gradient Domination for Smooth Objectives.** Finally, we apply gradient domination (Lemma 3) along with the iterative convergence proof of gradient ascent for smooth functions. The intuition is that since the per-step sub-optimality is dominated by the gradient and the smooth functions are bounded by quadratic envelopes of parameters, applying gradient ascent iteratively would bring the sub-optimality down to small error levels after enough iterations. We formally prove this in Theorem 3.

> **Theorem 3** (Convergence of PePG for Smooth PeMDPs).
> *We set learning rate* $\eta = \mathcal{O}\left(\frac{(1-\gamma)^2}{|\mathcal{A}|}\right)$. *Then,*
> $$\min_{t<T}\mathrm{SubOpt}(\boldsymbol{\pi}_{\boldsymbol{\theta}_t}) \leq \epsilon + \mathcal{O}\left(\frac{\mathsf{Cov}(L_r + L_{\mathbf{P}}R_{\max})}{(1-\gamma)^2}\right),$$
> *when (a)* $T = \Omega\left(\frac{|\mathcal{S}||\mathcal{A}|^2R_{\max}^2\mathsf{Cov}^2}{\epsilon^2(1-\gamma)^3}\right)$ *for unregularised objective, and*

> *(b)* $T = \Omega\left(\frac{|\mathcal{S}||\mathcal{A}|^2R_{\max}^2\mathsf{Cov}^2}{\epsilon^2(1-\gamma)^3}\right)$ *for entropy-regularised objective with* $\lambda = \frac{(1-\gamma)R_{\max}}{1+2\log|\mathcal{A}|}$.

**Implications.** *1. Efficiency Gain.* PePG converges to an $\epsilon$-optimal policy in $\mathcal{O}\left(\frac{|\mathcal{S}||\mathcal{A}|^2}{\epsilon^2(1-\gamma)^3}\right)$ iterations. *This reduces the sample complexity by at least* $\mathcal{O}\left(\frac{|\mathcal{S}|^2}{\epsilon^2(1-\gamma)^3}\right)$ in comparison to the existing stability-seeking algorithms that directly optimise the occupancy measures (Mandal et al., 2023; Mandal & Radanovic, 2024; Rank et al., 2024).

*2. Value of $\lambda$.* Regularisation parameters for the existing algorithms must be bigger than $\mathcal{O}\left(\frac{|\mathcal{S}|}{(1-\gamma)^4}\right)$. This counter-intuitive in practice. We show that setting $\lambda = \mathcal{O}\left(\frac{1}{\log|\mathcal{A}|}\right)$ suffices for proving convergence to $\epsilon$-optimality.

3. *Coverage Condition.* Minimum number of samples required to achieve convergence is proportional to $\mathsf{Cov}^2$ for smooth PeMDPs. This is a ubiquitous quantity dictating convergence of PG-methods in classical RL (Agarwal et al., 2021; Mei et al., 2020), and retraining methods in performative RL (Mandal et al., 2023; Rank et al., 2024) as we have to pay for the data coverage of the optimal policy with respect to that of the policy played at any iteration.

4. *Bias & Suboptimality Gap.* The $\mathcal{O}\left(\frac{\mathsf{Cov}}{(1-\gamma)^2}\right)$ bias appearing in Theorem 3 is analogous to the effect of using relaxed weak gradient domination result (Yuan et al., 2022, Corollary 3.7). It argues that if the policy gradient in classical MDPs satisfies $\epsilon' + \|\nabla_{\boldsymbol{\theta}}V(\theta)\| \geq 2\sqrt{\mu}\,\mathrm{SubOpt}(\boldsymbol{\pi_\theta})$ for some $\mu > 0$ and $\epsilon' > 0$, then the corresponding PG algorithm guarantees $\min_{t<T}\mathrm{SubOpt}(\boldsymbol{\pi}_{\boldsymbol{\theta}_t}) \leq \mathcal{O}(\epsilon + \epsilon')$ for big enough $T$. Lemma 3 constructs the performative counterpart of this relaxed weak gradient domination with $\epsilon' = \mathcal{O}\left(\frac{\mathsf{Cov}}{(1-\gamma)^2}\right)$. Note that this effect vanishes for PePG if the Lipschitz constants $L_r$ and $L_{\mathbf{P}}$ in Lemma 2 decay to 0, i.e., if the performative effect on environment stabilises with time.

**Convergence of PePG for Exponential Family PeMDPs.** Though for our proposed convergence analysis, it is enough to assume smooth parametrisation of transitions and rewards without fixing a specific parametric family, we further spe-

cialise our analysis for *exponential family PeMDPs* to (a) show exact gradient computations, and (b) establish an explicit relation between parametrisation of PeMDP and smoothness of performative value functions.

The exponential family PeMDPs have an exponential family distribution as transition functions with non-negative feature map $\psi(\cdot) : \mathcal{S} \to \mathbb{R}$, and linear reward functions with respect to the policy parameters. Specifically, $\{\mathcal{M}(\boldsymbol{\theta}) = \mathcal{M}(\boldsymbol{\pi_\theta}) \mid \boldsymbol{\theta} \in \mathbb{R}^{|\mathcal{S}| \times |\mathcal{A}|}\}$ such that, $\forall s \in \mathcal{S}$ and $a \in \mathcal{A}$, $r_{\boldsymbol{\pi_\theta}}(s,a) = \mathcal{P}_{[-R_{\max}, R_{\max}]}[\xi \boldsymbol{\theta}_{s,a}]$, and $\mathbf{P}_{\boldsymbol{\pi_\theta}}(s'|s,a) = \exp\left(\boldsymbol{\theta}_{s',a}\psi(s') - \log(\sum_{s''} e^{\boldsymbol{\theta}_{s,a}\psi(s'')})\right)$, where $\psi(\cdot) \leq \psi_{\max}$, and $\xi \in [0, R_{\max}]$ to align with Assumption 1. Note that modelling the Markovian transition function as exponential families is common in bandits and RL theory (Moulos, 2019; Al Marjani et al., 2021; Ouhamma et al., 2023; Karthik et al., 2024).

Instantiating the three step analysis of smooth PeMDPs, we prove convergence of PePG for exponential family PeMDPs. We tabulate the sample complexity of PePG for this parametrisation in Table 1. Corollary 1-2 and Theorem 4 show that smoothness guaranties of smooth PeMDP are retained here, though we observe a gain in terms of per-step improvement (gradient domination), and thus, in the achievable sub-optimality gap.

> **Theorem 4** (Convergence of PePG for Exponential family PeMDPs). *We set learning rate $\eta = \mathcal{O}\left(\frac{(1-\gamma)^2}{|\mathcal{A}|}\right)$, and $\psi_{\max} = \mathcal{O}(\frac{1-\gamma}{\gamma})$. Then,*
>
> $$\min_{t<T} \mathrm{SubOpt}(\boldsymbol{\pi}_{\boldsymbol{\theta}_t}) \leq \epsilon + \mathcal{O}\left(\frac{R_{\max}}{1-\gamma}\right),$$
>
> *when –*
> *(a) $T = \Omega\left(\frac{|\mathcal{S}||\mathcal{A}|R_{\max}\mathsf{Cov}^2}{\epsilon^2(1-\gamma)^3} \max\left\{R_{\max}|\mathcal{A}|, \frac{\gamma}{(1-\gamma)}\right\}\right)$ for unregularised objective, and*
> *(b) $T = \Omega\left(\frac{|\mathcal{S}||\mathcal{A}|^2 R_{\max}^2 \mathsf{Cov}^2}{\epsilon^2(1-\gamma)^3}\right)$ for entropy-regularised objective with $\lambda = \frac{(1-\gamma)R_{\max}}{1+2\log|\mathcal{A}|}$.*

**Discussions.** *1. Reduction of Bias in Performative Gradient Domination.* For exponential family PeMDPs with softmax policy class, we observe an improvement of $\mathcal{O}\left(\frac{\mathsf{Cov}}{(1-\gamma)}\right)$ in the per-step bias term compared to Lemma 3.

> **Lemma 4** (Performative Gradient Domination for Exponential Family PeMDPs with Softmax Policy Class). *(a) For unregularised value function,*
>
> $$\mathrm{SubOpt}(\boldsymbol{\pi_\theta}) \leq \sqrt{|\mathcal{S}||\mathcal{A}|}\mathsf{Cov}\|\nabla_{\boldsymbol{\theta}} V_{\boldsymbol{\pi_\theta}}^{\boldsymbol{\pi_\theta}}(\boldsymbol{\nu})\|_2 + \frac{R_{\max}}{1-\gamma}.$$

> *(b) For entropy-regularisation,* $\mathrm{SubOpt}(\boldsymbol{\pi_\theta}|\lambda) \leq$
>
> $$\sqrt{|\mathcal{S}||\mathcal{A}|}\mathsf{Cov}\|\nabla_{\boldsymbol{\theta}} V_{\boldsymbol{\pi_\theta}}^{\boldsymbol{\pi_\theta}}(\boldsymbol{\nu})\|_2 + \frac{R_{\max} + \lambda\log|\mathcal{A}|}{1-\gamma}.$$

*2. Improvement over the SOTA in Performative RL.* Theorem 3, and 4 indicate that PePG reaches closer to optimality whereas a stability seeking algorithm can yield policies which are $\mathcal{O}\left(\mathsf{Cov}/(1-\gamma)^4\right)$ far from the optimal value function (Mandal & Radanovic, 2024, Theorem 7). We find the existing algorithms to achieve $\mathcal{O}\left(1/(1-\gamma)^6\right)$ (Mandal et al., 2023) and $\mathcal{O}\left(1/(1-\gamma)^8\right)$ (Sahitaj et al., 2025) sub-optimality in different settings of performative RL.[3] This emphasises on the lacunae of stability seeking algorithms and effectiveness of PePG to achieve optimality.

## 5. Experimental Analysis

In this section, we empirically compare the performance of PePG and analyse its behaviour against the state-of-the-art stability-seeking algorithms.[4]

**Baselines and Experimental Setup.** We evaluate PePG (with and without entropy regularisation) alongside MDRR (Rank et al., 2024) and RPO-FS (Mandal et al., 2023), which represent the current state-of-the-art in performative RL under gradually shifting environments. MDRR has demonstrated significant improvements over traditional repeated retraining methods like RPO-FS, by leveraging historical data from multiple deployments. We refer to Appendix J for details on the setup and the environment.

**Results and Observations.** Experimental evaluation across 1000 iterations reveals fundamental differences between PePG , MDRR and RPO-FS in the performative setting.

**I. Optimality.** Left panel of (Figure 2) reveals a clear performance hierarchy among the three methods. PePG achieves the highest value function performance, reaching approximately 18 for standard PePG and 11 for regularized PePG (Reg PePG), both showing consistent improvement from initial values around 0. This sustained upward progression over iterations highlights PePG's effectiveness in discovering better performative equilibria rather than settling for the first stable solution encountered like MDRR or RPO-FS.

**II. Comparison of Optimality- and Stability-seeking Algorithms.** Right panel of Figure 2 exposes a critical limitation of algorithms designed primarily for stability rather than optimality. RPO-FS remains most stable with successive occupancy differences below $10^{-14}$ throughout training,

---

[3]We omit coverage term as its expression varies with settings.
[4]Code of PePG is available in https://github.com/brahimdriss/PePG. Further ablation studies with respect to hyperparameters are in Appendix K.

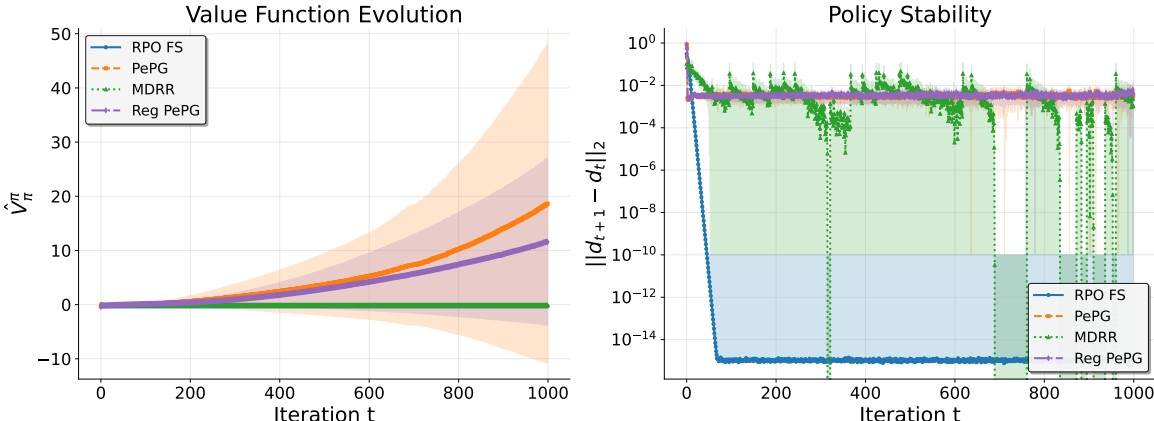

*Figure 2.* Comparison of evolution in expected average return (both regularised and unregularised) and stability of PePG with SOTA stability-achieving methods. Each algorithm is run for 20 random seeds and 1000 iterations.

achieving its design goal of rapid convergence. However, this stability comes at the cost of solution quality, as RPO-FS remains at the lowest performance level near 0. MDRR shows intermediate behaviour with stability fluctuations around $10^{-2}$ to $10^{-4}$, punctuated by occasional sharp drops to near $10^{-14}$ (approximately every 200-300 iterations). Despite these periodic stabilization events, MDRR does not achieve notable performance improvement, demonstrating that stability-focused methods can get trapped at suboptimal equilibria. In contrast, both PePG variants maintain consistent moderate variability around $10^{-2}$ throughout the 1000 iterations, indicating persistent exploration without convergence. These results demonstrate that optimality-seeking algorithms benefit from maintaining policy variability rather than prematurely stabilizing, even though this comes with higher occupancy measure distances between iterations. Also, PePG exhibits marginally better performance compared to its regularised variant albeit with larger variance across its runs. Note, we measure stability of PePG by $\|d_{t+1} - d_t\|_2$ to align with stability-notion of the baselines, as they optimise the occupancy measure per step.

**III. Computational Gain.** In Appendix J, we further show that PePG incurs $\sim 1.25 - 2.5\times$ less computational time per iteration in comparison with the baselines.

## 6. Discussions, Limitations, and Future Works

We study the problem of Performative Reinforcement learning in tabular MDPs, where the agent's actions cause potential shift in underlying reward and transition dynamics. First, we derive the novel performative counterpart of classic Performance Difference Lemma and Policy Gradient Theorem that successfully capture the performative dynamics of the underlying environment. Next, we propose the first PG algorithm, PePG, which attains $\epsilon$-performative optimality unlike the existing stability-seeking algorithms, affirmatively solving an extended open problem in (Mandal et al., 2023).

Our analysis requires smoothness of rewards and transitions with respect to the deployed policies. Though this holds true for multiple families of PeMDPs, it breaks if we encounter Bernoulli rewards and piecewise transitions. Thus, it would be interesting to extend this analysis for non-smooth PeMDPs. Another interesting future direction is to scale PePG for continuous state-action space while incorporating variance-reduction techniques (Wu et al., 2018; Papini et al., 2018). Finally, the gridworld environment is the only benchmark currently available for testing performative RL algorithms. In this direction, constructing a performative test-bed or simulator for both discrete and continuous state-action spaces, is an important future work.

## Acknowledgements

The authors would like to acknowledge the Inria-ISI Kolkata associate team SeRAI for supporting the collaboration. DB and UD would also like to acknowledge ANR JCJC project REPUBLIC (ANR-22-CE23-0003-01) and PEPR project FOUNDRY (ANR23-PEIA-0003). BD is also partially supported by the project ANR JCJC project NeuRL (ANR-23-CE23-0006). UM would like to acknowledge "Verified Deep Learning: Formal Methods Perspective" project under Ansuman Banerjee of ACMU, Indian Statistical Institute, Kolkata.

## Impact Statement

This paper presents an algorithmic and theoretical study whose goal is to advance the field of performative reinforcement learning. There are many potential algorithmic decision making problems where performativity naturally appears leading to societal consequences of our work, none which we feel must be specifically highlighted here.

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

# Appendix

## Table of Contents

# A. Notations

| Notation | Description |
|---|---|
| $\mathcal{S}$ | state space |
| $\mathcal{A}$ | action space |
| $\gamma$ | discount factor |
| $\pi_{\boldsymbol{\theta}}$ | policy parametrized by $\boldsymbol{\theta}$ |
| $\Pi(\Theta)$ | policy space |
| $\mathbf{P}_{\boldsymbol{\pi}}$ | transition under the environment induced by policy $\boldsymbol{\pi}$ |
| $r_{\boldsymbol{\pi}}$ | reward under the environment induced by policy $\boldsymbol{\pi}$ |
| $\pi_s^{\star}$ | performatively stable policy |
| $\pi_o^{\star}$ | performatively optimal policy |
| $\mathbf{P}_{\boldsymbol{\pi}_o^{\star}}$ | reward under the environment induced by performatively optimal policy |
| $r_{\boldsymbol{\pi}_o^{\star}}$ | reward under the environment induced by performatively optimal policy |
| $d_{\boldsymbol{\pi}_o^{\star}}^{\boldsymbol{\pi}_o^{\star}}$ | state-action occupancy of optimal policy |
| $V_{\boldsymbol{\pi}_o^{\star}}^{\boldsymbol{\pi}_o^{\star}}$ | value function of optimal policy |
| $d_{\boldsymbol{\pi}_2}^{\boldsymbol{\pi}_1}$ | state-action occupancy of playing policy $\boldsymbol{\pi}_2$ in the environment induced by policy $\boldsymbol{\pi}_1$ |
| $V_{\boldsymbol{\pi}_1}^{\boldsymbol{\pi}_2}$ | value function for playing policy $\boldsymbol{\pi}_2$ in the environment induced by policy $\boldsymbol{\pi}_1$ |
| $Q_{\boldsymbol{\pi}_1}^{\boldsymbol{\pi}_2}$ | Q-value function for playing policy $\boldsymbol{\pi}_2$ in the environment induced by policy $\boldsymbol{\pi}_1$ |
| $A_{\boldsymbol{\pi}_1}^{\boldsymbol{\pi}_2}$ | advantage function for playing policy $\boldsymbol{\pi}_2$ in the environment induced by policy $\boldsymbol{\pi}_1$ |
| $\tilde{Q}_{\boldsymbol{\pi}_1}^{\boldsymbol{\pi}_2}$ | entropy regularised Q-value function for playing policy $\boldsymbol{\pi}_2$ in the environment induced by policy $\boldsymbol{\pi}_1$ |
| $\tilde{A}_{\boldsymbol{\pi}_1}^{\boldsymbol{\pi}_2}$ | entropy regularised advantage function for playing policy $\boldsymbol{\pi}_2$ in the environment induced by policy $\boldsymbol{\pi}_1$ |
| $\Delta_K$ | $K$-dimensional simplex |
| $\boldsymbol{\rho}$ | Initial state distribution $\in \Delta_{\mathcal{S}}$ |
| Cov | Coverage $\triangleq \max_{\boldsymbol{\theta},\nu} \left\| \dfrac{d_{\pi_{\boldsymbol{\theta}},\boldsymbol{\rho}}^{\pi_{\boldsymbol{\theta}}^{\star}}}{d_{\pi_{\boldsymbol{\theta}},\nu}^{\pi_{\boldsymbol{\theta}}}} \right\|_{\infty}$ |
| $\tilde{V}_{\boldsymbol{\pi}_1}^{\boldsymbol{\pi}_2}$ | entropy regularised value function for playing policy $\boldsymbol{\pi}_2$ in the environment induced by policy $\boldsymbol{\pi}_1$ |

# B. Details of the Toy Example: Loan Approvement Problem

**Environment.** We consider a simple setup, where a population of loan applicants are represented by a scalar feature $x \in \mathbb{R}$, such that $x \sim \mathcal{N}(\mu, \sigma^2)$, where $\mu$ is the mean and variance $\sigma^2 > 0$ is fixed and known.

**Bank's Policy.** The bank chooses a *threshold policy* parametrised by $\theta \in \mathbb{R}$. A loan is granted to an applicant with feature $x$, if $x \geq \theta$. For a smoothed analysis, we use a differentiable policy class: $\pi_\theta(x) = \sigma\big(k(x - \theta)\big)$, where $\sigma(z) = \frac{1}{1+e^{-z}}$ is the logistic sigmoid and $k > 0$ controls smoothness.

**Rewards.** If a loan is granted to applicant $x$, the bank receives a random payoff:

$$r(x) = \begin{cases} +R & \text{if applicant repays,} \\ \\ -L & \text{if applicant defaults,} \end{cases}$$

with repayment probability $\mathbb{P}(\text{repay} \mid x) = \sigma(\gamma x - c)$, where $\gamma > 0$ controls sensitivity and $c$ is a calibration constant. Thus, the expected reward from granting to $x$ is

$$u(x) = \sigma(\gamma x - c) \cdot R - \big(1 - \sigma(\gamma x - c)\big) \cdot L.$$

**Expected Utility.** Given feature distribution $x \sim \mathcal{N}(\mu, \sigma^2)$, the bank's expected utility for policy $\theta$ is

$$U(\theta, \mu) = \mathbb{E}_{x \sim \mathcal{N}(\mu, \sigma^2)}\big[\pi_\theta(x) \cdot u(x)\big].$$

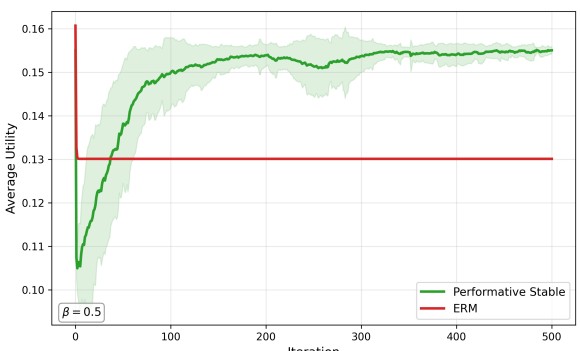

*Figure 3.* Average reward (over 10 runs) obtained by ERM optimal policies across performative strength $\beta$.

**Performative Feedback.** We define a grant rate: $g(\theta, \mu) = \mathbb{E}_{x \sim \mathcal{N}(\mu, \sigma^2)}\big[\pi_\theta(x)\big]$. We assume that dependence of $\mu$ on policy is tailored by a bounded performative update rule: $\mu_{t+1} = (1 - \beta)\mu_t + \beta \cdot f\big(g(\theta, \mu_t)\big)$, where $\beta \in [0, 1]$ is called the *Performative Strength* and $f(g) \in [-M, M]$ projects grant rate to a bounded domain.

Hence, at equilibrium, the induced feature distribution satisfies the fixed point condition:

$$\mu^*(\theta) = (1 - \beta)\mu^*(\theta) + \beta f\big(g(\theta, \mu^*(\theta))\big).$$

## B.1. Optimisation Problems

**ERM Optimum.** Ignoring performative effects (i.e. assuming $\mu = \mu_0$ is fixed), the ERM-optimal policy solves $\theta^{\text{ERM}} = \arg\max_\theta U(\theta, \mu_0)$.

**Performatively Optimal Optimisation.** Accounting for performative dynamics of the underlying environment, the performatively optimal policy solves $\theta^{\text{Perf}} = \arg\max_\theta U\big(\theta, \mu^*(\theta)\big)$.

In Algorithm 3, we present a simple agent-environment interaction/learning protocol under performative feedbacks.

---

**Algorithm 3** Learning Protocol of the Toy Example with Reinforcement Learning

---

1: **Input:** Performative strength $\beta$, $R, L, c, \gamma, M, k$, variance $\sigma^2 > 0$, .
2: **Initialize:** Policy parameter $\theta$ and initial value of mean $\mu_0$.
3: **for** $t = 1$ to T **do**
4:     Sample $x_t \sim \mathcal{N}(\mu_t, \sigma^2)$.
5:     Grant loan with probability $\pi_{\theta_t}(x_t)$.
6:     Observe reward $r_t$.
7:     Update $\theta_t$ using Policy Gradient (REINFORCE (Sutton et al., 1999; Agarwal et al., 2019)).
8:     Update population mean via performative dynamics: $\mu_{t+1} = (1 - \beta)\mu_t + \beta f\big(g(\theta_t, \mu_t)\big)$.
9: **end for**

---

# C. Extended Related Works

**Performative Prediction.** The study of performative prediction started with the pioneering work of (Perdomo et al., 2020), where they leveraged repeated retraining with the aim to converge towards a performatively stable point. We see extension of this work trying to achieve performative optimality (Izzo et al., 2021; 2022; Miller et al., 2021). This further opened a plethora of works in various other domains such as Multi-agent systems (Narang et al., 2023; Li et al., 2022; Piliouras & Yu, 2023), control systems (Cai et al., 2024; Barakat et al., 2025), stochastic optimisation (Li & Wai, 2022; Mendler-Dünner et al., 2020), games (Wang et al., 2023; Góis et al., 2024) etc. There has been several attempt towards achieving performative optimality or stability for real-life tasks like recommendation (Eilat & Rosenfeld, 2023), to measure the power of firms (Hardt et al., 2022; Mofakhami et al., 2023), in healthcare (Zhang et al., 2022) etc. Another interesting setting is the *stateful* performative prediction i.e. prediction under gradual shifts in the distribution (Brown et al., 2022; Izzo et al., 2022; Ray et al., 2022), that paved the way for incorporating performative prediction in Reinforcement Learning.

**Performative Reinforcement Learning.** (Bell et al., 2021) were the first to propose a setting where the transition and reward of an underlying MDP depend non-deterministically on the deployed policy, thus capturing the essence of performativity to some extent. However, (Mandal et al., 2023) can be considered the pioneer in introducing the notion of "*Performative Reinforcement Learning*" and its solution concepts, performatively stable and optimal policy. They propose direct optimization and ascent based techniques which manage to attain performative stability upon repeated retraining. Extensions to this work, (Rank et al., 2024) and (Mandal & Radanovic, 2024) manage to solve the same problem with delayed retraining for linear MDPs. However, there exists no literature that proposes a performative RL algorithm that converges to the performative optimal policy.

Specifically, (Mandal et al., 2023) frames the question of using policy gradient to find stable policies as an open problem. The authors further contemplate, as PG objective functions are non-concave in the policy space, whether it is possible to converge towards a stable policy. Thus, in this paper, we affirmatively solve an extension of this open problem for tabular PeMDPs.

**Policy Gradient Algorithms.** Policy gradient algorithms build a central paradigm in reinforcement learning, directly optimizing parametrised policies by estimating the gradient of expected return. The foundational policy gradient theorem (Sutton et al., 1999) established an expression for this gradient in terms of the score and action-value function, while Williams (1992) introduced the REINFORCE algorithm, providing an unbiased likelihood-ratio estimator. Convergence properties of stochastic gradient ascent in policy space were analysed in these early works. Subsequently, Konda & Tsitsiklis (2000) formalized actor–critic methods via two-timescale stochastic approximation, and Kakade (2002) proposed the natural policy gradient, leveraging the Fisher information geometry to accelerate learning. Extensions to trust region methods (Schulman et al., 2015), proximal policy optimization (Schulman et al., 2017), and entropy-regularized objectives (Mnih et al., 2016) have made policy gradient methods widely practical in high-dimensional settings. Recent theoretical advances provide finite-sample convergence guarantees and complexity analyses (Agarwal et al., 2021; Yuan et al., 2022), as well as robustness to distributional shift and adversarial perturbations (Zhang et al., 2020; Xu et al., 2020). Collectively, this body of work establishes policy gradient methods as both practically effective and theoretically grounded method for solving MDP.

## D. Impact of Policy Updates on PeMDPs (Section 3.1)

**Lemma 1** (Peformative Performance Difference Lemma). *The difference in performative value functions induced by $\boldsymbol{\pi}$ and $\boldsymbol{\pi}' \in \Pi$ while starting from the initial state distribution $\boldsymbol{\rho}$ is*

$$(1) \quad V_{\boldsymbol{\pi}}^{\boldsymbol{\pi}}(\boldsymbol{\rho}) - V_{\boldsymbol{\pi}'}^{\boldsymbol{\pi}'}(\boldsymbol{\rho}) = \frac{1}{1-\gamma} \mathop{\mathbb{E}}_{(s,a)\sim \boldsymbol{d}_{\boldsymbol{\pi}',\rho}^{\boldsymbol{\pi}}} [A_{\boldsymbol{\pi}'}^{\boldsymbol{\pi}'}(s,a)]$$

$$+ \frac{1}{1-\gamma} \mathop{\mathbb{E}}_{(s,a)\sim \boldsymbol{d}_{\boldsymbol{\pi}',\rho}^{\boldsymbol{\pi}}} \left[ (r_{\boldsymbol{\pi}}(s,a) - r_{\boldsymbol{\pi}'}(s,a)) + \gamma(\mathbf{P}_{\boldsymbol{\pi}}(\cdot|s,a) - \mathbf{P}_{\boldsymbol{\pi}'}(\cdot|s,a))^\top V_{\boldsymbol{\pi}}^{\boldsymbol{\pi}}(\cdot) \right]. \quad (9)$$

*where $A_{\boldsymbol{\pi}'}^{\boldsymbol{\pi}'}(s,a) \triangleq Q_{\boldsymbol{\pi}'}^{\boldsymbol{\pi}'}(s,a) - V_{\boldsymbol{\pi}'}^{\boldsymbol{\pi}'}(s)$ is the performative advantage function for any state $s \in \mathcal{S}$ and action $a \in \mathcal{A}$.*

$$(2) \quad V_{\boldsymbol{\pi}}^{\boldsymbol{\pi}}(\boldsymbol{\rho}) - V_{\boldsymbol{\pi}'}^{\boldsymbol{\pi}'}(\boldsymbol{\rho}) = \frac{1}{1-\gamma} \mathop{\mathbb{E}}_{(s,a)\sim \boldsymbol{d}_{\boldsymbol{\pi}',\rho}^{\boldsymbol{\pi}}} \left[ A_{\boldsymbol{\pi}'}^{\boldsymbol{\pi}'}(s,a) \right]$$

$$+ \frac{1}{1-\gamma} \mathop{\mathbb{E}}_{(s,a)\sim \boldsymbol{d}_{\boldsymbol{\pi},\rho}^{\boldsymbol{\pi}}} \left[ (r_{\boldsymbol{\pi}}(s,a) - r_{\boldsymbol{\pi}'}(s,a)) + \gamma(\mathbf{P}_{\boldsymbol{\pi}}(\cdot|s,a) - \mathbf{P}_{\boldsymbol{\pi}'}(\cdot|s,a))^\top V_{\boldsymbol{\pi}'}^{\boldsymbol{\pi}}(\cdot) \right]. \quad (10)$$

*where $A_{\boldsymbol{\pi}'}^{\boldsymbol{\pi}'}(s,a) \triangleq Q_{\boldsymbol{\pi}'}^{\boldsymbol{\pi}'}(s,a) - V_{\boldsymbol{\pi}'}^{\boldsymbol{\pi}'}(s)$ is the performative advantage function for any state $s \in \mathcal{S}$ and action $a \in \mathcal{A}$.*

$$(3) \quad V_{\boldsymbol{\pi}}^{\boldsymbol{\pi}}(\boldsymbol{\rho}) - V_{\boldsymbol{\pi}'}^{\boldsymbol{\pi}'}(\boldsymbol{\rho}) = \frac{1}{1-\gamma} \mathop{\mathbb{E}}_{(s,a)\sim \boldsymbol{d}_{\boldsymbol{\pi},\rho}^{\boldsymbol{\pi}}} \left[ A_{\boldsymbol{\pi}}^{\boldsymbol{\pi}'}(s,a) \right]$$

$$+ \frac{1}{1-\gamma} \mathop{\mathbb{E}}_{(s,a)\sim \boldsymbol{d}_{\boldsymbol{\pi}',\rho}^{\boldsymbol{\pi}'}} \left[ (r_{\boldsymbol{\pi}}(s,a) - r_{\boldsymbol{\pi}'}(s,a)) + \gamma(\mathbf{P}_{\boldsymbol{\pi}}(\cdot|s,a) - \mathbf{P}_{\boldsymbol{\pi}'}(\cdot|s,a))^\top V_{\boldsymbol{\pi}}^{\boldsymbol{\pi}'}(\cdot) \right]. \quad (11)$$

*where $A_{\boldsymbol{\pi}}^{\boldsymbol{\pi}'}(s,a) \triangleq Q_{\boldsymbol{\pi}}^{\boldsymbol{\pi}'}(s,a) - V_{\boldsymbol{\pi}}^{\boldsymbol{\pi}'}(s)$ is the performative advantage function for any state $s \in \mathcal{S}$ and action $a \in \mathcal{A}$.*

We only use the first version of this lemma in the main draft, and also hereafter, for the proofs.

*Proof of Lemma 1.* We do this proof in two steps. First step involves a decomposition of the difference in value function into two terms : (i) difference in value function after deploying the same policy while agent plays two different policies i.e. the difference that explains stability of the deployed policy, and (ii) difference in value function for deploying two different policies i.e. performance difference for changing the deployed policy. While the second term can be bounded using classic performance difference lemma, in the next and final step, we control the stability inducing term (i).

*Part(1)* – **Step 1: Decomposition.** We start by decomposing the performative performance difference to get a stability and a performance difference terms separately.

$$V_{\boldsymbol{\pi}}^{\boldsymbol{\pi}}(\boldsymbol{\rho}) - V_{\boldsymbol{\pi}'}^{\boldsymbol{\pi}'}(\boldsymbol{\rho}) = \underbrace{V_{\boldsymbol{\pi}}^{\boldsymbol{\pi}}(\boldsymbol{\rho}) - V_{\boldsymbol{\pi}'}^{\boldsymbol{\pi}}(\boldsymbol{\rho})}_{\text{performative shift term}} + \underbrace{V_{\boldsymbol{\pi}'}^{\boldsymbol{\pi}}(\boldsymbol{\rho}) - V_{\boldsymbol{\pi}'}^{\boldsymbol{\pi}'}(\boldsymbol{\rho})}_{\text{performance difference term}}$$

$$= V_{\boldsymbol{\pi}}^{\boldsymbol{\pi}}(\boldsymbol{\rho}) - V_{\boldsymbol{\pi}'}^{\boldsymbol{\pi}}(\boldsymbol{\rho}) + \frac{1}{1-\gamma} \mathop{\mathbb{E}}_{(s,a)\sim \boldsymbol{d}_{\boldsymbol{\pi}',\rho}^{\boldsymbol{\pi}}} [A_{\boldsymbol{\pi}'}^{\boldsymbol{\pi}'}(s,a)] \quad (12)$$

The last equality is a consequence of the classical performance difference lemma (Kakade & Langford, 2002b).
**Step 2: Controlling the performative shift term.** First, let us define $\mathbf{P}_{\boldsymbol{\pi}}^{\boldsymbol{\pi}}(s',s) \triangleq \sum_{a\in\mathcal{A}} \mathbf{P}_{\boldsymbol{\pi}}(s'|s,a)\boldsymbol{\pi}(a|s)$, and $\langle \mathbf{P}_{\boldsymbol{\pi}}^{\boldsymbol{\pi}}(\cdot,s_0), V_{\boldsymbol{\pi}}^{\boldsymbol{\pi}}(\cdot) \rangle \triangleq \sum_{s\in\mathcal{S}} V_{\boldsymbol{\pi}}^{\boldsymbol{\pi}}(s)\mathbf{P}_{\boldsymbol{\pi}}^{\boldsymbol{\pi}}(s,s_0)$, for some $s_0 \sim \boldsymbol{\rho}$.

We first observe that

$$V_{\boldsymbol{\pi}}^{\boldsymbol{\pi}}(s_0) - V_{\boldsymbol{\pi}'}^{\boldsymbol{\pi}}(s_0) = \mathop{\mathbb{E}}_{a\sim\boldsymbol{\pi}(\cdot|s_0)} \left[ r_{\boldsymbol{\pi}}(s_0,a) - r_{\boldsymbol{\pi}'}(s_0,a) \right] + \gamma \mathop{\mathbb{E}}_{s\sim\mathbf{P}_{\boldsymbol{\pi}}^{\boldsymbol{\pi}}(\cdot,s_0)}[V_{\boldsymbol{\pi}}^{\boldsymbol{\pi}}(s)] - \gamma \mathop{\mathbb{E}}_{s\sim\mathbf{P}_{\boldsymbol{\pi}'}^{\boldsymbol{\pi}}(\cdot,s_0)}[V_{\boldsymbol{\pi}'}^{\boldsymbol{\pi}}(s)]$$

$$= \mathop{\mathbb{E}}_{a\sim\boldsymbol{\pi}(\cdot|s_0)} \left[ r_{\boldsymbol{\pi}}(s_0,a) - r_{\boldsymbol{\pi}'}(s_0,a) \right]$$

$$+ \gamma \sum_s \left( \mathbf{P}_{\boldsymbol{\pi}}^{\boldsymbol{\pi}}(s, s_0) - \mathbf{P}_{\boldsymbol{\pi}'}^{\boldsymbol{\pi}}(s, s_0) \right) V_{\boldsymbol{\pi}}^{\boldsymbol{\pi}}(s) + \gamma \sum_s \mathbf{P}_{\boldsymbol{\pi}'}^{\boldsymbol{\pi}}(s, s_0) \left( V_{\boldsymbol{\pi}}^{\boldsymbol{\pi}}(s) - V_{\boldsymbol{\pi}'}^{\boldsymbol{\pi}}(s) \right)$$

$$= \frac{1}{1 - \gamma} \mathop{\mathbb{E}}_{(s,a) \sim \boldsymbol{d}_{\boldsymbol{\pi}', s_0}^{\boldsymbol{\pi}}} \left[ r_{\boldsymbol{\pi}}(s, a) - r_{\boldsymbol{\pi}'}(s, a) + \gamma (\mathbf{P}_{\boldsymbol{\pi}}(\cdot | s, a) - \mathbf{P}_{\boldsymbol{\pi}'}(\cdot | s, a))^\top V_{\boldsymbol{\pi}}^{\boldsymbol{\pi}}(\cdot) \right]$$

The last equality is obtained by recurring the preceding step iteratively.

We combine **Steps 1** and **2** and take expectation over $s_0 \sim \boldsymbol{\rho}$, we get

$$V_{\boldsymbol{\pi}}^{\boldsymbol{\pi}}(\boldsymbol{\rho}) - V_{\boldsymbol{\pi}'}^{\boldsymbol{\pi}'}(\boldsymbol{\rho}) = \frac{1}{1 - \gamma} \mathop{\mathbb{E}}_{(s,a) \sim \boldsymbol{d}_{\boldsymbol{\pi}', \boldsymbol{\rho}}^{\boldsymbol{\pi}}} \left[ A_{\boldsymbol{\pi}'}^{\boldsymbol{\pi}'}(s, a) + (r_{\boldsymbol{\pi}}(s, a) - r_{\boldsymbol{\pi}'}(s, a)) + \gamma (\mathbf{P}_{\boldsymbol{\pi}}(\cdot | s, a) - \mathbf{P}_{\boldsymbol{\pi}'}(\cdot | s, a))^\top V_{\boldsymbol{\pi}}^{\boldsymbol{\pi}}(\cdot) \right].$$

*Part(2)* – The second equality is obtained by changing the Step 2 as follows:

$$V_{\boldsymbol{\pi}}^{\boldsymbol{\pi}}(\boldsymbol{\rho}) - V_{\boldsymbol{\pi}'}^{\boldsymbol{\pi}}(\boldsymbol{\rho}) = \mathop{\mathbb{E}}_{a \sim \boldsymbol{\pi}(\cdot | \boldsymbol{\rho})} \left[ r_{\boldsymbol{\pi}}(s_0, a) - r_{\boldsymbol{\pi}'}(s_0, a) \right] + \gamma \mathop{\mathbb{E}}_{s \sim \mathbf{P}_{\boldsymbol{\pi}}^{\boldsymbol{\pi}}(\cdot, s_0)} [V_{\boldsymbol{\pi}}^{\boldsymbol{\pi}}(s)] - \gamma \mathop{\mathbb{E}}_{s \sim \mathbf{P}_{\boldsymbol{\pi}'}^{\boldsymbol{\pi}}(\cdot, s_0)} [V_{\boldsymbol{\pi}'}^{\boldsymbol{\pi}}(s)]$$

$$= \mathop{\mathbb{E}}_{a \sim \boldsymbol{\pi}(\cdot | \boldsymbol{\rho})} \left[ r_{\boldsymbol{\pi}}(s_0, a) - r_{\boldsymbol{\pi}'}(s_0, a) \right]$$

$$+ \gamma \sum_s \left( \mathbf{P}_{\boldsymbol{\pi}}^{\boldsymbol{\pi}}(s, s_0) - \mathbf{P}_{\boldsymbol{\pi}'}^{\boldsymbol{\pi}}(s, s_0) \right) V_{\boldsymbol{\pi}'}^{\boldsymbol{\pi}}(s) + \gamma \sum_s \mathbf{P}_{\boldsymbol{\pi}}^{\boldsymbol{\pi}}(s, s_0) \left( V_{\boldsymbol{\pi}}^{\boldsymbol{\pi}}(s) - V_{\boldsymbol{\pi}'}^{\boldsymbol{\pi}}(s) \right)$$

$$\implies V_{\boldsymbol{\pi}}^{\boldsymbol{\pi}}(\boldsymbol{\rho}) - V_{\boldsymbol{\pi}'}^{\boldsymbol{\pi}'}(\boldsymbol{\rho}) = \frac{1}{1 - \gamma} \mathop{\mathbb{E}}_{(s,a) \sim \boldsymbol{d}_{\boldsymbol{\pi}', \boldsymbol{\rho}}^{\boldsymbol{\pi}}} \left[ A_{\boldsymbol{\pi}'}^{\boldsymbol{\pi}'}(s, a) \right]$$

$$+ \frac{1}{1 - \gamma} \mathop{\mathbb{E}}_{(s,a) \sim \boldsymbol{d}_{\boldsymbol{\pi}, \boldsymbol{\rho}}^{\boldsymbol{\pi}}} \left[ (r_{\boldsymbol{\pi}}(s, a) - r_{\boldsymbol{\pi}'}(s, a)) + \gamma (\mathbf{P}_{\boldsymbol{\pi}}(\cdot | s, a) - \mathbf{P}_{\boldsymbol{\pi}'}(\cdot | s, a))^\top V_{\boldsymbol{\pi}'}^{\boldsymbol{\pi}}(\cdot) \right].$$

The last equality is obtained by recurring the preceding step iteratively.

*Part(3)* – The third equality is obtained through the following steps.

$$V_{\boldsymbol{\pi}}^{\boldsymbol{\pi}}(\boldsymbol{\rho}) - V_{\boldsymbol{\pi}'}^{\boldsymbol{\pi}'}(\boldsymbol{\rho}) = V_{\boldsymbol{\pi}}^{\boldsymbol{\pi}}(\boldsymbol{\rho}) - V_{\boldsymbol{\pi}}^{\boldsymbol{\pi}'}(\boldsymbol{\rho}) + V_{\boldsymbol{\pi}}^{\boldsymbol{\pi}'}(\boldsymbol{\rho}) - V_{\boldsymbol{\pi}'}^{\boldsymbol{\pi}'}(\boldsymbol{\rho})$$

$$= \frac{1}{1 - \gamma} \mathop{\mathbb{E}}_{(s,a) \sim \boldsymbol{d}_{\boldsymbol{\pi}, \boldsymbol{\rho}}^{\boldsymbol{\pi}}} [A_{\boldsymbol{\pi}}^{\boldsymbol{\pi}'}(s, a)] + V_{\boldsymbol{\pi}}^{\boldsymbol{\pi}'}(\boldsymbol{\rho}) - V_{\boldsymbol{\pi}'}^{\boldsymbol{\pi}'}(\boldsymbol{\rho})$$

$$= \frac{1}{1 - \gamma} \mathop{\mathbb{E}}_{(s,a) \sim \boldsymbol{d}_{\boldsymbol{\pi}, \boldsymbol{\rho}}^{\boldsymbol{\pi}}} [A_{\boldsymbol{\pi}}^{\boldsymbol{\pi}'}(s, a)] + \mathop{\mathbb{E}}_{a \sim \boldsymbol{\pi}'(\cdot | \boldsymbol{\rho})} \left[ r_{\boldsymbol{\pi}}(s_0, a) - r_{\boldsymbol{\pi}'}(s_0, a) \right]$$

$$+ \gamma \sum_s \left( \mathbf{P}_{\boldsymbol{\pi}}^{\boldsymbol{\pi}'}(s, s_0) - \mathbf{P}_{\boldsymbol{\pi}'}^{\boldsymbol{\pi}'}(s, s_0) \right) V_{\boldsymbol{\pi}}^{\boldsymbol{\pi}'}(s) + \gamma \sum_s \mathbf{P}_{\boldsymbol{\pi}'}^{\boldsymbol{\pi}'}(s, s_0) \left( V_{\boldsymbol{\pi}}^{\boldsymbol{\pi}'}(s) - V_{\boldsymbol{\pi}'}^{\boldsymbol{\pi}'}(s) \right)$$

$$= \frac{1}{1 - \gamma} \mathop{\mathbb{E}}_{(s,a) \sim \boldsymbol{d}_{\boldsymbol{\pi}, \boldsymbol{\rho}}^{\boldsymbol{\pi}}} \left[ A_{\boldsymbol{\pi}}^{\boldsymbol{\pi}'}(s, a) \right]$$

$$+ \frac{1}{1 - \gamma} \mathop{\mathbb{E}}_{(s,a) \sim \boldsymbol{d}_{\boldsymbol{\pi}', \boldsymbol{\rho}}^{\boldsymbol{\pi}'}} \left[ (r_{\boldsymbol{\pi}}(s, a) - r_{\boldsymbol{\pi}'}(s, a)) + \gamma (\mathbf{P}_{\boldsymbol{\pi}}(\cdot | s, a) - \mathbf{P}_{\boldsymbol{\pi}'}(\cdot | s, a))^\top V_{\boldsymbol{\pi}}^{\boldsymbol{\pi}'}(\cdot) \right].$$

$$\square$$

**Lemma 2** (Bounding Performative Performance Difference for Gradually Shifting Environments). *Let us assume that both rewards and transitions are Lipschitz functions of policy, i.e.* $\|r_{\boldsymbol{\pi}} - r_{\boldsymbol{\pi}'}\|_1 \leq L_r \|\boldsymbol{\pi} - \boldsymbol{\pi}'\|_\infty$ *and* $\|\mathbf{P}_{\boldsymbol{\pi}} - \mathbf{P}_{\boldsymbol{\pi}'}\|_1 \leq L_{\mathbf{P}} \|\boldsymbol{\pi} - \boldsymbol{\pi}'\|_\infty$, *for some* $L_r, L_{\mathbf{P}} \geq 0$. *Then, under Assumption 1, the performative shift in the sub-optimality gap of a policy* $\boldsymbol{\pi}_{\boldsymbol{\theta}}$ *satisfies*

$$\left| \mathrm{SubOpt}(\boldsymbol{\pi}_{\boldsymbol{\theta}}) - \frac{1}{1 - \gamma} \mathop{\mathbb{E}}_{(s,a) \sim d_{\boldsymbol{\pi}_{\boldsymbol{\theta}}, \boldsymbol{\rho}}^{\boldsymbol{\pi}_{\boldsymbol{\theta}}}} [A_{\boldsymbol{\pi}_{\boldsymbol{\theta}}}^{\boldsymbol{\pi}_{\boldsymbol{\theta}}}(s, a)] \right| \leq \frac{2\sqrt{2}}{1 - \gamma} (L_r + \frac{\gamma}{1 - \gamma} L_{\mathbf{P}} R_{\max}) \mathop{\mathbb{E}}_{s_0 \sim \boldsymbol{\rho}} D_{\mathrm{H}} (\boldsymbol{\pi}_o^\star(\cdot | s_0) \| \boldsymbol{\pi}_{\boldsymbol{\theta}}(\cdot | s_0)), \quad (13)$$

*where* $D_{\mathrm{H}} (\mathbf{x} \| \mathbf{y})$ *denotes the Hellinger distance between* $\mathbf{x}$ *and* $\mathbf{y}$.

*Proof of Lemma 2.* We do this proof in three steps. We start from the final expression in Lemma 1, then in step 2 we impose bounds on reward and transition differences leveraging the Lipschitz assumption. Lastly, we bound the policy

difference in first order norm using relation between Total Variation (TV) and Hellinger distance.

**Step 1:** From Lemma 1, we get

$$\text{SubOpt}(\pi_{\theta} \mid \rho) = \frac{1}{1-\gamma} \underset{(s,a)\sim d_{\pi_{\theta},\rho}^{\pi_o^{\star}}}{\mathbb{E}} \left[ A_{\pi_{\theta}}^{\pi_{\theta}}(s,a) + (r_{\pi_o^{\star}}(s,a) - r_{\pi_{\theta}}(s,a)) + \gamma (\mathbf{P}_{\pi_o^{\star}}(\cdot|s,a) - \mathbf{P}_{\pi_{\theta}}(\cdot|s,a))^{\top} V_{\pi_o^{\star}}^{\pi_o^{\star}}(\cdot) \right].$$

Thus,

$$\left| \text{SubOpt}(\pi_{\theta}) - \frac{1}{1-\gamma} \underset{(s,a)\sim d_{\pi_{\theta},\rho}^{\pi_o^{\star}}}{\mathbb{E}} [A_{\pi_{\theta}}^{\pi_{\theta}}(s,a)] \right|$$

$$= \frac{1}{1-\gamma} \left| \underset{(s,a)\sim d_{\pi_{\theta},\rho}^{\pi_o^{\star}}}{\mathbb{E}} (r_{\pi_o^{\star}}(s,a) - r_{\pi_{\theta}}(s,a)) + \gamma (\mathbf{P}_{\pi_o^{\star}}(\cdot|s,a) - \mathbf{P}_{\pi_{\theta}}(\cdot|s,a))^{\top} V_{\pi_o^{\star}}^{\pi_o^{\star}}(\cdot) \right| \tag{14}$$

**Step 2:** Using Jensen's inequality together with the fact that $d_{\pi_{\theta}}^{\pi_o^{\star}}(s,a|\rho) \leq 1$, for rewards, we get

$$\left| \underset{(s,a)\sim d_{\pi_{\theta},\rho}^{\pi_o^{\star}}}{\mathbb{E}} \left[ r_{\pi_o^{\star}}(s,a) - r_{\pi_{\theta}}(s,a) \right] \right| \leq \underset{(s,a)\sim d_{\pi_{\theta},\rho}^{\pi_o^{\star}}}{\mathbb{E}} \left| r_{\pi_o^{\star}}(s,a) - r_{\pi_{\theta}}(s,a) \right| \leq \| r_{\pi_o^{\star}} - r_{\pi_{\theta}} \|_1$$

Similarly for transitions, we get

$$\left| \underset{(s,a)\sim d_{\pi_{\theta},\rho}^{\pi_o^{\star}}}{\mathbb{E}} \left[ (\mathbf{P}_{\pi_o^{\star}} - \mathbf{P}_{\pi_{\theta}})^{\top} V_{\pi}^{\pi} \right] \right| \leq \underset{(s,a)\sim d_{\pi_{\theta},\rho}^{\pi_o^{\star}}}{\mathbb{E}} \left| (\mathbf{P}_{\pi_o^{\star}} - \mathbf{P}_{\pi_{\theta}})^{\top} V_{\pi}^{\pi} \right|$$

$$\overset{(a)}{\leq} \underset{(s,a)\sim d_{\pi_{\theta},\rho}^{\pi_o^{\star}}}{\mathbb{E}} \left[ \| \mathbf{P}_{\pi_o^{\star}} - \mathbf{P}_{\pi_{\theta}} \|_1 \cdot \| V_{\pi_o^{\star}}^{\pi_o^{\star}} \|_{\infty} \right]$$

$$= \| \mathbf{P}_{\pi_o^{\star}} - \mathbf{P}_{\pi_{\theta}} \|_1 \cdot \| V_{\pi_o^{\star}}^{\pi_o^{\star}} \|_{\infty},$$

(a) holds due to Hölder's inequality.

Now, leveraging the triangle inequality and Lipschitzness assumption on reward and transitions, we further get

$$\left| \underset{(s,a)\sim d_{\pi_{\theta},\rho}^{\pi_o^{\star}}}{\mathbb{E}} \left[ r_{\pi_o^{\star}}(s,a) - r_{\pi_{\theta}}(s,a) + \gamma (\mathbf{P}_{\pi_o^{\star}} - \mathbf{P}_{\pi_{\theta}})^{\top} V_{\pi}^{\pi} \right] \right| \leq L_r \| \pi_o^{\star} - \pi_{\theta} \|_{\infty} + \gamma L_{\mathbf{P}} \left\| V_{\pi_o^{\star}}^{\pi_o^{\star}} \right\|_{\infty} \| \pi_o^{\star} - \pi_{\theta} \|_{\infty}$$

Finally, due to Assumption 1, we get $\left\| V_{\pi_o^{\star}}^{\pi_o^{\star}} \right\|_{\infty} \leq \frac{R_{\max}}{1-\gamma}$, and thus,

$$\left| \underset{(s,a)\sim d_{\pi_{\theta},\rho}^{\pi_o^{\star}}}{\mathbb{E}} \left[ r_{\pi_o^{\star}}(s,a) - r_{\pi_{\theta}}(s,a) + \gamma (\mathbf{P}_{\pi_o^{\star}} - \mathbf{P}_{\pi_{\theta}})^{\top} V_{\pi_o^{\star}}^{\pi_o^{\star}} \right] \right| \leq L_r \| \pi_o^{\star} - \pi_{\theta} \|_{\infty} + \frac{\gamma}{1-\gamma} L_{\mathbf{P}} R_{\max} \| \pi_o^{\star} - \pi_{\theta} \|_{\infty}$$

**Step 3:** We know $\| \pi_o^{\star} - \pi_{\theta} \|_{\infty} \leq \| \pi_o^{\star} - \pi_{\theta} \|_1 = 2\text{TV}\left( \pi_o^{\star} \parallel \pi_{\theta} \right) \leq 2\sqrt{2} D_{\text{H}}\left( \pi_o^{\star} \parallel \pi_{\theta} \right)$. Thus,

$$\left| \underset{(s,a)\sim d_{\pi_{\theta},\rho}^{\pi_o^{\star}}}{\mathbb{E}} \left[ r_{\pi_o^{\star}}(s,a) - r_{\pi_{\theta}}(s,a) + \gamma (\mathbf{P}_{\pi_o^{\star}} - \mathbf{P}_{\pi_{\theta}})^{\top} V_{\pi_o^{\star}}^{\pi_o^{\star}} \right] \right|$$

$$\leq 2\sqrt{2} \left( L_r + \frac{\gamma}{1-\gamma} L_{\mathbf{P}} R_{\max} \right) D_{\text{H}}\left( \pi_o^{\star}(\cdot \mid s_0) \parallel \pi_{\theta}(\cdot \mid s_0) \right) \tag{15}$$

We conclude this proof by putting the upper bound in Equation (15) in Equation (14) and taking expectation over $s_0 \sim \rho$ to get the desired expression.

$\square$

## E. Smoothness of Performative Value Function and Entropy Regulariser

**Lemma 5** (Performative Smoothness Lemma). *Let* $\boldsymbol{\pi}_\alpha \triangleq \boldsymbol{\pi}_{\theta+\alpha u}$*, and let* $V_\alpha^\alpha(s_0)$ *be the corresponding value at a fixed state* $s_0$*, i.e.,* $V_\alpha^\alpha(s_0) \triangleq V_{\boldsymbol{\pi}_\alpha}^{\boldsymbol{\pi}_\alpha}(s_0)$ *. If the following conditions hold true,*

$$\sum_{a \in \mathcal{A}} \left| \frac{\mathrm{d}\boldsymbol{\pi}_\alpha(a \mid s_0)}{\mathrm{d}\alpha} \right|_{\alpha=0} \right| \leq C_1, \quad \sum_{a \in \mathcal{A}} \left| \frac{\mathrm{d}^2\boldsymbol{\pi}_\alpha(a \mid s_0)}{\mathrm{d}\alpha^2} \right|_{\alpha=0} \right| \leq C_2, \sum_{s \in \mathcal{S}} \left| \frac{\mathrm{d}\mathbf{P}_\alpha(s \mid s_0, a_0)}{\mathrm{d}\alpha} \right|_{\alpha=0} \right| \leq T_1,$$

$$\sum_{s \in \mathcal{S}} \left| \frac{\mathrm{d}^2\mathbf{P}_\alpha(s \mid s_0, a_0)}{\mathrm{d}\alpha^2} \right|_{\alpha=0} \right| \leq T_2, \sum_{a \in \mathcal{A}} \left| \frac{\mathrm{d}r_\alpha(s_0, a)}{\mathrm{d}\alpha} \right|_{\alpha=0} \right| \leq R_1, \quad \sum_{a \in \mathcal{A}} \left| \frac{\mathrm{d}^2 r_\alpha(s_0, a)}{\mathrm{d}\alpha^2} \right|_{\alpha=0} \right| \leq R_2,$$

*we get*

$$\max_{\|u\|_2=1} \left\| \frac{\mathrm{d}^2 V_\alpha^\alpha(s_0)}{\mathrm{d}\alpha^2} \right|_{\alpha=0} \right\| \leq \frac{C_2}{1-\gamma} + 2C_1\beta_1 + C_2\beta_2 \triangleq L,$$

*where* $\beta_1 = \frac{\gamma}{(1-\gamma)^2}(C_1 + T_1) + \frac{R_1}{1-\gamma}$ *and* $\beta_2 = \frac{2\gamma^2}{(1-\gamma)^3}(C_1 + T_1)^2 + \frac{\gamma}{(1-\gamma)^2}(C_2 + 2C_1T_1 + T_2) + \frac{2\gamma R_1}{(1-\gamma)^2}(C_2 + 2C_1T_1 + T_2) + \frac{R_2}{1-\gamma} + \frac{\gamma C_1 R_1}{(1-\gamma)^2}$.

*Proof.* **Step 1:** To prove the second order smoothness of the value function we start by taking its second derivative. Consider the expected return under policy $\boldsymbol{\pi}_\alpha$:

$$V_\alpha^\alpha(s_0) = \sum_a \boldsymbol{\pi}_\alpha(a \mid s_0) Q_\alpha^\alpha(s_0, a)$$

Differentiating twice with respect to $\alpha$, we obtain:

$$\frac{\mathrm{d}^2 V_\alpha^\alpha(s_0)}{\mathrm{d}\alpha^2} = \sum_a \frac{\mathrm{d}^2\boldsymbol{\pi}_\alpha(a \mid s_0)}{\mathrm{d}\alpha^2} Q_\alpha^\alpha(s_0, a) + 2\sum_a \frac{\mathrm{d}\boldsymbol{\pi}_\alpha(a \mid s_0)}{\mathrm{d}\alpha} \frac{\mathrm{d}Q_\alpha^\alpha(s_0, a)}{\mathrm{d}\alpha} + \sum_a \boldsymbol{\pi}_\alpha(a \mid s_0) \frac{\mathrm{d}^2 Q_\alpha^\alpha(s_0, a)}{\mathrm{d}\alpha^2}$$

$Q_\alpha^\alpha(s_0, a_0)$ is the Q-function corresponding to the policy $\boldsymbol{\pi}_\alpha$ at state $s_0$ and action $a_0$. Observe that $Q_\alpha^\alpha(s_0, a_0)$ can further be written as:

$$Q_\alpha^\alpha(s_0, a_0) = e_{(s_0, a_0)}^\top (I - \gamma\tilde{\mathbf{P}}(\alpha))^{-1} r_\alpha = e_{(s_0, a_0)}^\top M(\alpha) r_\alpha$$

where $M(\alpha) \triangleq (I - \gamma\mathbf{P}(\alpha))^{-1}$ and $\tilde{\mathbf{P}}(\alpha)$ is the state-action transition matrix under policy $\boldsymbol{\pi}_\alpha$, defined as:

$$[\tilde{\mathbf{P}}(\alpha)](s', a' \mid s, a) \triangleq \boldsymbol{\pi}_\alpha(a' \mid s') \mathbf{P}_\alpha(s' \mid s, a)$$

Differentiating $Q_\alpha^\alpha(s, a)$ with respect to $\alpha$ gives:

$$\frac{\mathrm{d}Q_\alpha^\alpha(s_0, a_0)}{\mathrm{d}\alpha} = \gamma e_{(s_0, a_0)}^\top M(\alpha) \frac{\mathrm{d}\tilde{\mathbf{P}}(\alpha)}{\mathrm{d}\alpha} M(\alpha) r_\alpha + e_{(s_0, a_0)}^\top M(\alpha) \frac{\mathrm{d}r_\alpha}{\mathrm{d}\alpha}$$

And correspondingly,

$$\begin{aligned}
\frac{\mathrm{d}^2 Q_\alpha^\alpha(s_0, a_0)}{\mathrm{d}\alpha^2} = {} & 2\gamma^2 e_{(s_0, a_0)}^\top M(\alpha) \frac{\mathrm{d}\tilde{\mathbf{P}}(\alpha)}{\mathrm{d}\alpha} M(\alpha) \frac{\mathrm{d}\tilde{\mathbf{P}}(\alpha)}{\mathrm{d}\alpha} M(\alpha) r_\alpha + \gamma e_{(s_0, a_0)}^\top M(\alpha) \frac{\mathrm{d}^2\tilde{\mathbf{P}}(\alpha)}{\mathrm{d}\alpha^2} M(\alpha) r_\alpha \\
& + \gamma e_{(s_0, a_0)}^\top M(\alpha) \frac{\mathrm{d}\tilde{\mathbf{P}}(\alpha)}{\mathrm{d}\alpha} M(\alpha) \frac{\mathrm{d}r_\alpha}{\mathrm{d}\alpha} + e_{(s_0, a_0)}^\top M(\alpha) \frac{\mathrm{d}^2 r_\alpha}{\mathrm{d}\alpha^2} \\
& + \gamma e_{(s_0, a_0)}^\top M(\alpha) \frac{\mathrm{d}\tilde{\mathbf{P}}(\alpha)}{\mathrm{d}\alpha} M(\alpha) \frac{\mathrm{d}r_\alpha}{\mathrm{d}\alpha}
\end{aligned} \tag{16}$$

**Step 2:** Now we need to find the derivative of $\tilde{\mathbf{P}}(\alpha)$ w.r.t $\alpha$ in order to substitute in (16). Hence, we can differentiate $\tilde{\mathbf{P}}(\alpha)$ with respect to $\alpha$ to obtain:

$$\frac{\mathrm{d}\tilde{\mathbf{P}}(\alpha)}{\mathrm{d}\alpha} \bigg|_{\alpha=0} (s', a' \mid s, a) = \frac{\mathrm{d}\boldsymbol{\pi}_\alpha(a' \mid s')}{\mathrm{d}\alpha} \bigg|_{\alpha=0} \mathbf{P}_\alpha(s' \mid s, a) + \frac{\mathrm{d}\mathbf{P}_\alpha(s' \mid s, a)}{\mathrm{d}\alpha} \bigg|_{\alpha=0} \boldsymbol{\pi}_\alpha(a' \mid s')$$

Now, for an arbitrary vector $\mathbf{x}$, we have:

$$\left[ \left. \frac{\mathrm{d}\tilde{\mathbf{P}}(\alpha)}{\mathrm{d}\alpha} \right|_{\alpha=0} \mathbf{x} \right]_{(s,a)} = \sum_{s',a'} \left. \frac{\mathrm{d}\boldsymbol{\pi}_\alpha(a' \mid s')}{\mathrm{d}\alpha} \right|_{\alpha=0} \mathbf{P}_\alpha(s' \mid s, a) \mathbf{x}_{s',a'}$$

$$+ \sum_{s',a'} \left. \frac{\mathrm{d}\mathbf{P}_\alpha(s' \mid s, a)}{\mathrm{d}\alpha} \right|_{\alpha=0} \boldsymbol{\pi}_\alpha(a' \mid s') \mathbf{x}_{s',a'}$$

Taking the maximum over unit vectors $\mathbf{u}$ in $\ell_2$-norm:

$$\max_{\|\mathbf{u}\|_2=1} \left\| \left. \frac{\mathrm{d}\tilde{\mathbf{P}}(\alpha)}{\mathrm{d}\alpha} \right|_{\alpha=0} \mathbf{x} \right\|_\infty \leq \max_{s,a} \max_{\|\mathbf{u}\|_2=1} \left| \sum_{s',a'} \left. \frac{\mathrm{d}\boldsymbol{\pi}_\alpha(a' \mid s')}{\mathrm{d}\alpha} \right|_{\alpha=0} \mathbf{P}_\alpha(s' \mid s, a) \mathbf{x}_{s',a'} \right|$$

$$+ \max_{s,a} \max_{\|\mathbf{u}\|_2=1} \left| \sum_{s',a'} \left. \frac{\mathrm{d}\mathbf{P}_\alpha(s' \mid s, a)}{\mathrm{d}\alpha} \right|_{\alpha=0} \boldsymbol{\pi}_\alpha(a' \mid s') \mathbf{x}_{s',a'} \right|$$

$$\leq \max_{s,a} \sum_{s'} \mathbf{P}_\alpha(s' \mid s, a) \sum_{a'} \left| \left. \frac{\mathrm{d}\boldsymbol{\pi}_\alpha(a' \mid s')}{\mathrm{d}\alpha} \right|_{\alpha=0} \right| \cdot \|\mathbf{x}\|_\infty$$

$$+ \max_{s,a} \sum_{a'} \boldsymbol{\pi}_\alpha(a' \mid s') \sum_{s'} \left| \left. \frac{\mathrm{d}\mathbf{P}_\alpha(s' \mid s, a)}{\mathrm{d}\alpha} \right|_{\alpha=0} \right| \cdot \|\mathbf{x}\|_\infty$$

$$\leq (C_1 + T_1)\|\mathbf{x}\|_\infty$$

$$\leq C_1 + T_1 \tag{17}$$

Similarly, differentiating $\tilde{\mathbf{P}}(\alpha)$ twice w.r.t. $\alpha$, we get

$$\left[ \left. \frac{\mathrm{d}^2\tilde{\mathbf{P}}(\alpha)}{\mathrm{d}\alpha^2} \right|_{\alpha=0} \right]_{(s,a) \to (s',a')} = \left. \frac{\mathrm{d}^2\boldsymbol{\pi}_\alpha(a' \mid s')}{(\mathrm{d}\alpha)^2} \right|_{\alpha=0} \mathbf{P}_\alpha(s' \mid s, a) + \left. \frac{\mathrm{d}^2\mathbf{P}_\alpha(s' \mid s, a)}{\mathrm{d}\alpha^2} \right|_{\alpha=0} \boldsymbol{\pi}_\alpha(a' \mid s')$$

$$+ 2 \left. \frac{\mathrm{d}\boldsymbol{\pi}_\alpha(a' \mid s')}{\mathrm{d}\alpha} \right|_{\alpha=0} \left. \frac{\mathrm{d}\mathbf{P}_\alpha(s' \mid s, a)}{\mathrm{d}\alpha} \right|_{\alpha=0}$$

Hence, we can consider the following norm bound:

$$\max_{\|\mathbf{u}\|_2=1} \left\| \left. \frac{\mathrm{d}^2\tilde{\mathbf{P}}(\alpha)}{\mathrm{d}\alpha^2} \right|_{\alpha=0} \mathbf{x} \right\|_1 \leq C_2\|\mathbf{x}\|_\infty + 2C_1 T_1\|\mathbf{x}\|_\infty + T_2\|\mathbf{x}\|_\infty \leq C_2 + 2C_1 T_1 + T_2 \tag{18}$$

**Step 3:** Now we need to put the pieces back together in order to calculate the second derivative of $V_\alpha^\alpha$ w.r.t $\alpha$. Let us recall $M(\alpha)$. Using the power series expansion of the matrix inverse, we can write $M(\alpha)$ as:

$$M(\alpha) = (I - \gamma\tilde{\mathbf{P}}(\alpha))^{-1} = \sum_{n=0}^\infty \gamma^n \tilde{\mathbf{P}}(\alpha)^n$$

which implies that $M(\alpha) \geq 0$ (component-wise), and

$$M(\alpha)\mathbf{1} = \frac{1}{1-\gamma}\mathbf{1},$$

i.e., each row of $M(\alpha)$ is positive and sums to $\frac{1}{1-\gamma}$.

This implies:

$$\max_{\|u\|_2=1} \|M(\alpha)\mathbf{x}\|_\infty \leq \frac{1}{1-\gamma}\|\mathbf{x}\|_\infty.$$

This gives, using the expressions for $\frac{\mathrm{d}^2 Q_\alpha^\alpha(s_0, a_0)}{\mathrm{d}\alpha^2}$ and $\frac{\mathrm{d} Q_\alpha^\alpha(s_0, a_0)}{\mathrm{d}\alpha}$, an upper bound on their magnitudes based on $\|\mathbf{x}\|_\infty$ and constants arising from bounds on the derivatives of $\tilde{\mathbf{P}}(\alpha)$ and $r_\alpha$.

$$
\max_{\|\mathbf{u}\|_2=1} \left\| \frac{\mathrm{d}^2 Q_\alpha^\alpha(s_0, a_0)}{\mathrm{d}\alpha^2} \right\|_\infty
$$

$$
\leq 2\gamma^2 \left\| M(\alpha) \frac{\mathrm{d}\tilde{\mathbf{P}}(\alpha)}{\mathrm{d}\alpha} M(\alpha) \frac{\mathrm{d}\tilde{\mathbf{P}}(\alpha)}{\mathrm{d}\alpha} M(\alpha) r_\alpha \right\|_\infty + \gamma \left\| M(\alpha) \frac{\mathrm{d}^2\tilde{\mathbf{P}}(\alpha)}{\mathrm{d}\alpha^2} M(\alpha) r_\alpha \right\|_\infty
$$

$$
+ \gamma \left\| M(\alpha) \frac{\mathrm{d}^2\tilde{\mathbf{P}}(\alpha)}{\mathrm{d}\alpha^2} M(\alpha) \frac{\mathrm{d} r_\alpha}{\mathrm{d}\alpha} \right\|_\infty + \left\| M(\alpha) \frac{\mathrm{d}^2 r_\alpha}{\mathrm{d}\alpha^2} \right\|_\infty + 2\gamma \left\| M(\alpha) \frac{\mathrm{d}\tilde{\mathbf{P}}(\alpha)}{\mathrm{d}\alpha} M(\alpha) \frac{\mathrm{d} r_\alpha}{\mathrm{d}\alpha} \right\|_\infty
$$

Bounding using known bounds on transitions and rewards:

$$
\max_{\|\mathbf{u}\|_2=1} \left\| \frac{\mathrm{d}^2 Q_\alpha^\alpha(s_0, a_0)}{\mathrm{d}\alpha^2} \right\|_\infty \leq \frac{2\gamma^2}{(1-\gamma)^3}(C_1 + T_1)^2 + \frac{\gamma}{(1-\gamma)^2}(C_2 + 2C_1 T_1 + T_2)
$$

$$
+ \frac{2\gamma R_1}{(1-\gamma)^2}(C_2 + 2C_1 T_1 + T_2) + \frac{R_2}{1-\gamma} + \frac{\gamma C_1 R_1}{(1-\gamma)^2} = \beta_2
$$

Corresponding bound on the first derivative is:

$$
\max_{\|\mathbf{u}\|_2=1} \left\| \frac{\mathrm{d} Q_\alpha^\alpha(s_0, a_0)}{\mathrm{d}\alpha} \right\|_\infty \leq \gamma \left\| M(\alpha) \frac{\mathrm{d}\tilde{\mathbf{P}}(\alpha)}{\mathrm{d}\alpha} M(\alpha) \frac{\mathrm{d} r_\alpha}{\mathrm{d}\alpha} \right\|_\infty + \left\| M(\alpha) \frac{\mathrm{d} r_\alpha}{\mathrm{d}\alpha} \right\|_\infty
$$

$$
\leq \frac{\gamma}{(1-\gamma)^2}(C_1 + T_1) + \frac{R_1}{1-\gamma} = \beta_1
$$

**Step 4:** Finally, putting all the bounds together to evaluate the upper bound of the desired quantity, we get,

$$
\max_{\|\mathbf{u}\|_2=1} \left\| \frac{\mathrm{d}^2 V_\alpha^\alpha(s_0)}{\mathrm{d}\alpha^2} \right\|_\infty \leq \frac{C_2}{1-\gamma} + 2C_1 \beta_1 + \beta_2 \tag{19}
$$

$\square$

**Corollary 1** (Smoothness guaranty for Exponential PeMDPs)**.** *For exponential PeMDPs, we characterise*

$$
C_1 = 2, \quad C_2 = 6, \quad T_1 = \max_s |\psi(s)| \triangleq \psi_{\max}, \quad T_2 = \max_s |\psi(s)|^2, \quad R_1 = \xi|\mathcal{A}|, \quad R_2 = 0
$$

*Thus,*

$$
\max_{\|u\|_2=1} \left\| \frac{\mathrm{d}^2 V_\alpha^\alpha(s_0)}{\mathrm{d}\alpha^2} \bigg|_{\alpha=0} \right\| \leq \mathcal{O}\left( \max\left\{ \frac{\gamma R_{\max}|\mathcal{A}|}{(1-\gamma)^2}, \frac{\gamma^2}{(1-\gamma)^3} \right\} \right) \triangleq \mathcal{O}(L). \tag{20}
$$

*Proof.* We use the expressions already found in (37) to state the following:

$$
\sum_{a \in \mathcal{A}} \left| \frac{\mathrm{d}}{\mathrm{d}\alpha} \boldsymbol{\pi}_{\boldsymbol{\theta}+\alpha\mathbf{u}}(a \mid s) \bigg|_{\alpha=0} \right| \leq \sum_{a \in \mathcal{A}} \boldsymbol{\pi}_{\boldsymbol{\theta}}(a \mid s) \left| \mathbf{u}_s^\top (\mathbf{e}_a - \boldsymbol{\pi}(\cdot \mid s)) \right| \leq \max_{a \in \mathcal{A}} \left( \mathbf{u}_s^\top \mathbf{e}_a + \mathbf{u}_s^\top \boldsymbol{\pi}(\cdot \mid s) \right) \leq 2.
$$

Similarly, differentiating once again w.r.t. $\alpha$, we get

$$\sum_{a \in \mathcal{A}} \left| \frac{\mathrm{d}^2}{\mathrm{d}\alpha^2} \boldsymbol{\pi}_{\boldsymbol{\theta}+\alpha\mathbf{u}}(a \mid s) \right|_{\alpha=0} \right| \leq \max_{a \in \mathcal{A}} \left( \mathbf{u}_s^\top \mathbf{e}_a \mathbf{e}_a^\top \mathbf{u}_s + \mathbf{u}_s^\top \mathbf{e}_a \boldsymbol{\pi}(\cdot \mid s)^\top \mathbf{u}_s + \mathbf{u}_s^\top \boldsymbol{\pi}(\cdot \mid s) \mathbf{e}_a^\top \mathbf{u}_s \right.$$

$$\left. + 2\, \mathbf{u}_s^\top \boldsymbol{\pi}(\cdot \mid s) \boldsymbol{\pi}(\cdot \mid s)^\top \mathbf{u}_s + \mathbf{u}_s^\top \mathrm{diag}(\boldsymbol{\pi}(\cdot \mid s)) \mathbf{u}_s \right) \leq 6.$$

And hence for transition we get,

$$\sum_{s' \in \mathcal{S}} \left| \frac{\mathrm{d}}{\mathrm{d}\alpha} \mathbf{P}_{\boldsymbol{\pi}_{\boldsymbol{\theta}+\alpha\mathbf{u}}}(\cdot \mid s, a) \right|_{\alpha=0} \right| \leq \sum_{s' \in \mathcal{S}} \left( \left| \psi(s') \mathbf{P}_{\boldsymbol{\pi}_{\boldsymbol{\theta}}}(s' \mid s, a)\, \mathbf{u}_{s',a} \right| + \left| \mathbf{P}_{\boldsymbol{\pi}_{\boldsymbol{\theta}}}(s'|s, a) \sum_{s''} \mathbf{P}_{\boldsymbol{\pi}_{\boldsymbol{\theta}}}(s''|s, a)) \psi(s'') \mathbf{u}_{s'',a} \right| \right)$$

$$\leq 2 \max_s |\psi(s)| = \mathcal{O}\left( \psi_{\max} \right)$$

And similarly, it can be shown that:

$$\sum_{s' \in \mathcal{S}} \left| \frac{\mathrm{d}^2}{\mathrm{d}\alpha^2} \mathbf{P}_{\boldsymbol{\pi}_{\boldsymbol{\theta}+\alpha\mathbf{u}}}(\cdot \mid s, a) \right|_{\alpha=0} \right| \leq \mathcal{O}\left( \max_s |\psi(s)|^2 \right)$$

Similarly for rewards we get:

$$\sum_{a \in \mathcal{A}} \left| \frac{\mathrm{d}}{\mathrm{d}\alpha} r_{\boldsymbol{\pi}_{\boldsymbol{\theta}+\alpha\mathbf{u}}}(s, a) \right|_{\alpha=0} \right| \leq \xi |\mathcal{A}| \qquad , \qquad \sum_{a \in \mathcal{A}} \left| \frac{\mathrm{d}^2}{\mathrm{d}\alpha^2} r_{\boldsymbol{\pi}_{\boldsymbol{\theta}+\alpha\mathbf{u}}}(s, a) \right|_{\alpha=0} \right| = 0$$

Hence, we can use the following choice of constants for softmax parametrization,

$$C_1 = 2 \quad , \quad C_2 = 6$$
$$T_1 = \mathcal{O}(\max_s |\psi(s)|) \quad , \quad T_2 = \mathcal{O}(\max_s |\psi(s)|^2)$$
$$R_1 = \xi |\mathcal{A}| \quad , \quad R_2 = 0$$

to get the desired order of $\max_{\|u\|_2=1} \left\| \frac{\mathrm{d}^2 V_\alpha^\alpha(s_0)}{\mathrm{d}\alpha^2} \Big|_{\alpha=0} \right\|$.

$\square$

**Lemma 6** (Smoothness of Entropy Regularizer). *Define the discounted entropy regularizer as:*

$$\mathcal{H}_{\boldsymbol{\pi}_{\boldsymbol{\theta}_\alpha}}^{\boldsymbol{\pi}_{\boldsymbol{\theta}_\alpha}}(s) = \mathbb{E}_{\tau \sim \mathbf{P}_{\boldsymbol{\pi}}^{\boldsymbol{\pi}}} \left[ \sum_{t=0}^{\infty} -\gamma^t \log \boldsymbol{\pi}_{\boldsymbol{\theta}_\alpha}(a_t \mid s_t) \right]$$

*Under the same assumptions as Lemma 5, the following holds:*

$$\max_{\|u\|_2=1} \left\| \frac{\partial^2 \mathcal{H}_{\boldsymbol{\pi}_{\boldsymbol{\theta}_\alpha}}^{\boldsymbol{\pi}_{\boldsymbol{\theta}_\alpha}}(s)}{\partial \alpha^2} \Big|_{\alpha=0} \right\|_{\infty} \leq \beta_\lambda$$

*where*

$$\beta_\lambda = 2\gamma^2 \frac{3(1 + \log |\mathcal{A}|)}{1 - \gamma} + \gamma \frac{2 \log |\mathcal{A}|}{(1 - \gamma)^2}(C_1 + T_1) + 2\gamma \frac{\log |\mathcal{A}|}{(1 - \gamma)^2}(C_2 + 2C_1 T_1 + T_2) + \frac{\log |\mathcal{A}|}{(1 - \gamma)^3}(C_1 + T_1)^2 .$$

*Proof.* **Step 1:** Define the state-wise entropy term:

$$h_{\boldsymbol{\theta}_\alpha}(s) = -\sum_a \boldsymbol{\pi}_{\boldsymbol{\theta}_\alpha}(a \mid s) \log \boldsymbol{\pi}_{\boldsymbol{\theta}_\alpha}(a \mid s).$$

From (Mei et al., 2020) (Lemma 7) we report that,

$$\left\|\frac{\partial h_{\boldsymbol{\theta}_\alpha}}{\partial \alpha}\right\|_\infty \leq 2 \cdot \log|\mathcal{A}| \cdot \|u\|_2, \qquad \left\|\frac{\partial^2 h_{\boldsymbol{\theta}_\alpha}}{\partial \alpha^2}\right\|_\infty \leq 3 \cdot (1 + \log|\mathcal{A}|) \cdot \|\mathbf{u}\|_2^2. \tag{21}$$

Additionally, (Mei et al., 2020) also presents a second result expressing the second derivative of the entropy w.r.t $\alpha$,

$$\frac{\partial^2 \mathcal{H}_{\boldsymbol{\pi}_{\boldsymbol{\theta}_\alpha}}^{\boldsymbol{\pi}_{\boldsymbol{\theta}_\alpha}}(s)}{\partial \alpha^2} = 2\gamma^2\, \mathbf{e}_s^\top M(\alpha) \frac{\partial \mathbf{P}(\alpha)}{\partial \alpha} M(\alpha) \frac{\partial \mathbf{P}(\alpha)}{\partial \alpha} M(\alpha) h_{\boldsymbol{\theta}_\alpha}$$
$$+ \gamma\, \mathbf{e}_s^\top M(\alpha) \frac{\partial^2 \mathbf{P}(\alpha)}{\partial \alpha^2} M(\alpha) h_{\boldsymbol{\theta}_\alpha} + 2\gamma\, \mathbf{e}_s^\top M(\alpha) \frac{\partial \mathbf{P}(\alpha)}{\partial \alpha} M(\alpha) \frac{\partial h_{\boldsymbol{\theta}_\alpha}}{\partial \alpha} + \mathbf{e}_s^\top M(\alpha) \frac{\partial^2 h_{\boldsymbol{\theta}_\alpha}}{\partial \alpha^2}.$$

**Step 2:** Now we proceed with bounding the absolute value of each term which will contribute towards bounding the overall second derivative of the regulariser.

For the last term,

$$\left|\mathbf{e}_s^\top M(\alpha) \frac{\partial^2 h_{\boldsymbol{\theta}_\alpha}}{\partial \alpha^2}\Big|_{\alpha=0}\right| \leq \|\mathbf{e}_s^\top\|_1 \cdot \left\|M(\alpha) \frac{\partial^2 h_{\boldsymbol{\theta}_\alpha}}{\partial \alpha^2}\Big|_{\alpha=0}\right\|_\infty$$
$$\leq \frac{1}{1-\gamma} \cdot \left\|\frac{\partial^2 h_{\boldsymbol{\theta}_\alpha}}{\partial \alpha^2}\Big|_{\alpha=0}\right\|_\infty$$
$$\leq \frac{3 \cdot (1 + \log|\mathcal{A}|)}{1-\gamma} \cdot \|\mathbf{u}\|_2^2.$$

For the second last term,

$$\left|\mathbf{e}_s^\top M(\alpha) \frac{\partial \mathbf{P}(\alpha)}{\partial \alpha} M(\alpha) \frac{\partial h_{\boldsymbol{\theta}_\alpha}}{\partial \alpha}\Big|_{\alpha=0}\right| \leq \left\|M(\alpha) \frac{\partial \mathbf{P}(\alpha)}{\partial \alpha} M(\alpha) \frac{\partial h_{\boldsymbol{\theta}_\alpha}}{\partial \alpha}\Big|_{\alpha=0}\right\|_\infty$$
$$\leq \frac{1}{1-\gamma} \cdot \left\|\frac{\partial \mathbf{P}(\alpha)}{\partial \alpha} M(\alpha) \frac{\partial h_{\boldsymbol{\theta}_\alpha}}{\partial \alpha}\Big|_{\alpha=0}\right\|_\infty$$
$$\leq \frac{(C_1 + T_1) \cdot \|u\|_2}{1-\gamma} \cdot \left\|M(\alpha) \frac{\partial h_{\boldsymbol{\theta}_\alpha}}{\partial \alpha}\Big|_{\alpha=0}\right\|_\infty$$
$$\leq \frac{(C_1 + T_1) \cdot \|\mathbf{u}\|_2}{(1-\gamma)^2} \cdot \left\|\frac{\partial h_{\boldsymbol{\theta}_\alpha}}{\partial \alpha}\Big|_{\alpha=0}\right\|_\infty$$
$$\leq \frac{2 \cdot \log|\mathcal{A}|}{(1-\gamma)^2}(C_1 + T_1) \cdot \|\mathbf{u}\|_2^2.$$

For the second term,

$$\left|\mathbf{e}_s^\top M(\alpha) \frac{\partial^2 \mathbf{P}(\alpha)}{\partial \alpha^2} M(\alpha) h_{\boldsymbol{\theta}_\alpha}\Big|_{\alpha=0}\right| \leq \left\|M(\alpha) \frac{\partial^2 \mathbf{P}(\alpha)}{\partial \alpha^2} M(\alpha) h_{\boldsymbol{\theta}_\alpha}\Big|_{\alpha=0}\right\|_\infty$$
$$\leq \frac{1}{1-\gamma} \cdot \left\|\frac{\partial^2 \mathbf{P}(\alpha)}{\partial \alpha^2} M(\alpha) h_{\boldsymbol{\theta}_\alpha}\Big|_{\alpha=0}\right\|_\infty$$
$$\leq \frac{\|\mathbf{u}\|_2^2}{1-\gamma} \cdot \left\|M(\alpha) h_{\boldsymbol{\theta}_\alpha}\Big|_{\alpha=0}\right\|_\infty (C_2 + 2C_1 T_1 + T_2)$$

$$\leq \frac{\|\mathbf{u}\|_2^2}{(1-\gamma)^2} \cdot \left\| h_{\boldsymbol{\theta}_\alpha}\Big|_{\alpha=0} \right\|_\infty (C_2 + 2C_1 T_1 + T_2)$$

$$\leq \frac{\log|\mathcal{A}|}{(1-\gamma)^2} (C_2 + 2C_1 T_1 + T_2) \cdot \|\mathbf{u}\|_2^2.$$

For the first term,

$$\left| \mathbf{e}_s^\top M(\alpha) \frac{\partial \mathbf{P}(\alpha)}{\partial \alpha} M(\alpha) \frac{\partial \mathbf{P}(\alpha)}{\partial \alpha} M(\alpha) h_{\boldsymbol{\theta}_\alpha}\Big|_{\alpha=0} \right| \leq \left\| M(\alpha) \frac{\partial \mathbf{P}(\alpha)}{\partial \alpha} M(\alpha) \frac{\partial \mathbf{P}(\alpha)}{\partial \alpha} M(\alpha) h_{\boldsymbol{\theta}_\alpha}\Big|_{\alpha=0} \right\|_\infty$$

$$\leq \frac{1}{1-\gamma} \cdot \|\mathbf{u}\|_2 \cdot \frac{1}{1-\gamma} \cdot \|\mathbf{u}\|_2 \cdot \frac{1}{1-\gamma} \cdot \log|\mathcal{A}|$$
$$\cdot (C_1 + T_1)^2$$

$$= \frac{\log|\mathcal{A}|}{(1-\gamma)^3} (C_1 + T_1)^2 \cdot \|\mathbf{u}\|_2^2.$$

**Step 3:** Now combining all the above equations, we get the final expression,

$$\max_{\|\mathbf{u}\|_2=1} \left\| \frac{\partial^2 \mathcal{H}_{\boldsymbol{\pi}_{\boldsymbol{\theta}_\alpha}}^{\boldsymbol{\pi}_{\boldsymbol{\theta}_\alpha}}(s)}{\partial \alpha^2}\Big|_{\alpha=0} \right\|_\infty \leq \beta_\lambda$$

where

$$\beta_\lambda = 2\gamma^2 \cdot \frac{3 \cdot (1 + \log|\mathcal{A}|)}{1-\gamma} + \gamma \cdot \frac{2 \cdot \log|\mathcal{A}|}{(1-\gamma)^2}(C_1 + T_1)$$
$$+ 2\gamma \cdot \frac{\log|\mathcal{A}|}{(1-\gamma)^2}(C_2 + 2C_1 T_1 + T_2) + \frac{\log|\mathcal{A}|}{(1-\gamma)^3}(C_1 + T_1)^2$$

□

**Corollary 2.** *For PeMDPs with Softmax policy class and exponential transitions, the following holds:*

$$\max_{\|u\|_2=1} \left\| \frac{\mathrm{d}^2 \tilde{V}_\alpha^\alpha(s_0)}{\mathrm{d}\alpha^2}\Big|_{\alpha=0} \right\| \leq \mathcal{O}\left( \max\left\{ \frac{\gamma R_{\max}|\mathcal{A}|}{(1-\gamma)^2}, \frac{\lambda \log|\mathcal{A}|\psi_{\max}^2}{(1-\gamma)^3} \right\} \right).$$

*Proof.* By definition of smoothness, the "soft performative value function" $\tilde{V}_{\boldsymbol{\pi}}^{\boldsymbol{\pi}}$ is Lipschitz smooth with Lipschitz constant $L_\lambda$ where $L_\lambda \triangleq L + \lambda\beta_\lambda$. Once again, we can choose $C_1, C_2, T_1, T_2$ according to Corollary 1 for simplification to get the order $\beta_\lambda = \mathcal{O}\left( \frac{\log|\mathcal{A}|}{(1-\gamma)^3}\psi_{\max}^2 \right)$. Thus, the final bound for $L_\lambda$ as

$$L_\lambda = \mathcal{O}\left( \max\{L, \lambda\beta_\lambda\} \right) = \mathcal{O}\left( \max\left\{ \frac{\gamma R_{\max}|\mathcal{A}|}{(1-\gamma)^2}, \frac{\lambda \log|\mathcal{A}|\psi_{\max}^2}{(1-\gamma)^3} \right\} \right). \tag{22}$$

□

# F. Derivation of Performative Policy Gradients

**Theorem 2** (Performative Policy Gradient Theorem). *The gradient of the performative value function w.r.t $\boldsymbol{\theta}$ is as follows:*

*(a) For the unregularised objective,*

$$\nabla_{\boldsymbol{\theta}} V_{\boldsymbol{\pi_\theta}}^{\boldsymbol{\pi_\theta}}(\tau) = \mathop{\mathbb{E}}_{\tau \sim \mathbb{P}_{\boldsymbol{\pi_\theta}}^{\boldsymbol{\pi_\theta}}} \left[ \sum_{t=0}^{\infty} \gamma^t \Big( A_{\boldsymbol{\pi_\theta}}^{\boldsymbol{\pi_\theta}}(s_t, a_t) \left( \nabla_{\boldsymbol{\theta}} \log \boldsymbol{\pi_\theta}(a_t \mid s_t) + \nabla_{\boldsymbol{\theta}} \log P_{\boldsymbol{\pi_\theta}}(s_{t+1}|s_t, a_t) \right) + \nabla_{\boldsymbol{\theta}} r_{\boldsymbol{\pi_\theta}}(s_t, a_t) \Big) \right].$$

*(b) For the entropy-regularised objective, we define the soft advantage, soft Q, and soft value functions with respect to the soft rewards $\tilde{r}_{\boldsymbol{\pi_\theta}}$ satisfying $\tilde{A}_{\boldsymbol{\pi_\theta}}^{\boldsymbol{\pi_\theta}}(s, a) = \tilde{Q}_{\boldsymbol{\pi_\theta}}^{\boldsymbol{\pi_\theta}}(s, a) - \tilde{V}_{\boldsymbol{\pi_\theta}}^{\boldsymbol{\pi_\theta}}(s)$ that further yields*

$$\nabla_{\boldsymbol{\theta}} \tilde{V}_{\boldsymbol{\pi_\theta}}^{\boldsymbol{\pi_\theta}}(\tau) = \mathop{\mathbb{E}}_{\tau \sim \mathbb{P}_{\boldsymbol{\pi_\theta}}^{\boldsymbol{\pi_\theta}}} \left[ \sum_{t=0}^{\infty} \gamma^t \Big( \tilde{A}_{\boldsymbol{\pi_\theta}}^{\boldsymbol{\pi_\theta}}(s_t, a_t) \left( \nabla_{\boldsymbol{\theta}} \log \boldsymbol{\pi_\theta}(a_t \mid s_t) + \nabla_{\boldsymbol{\theta}} \log P_{\boldsymbol{\pi_\theta}}(s_{t+1}|s_t, a_t) \right) + \nabla_{\boldsymbol{\theta}} \tilde{r}_{\boldsymbol{\pi_\theta}}(s_t, a_t|\boldsymbol{\theta}) \Big) \right].$$

*Proof of Theorem 2.* We prove each part of this theorem separately.

*Proof of part (a).* First, we derive explicit closed form gradient for unregularised performative value function.

**Step 1.** Given a trajectory $\tau = \{s_0, a_0, \ldots, s_t, a_t, \ldots\}$, let us denote the unregularised objective function as

$$f_{\boldsymbol{\theta}}(\tau) = \sum_{t=0}^{\infty} \gamma^t r_{\boldsymbol{\pi_\theta}}(s_t, a_t)$$

Thus,

$$\begin{aligned}
\nabla_{\boldsymbol{\theta}} V_{\boldsymbol{\pi_\theta}}^{\boldsymbol{\pi_\theta}}(\tau) = \nabla_{\boldsymbol{\theta}} \mathop{\mathbb{E}}_{\tau \sim \mathbb{P}_{\boldsymbol{\pi_\theta}}^{\boldsymbol{\pi_\theta}}}[f_{\boldsymbol{\theta}}(\tau)] &= \nabla_{\boldsymbol{\theta}} \sum_{\tau} \mathbb{P}_{\boldsymbol{\pi_\theta}}^{\boldsymbol{\pi_\theta}}(\tau) f_{\boldsymbol{\theta}}(\tau) \\
&= \sum_{\tau} \nabla_{\boldsymbol{\theta}}(\mathbb{P}_{\boldsymbol{\pi_\theta}}^{\boldsymbol{\pi_\theta}}(\tau) f_{\boldsymbol{\theta}}(\tau)) \\
&= \sum_{\tau} (\nabla_{\boldsymbol{\theta}} \mathbb{P}_{\boldsymbol{\pi_\theta}}^{\boldsymbol{\pi_\theta}}(\tau)) f_{\boldsymbol{\theta}}(\tau) + \sum_{\tau} \mathbb{P}_{\boldsymbol{\pi_\theta}}^{\boldsymbol{\pi_\theta}}(\tau)(\nabla_{\boldsymbol{\theta}} f_{\boldsymbol{\theta}}(\tau)) \\
&\stackrel{(a)}{=} \sum_{\tau} \mathbb{P}_{\boldsymbol{\pi_\theta}}^{\boldsymbol{\pi_\theta}}(\tau)(\nabla_{\boldsymbol{\theta}} \log \mathbb{P}_{\boldsymbol{\pi_\theta}}^{\boldsymbol{\pi_\theta}}(\tau)) f_{\boldsymbol{\theta}}(\tau) + \mathop{\mathbb{E}}_{\tau \sim \mathbb{P}_{\boldsymbol{\pi_\theta}}^{\boldsymbol{\pi_\theta}}}[\nabla_{\boldsymbol{\theta}} f_{\boldsymbol{\theta}}(\tau)] \\
&= \mathop{\mathbb{E}}_{\tau \sim \mathbb{P}_{\boldsymbol{\pi_\theta}}^{\boldsymbol{\pi_\theta}}} \left[ (\nabla_{\boldsymbol{\theta}} \log \mathbb{P}_{\boldsymbol{\pi_\theta}}^{\boldsymbol{\pi_\theta}}(\tau)) f_{\boldsymbol{\theta}}(\tau) \right] + \mathop{\mathbb{E}}_{\tau \sim \mathbb{P}_{\boldsymbol{\pi_\theta}}^{\boldsymbol{\pi_\theta}}}[\nabla_{\boldsymbol{\theta}} f_{\boldsymbol{\theta}}(\tau)].
\end{aligned}$$

$(a)$ holds since $\nabla_{\boldsymbol{\theta}} \log \mathbb{P}_{\boldsymbol{\pi_\theta}}^{\boldsymbol{\pi_\theta}}(\tau) = \frac{\nabla_{\boldsymbol{\theta}} \mathbb{P}_{\boldsymbol{\pi_\theta}}^{\boldsymbol{\pi_\theta}}(\tau)}{\mathbb{P}_{\boldsymbol{\pi_\theta}}^{\boldsymbol{\pi_\theta}}(\tau)}$.

**Step 2.** Given the initial state distribution $\boldsymbol{\rho}$, we further have

$$\log \mathbb{P}_{\boldsymbol{\pi_\theta}}^{\boldsymbol{\pi_\theta}}(\tau) = \log \boldsymbol{\rho}(s_0) + \sum_{t=0}^{\infty} \log \boldsymbol{\pi_\theta}(a_t \mid s_t) + \sum_{t=0}^{\infty} \log \mathbf{P}_{\boldsymbol{\pi_\theta}}(s_{t+1}|s_t, a_t)$$

Taking the gradient with respect to $\boldsymbol{\theta}$, we obtain

$$\nabla_{\boldsymbol{\theta}} \log \mathbb{P}_{\boldsymbol{\pi_\theta}}^{\boldsymbol{\pi_\theta}}(\tau) = \sum_{t=0}^{\infty} \nabla_{\boldsymbol{\theta}} \log \boldsymbol{\pi_\theta}(a_t \mid s_t) + \sum_{t=0}^{\infty} \nabla_{\boldsymbol{\theta}} \log \mathbf{P}_{\boldsymbol{\pi_\theta}}(s_{t+1}|s_t, a_t)$$

**Step 3.** Now, by substituting the value of $\nabla_{\boldsymbol{\theta}} \log(\mathbf{P}_{\boldsymbol{\pi_\theta}}^{\boldsymbol{\pi_\theta}})$ in $\nabla_{\boldsymbol{\theta}} V_{\boldsymbol{\pi_\theta}}^{\boldsymbol{\pi_\theta}}(\tau)$, we get,

$$\nabla_{\boldsymbol{\theta}} V_{\boldsymbol{\pi_\theta}}^{\boldsymbol{\pi_\theta}}(\tau) = \nabla_{\boldsymbol{\theta}} \mathop{\mathbb{E}}_{\tau \sim \mathbb{P}_{\boldsymbol{\pi_\theta}}^{\boldsymbol{\pi_\theta}}}[f_{\boldsymbol{\theta}}(\tau)]$$

$$
= \underset{\tau \sim \mathbb{P}^{\boldsymbol{\pi}_{\boldsymbol{\theta}}}_{\boldsymbol{\pi}_{\boldsymbol{\theta}}}}{\mathbb{E}} \left[ \left( \sum_{t=0}^{\infty} \nabla_{\boldsymbol{\theta}} \log \boldsymbol{\pi}_{\boldsymbol{\theta}}(a_t \mid s_t) \right) \cdot \left( \sum_{t=0}^{\infty} \gamma^t r_{\boldsymbol{\pi}_{\boldsymbol{\theta}}}(s_t, a_t) \right) \right]
$$

$$
+ \underset{\tau \sim \mathbb{P}^{\boldsymbol{\pi}_{\boldsymbol{\theta}}}_{\boldsymbol{\pi}_{\boldsymbol{\theta}}}}{\mathbb{E}} \left[ \left( \sum_{t=0}^{\infty} \nabla_{\boldsymbol{\theta}} \log \mathbf{P}_{\boldsymbol{\pi}_{\boldsymbol{\theta}}}(s_{t+1}|s_t, a_t) \right) \cdot \left( \sum_{t=0}^{\infty} \gamma^t r_{\boldsymbol{\pi}_{\boldsymbol{\theta}}}(s_t, a_t) \right) \right]
$$

$$
+ \underset{\tau \sim \mathbb{P}^{\boldsymbol{\pi}_{\boldsymbol{\theta}}}_{\boldsymbol{\pi}_{\boldsymbol{\theta}}}}{\mathbb{E}} \left[ \sum_{t=0}^{\infty} \gamma^t \nabla_{\boldsymbol{\theta}} r_{\boldsymbol{\pi}_{\boldsymbol{\theta}}}(s_t, a_t) \right]
$$

$$
= \underset{\tau \sim \mathbb{P}^{\boldsymbol{\pi}_{\boldsymbol{\theta}}}_{\boldsymbol{\pi}_{\boldsymbol{\theta}}}}{\mathbb{E}} \left[ \sum_{t=0}^{\infty} \gamma^t A^{\boldsymbol{\pi}_{\boldsymbol{\theta}}}_{\boldsymbol{\pi}_{\boldsymbol{\theta}}}(s_t, a_t) \nabla_{\boldsymbol{\theta}} \log \boldsymbol{\pi}_{\boldsymbol{\theta}}(a_t \mid s_t) \right] + \underset{\tau \sim \mathbb{P}^{\boldsymbol{\pi}_{\boldsymbol{\theta}}}_{\boldsymbol{\pi}_{\boldsymbol{\theta}}}}{\mathbb{E}} \left[ \sum_{t=0}^{\infty} \gamma^t A^{\boldsymbol{\pi}_{\boldsymbol{\theta}}}_{\boldsymbol{\pi}_{\boldsymbol{\theta}}}(s_t, a_t) \nabla_{\boldsymbol{\theta}} \log \mathbf{P}_{\boldsymbol{\pi}_{\boldsymbol{\theta}}}(s_{t+1}|s_t, a_t) \right]
$$

$$
+ \underset{\tau \sim \mathbb{P}^{\boldsymbol{\pi}_{\boldsymbol{\theta}}}_{\boldsymbol{\pi}_{\boldsymbol{\theta}}}}{\mathbb{E}} \left[ \sum_{t=0}^{\infty} \gamma^t \nabla_{\boldsymbol{\theta}} r_{\boldsymbol{\pi}_{\boldsymbol{\theta}}}(s_t, a_t) \right].
$$

The last equality is due to the definition of advantage function

$$
A^{\boldsymbol{\pi}_{\boldsymbol{\theta}}}_{\boldsymbol{\pi}_{\boldsymbol{\theta}}}(s_t, a_t) \triangleq \sum_{i=t+1}^{\infty} \gamma^{t-i} r_{\boldsymbol{\pi}_{\boldsymbol{\theta}}}(s_i, \boldsymbol{\pi}_{\boldsymbol{\theta}}(s_i)) - \underset{\substack{s_{t'+1} \sim \mathbf{P}_{\boldsymbol{\pi}_{\boldsymbol{\theta}}}(\cdot|s_{t'}, a_{t'}) \\ \forall t' \in [t, \infty)}}{\mathbb{E}} \left[ \sum_{i=t+1}^{\infty} \gamma^{t-i} r_{\boldsymbol{\pi}_{\boldsymbol{\theta}}}(s_i, \boldsymbol{\pi}_{\boldsymbol{\theta}}(s_i)) | (s_t, a_t) \right]
$$

$$
\triangleq Q^{\boldsymbol{\pi}_{\boldsymbol{\theta}}}_{\boldsymbol{\pi}_{\boldsymbol{\theta}}}(s_t, a_t) - V^{\boldsymbol{\pi}_{\boldsymbol{\theta}}}_{\boldsymbol{\pi}_{\boldsymbol{\theta}}}(s_t)
$$

as in classical policy gradient theorem. Hence, we conclude the proof for part (a) of the theorem.

*Proof of part (b).* Now, we derive explicit gradient form for entropy-regularised value function.

Let us define the soft reward as $\tilde{r}_{\boldsymbol{\pi}_{\boldsymbol{\theta}}}(s_t, a_t) \triangleq r_{\boldsymbol{\pi}_{\boldsymbol{\theta}}}(s_t, a_t) - \lambda \log \boldsymbol{\pi}_{\boldsymbol{\theta}}(a_t|s_t)$. Again, we start by defining regularised objective function

$$
\tilde{f}_{\boldsymbol{\theta}}(\tau) = \sum_{t=0}^{\infty} \gamma^t \tilde{r}_{\boldsymbol{\pi}_{\boldsymbol{\theta}}}(s_t, a_t)
$$

Following the same steps as that of *Part (a)*, we get

$$
\nabla_{\boldsymbol{\theta}} \tilde{V}^{\boldsymbol{\pi}_{\boldsymbol{\theta}}}_{\boldsymbol{\pi}_{\boldsymbol{\theta}}}(\tau) = \nabla_{\boldsymbol{\theta}} \underset{\tau \sim \mathbb{P}^{\boldsymbol{\pi}_{\boldsymbol{\theta}}}_{\boldsymbol{\pi}_{\boldsymbol{\theta}}}}{\mathbb{E}} [\tilde{f}_{\boldsymbol{\theta}}(\tau)]
$$

$$
= \underset{\tau \sim \mathbb{P}^{\boldsymbol{\pi}_{\boldsymbol{\theta}}}_{\boldsymbol{\pi}_{\boldsymbol{\theta}}}}{\mathbb{E}} \left[ \sum_{t=0}^{\infty} \gamma^t \tilde{A}^{\boldsymbol{\pi}_{\boldsymbol{\theta}}}_{\boldsymbol{\pi}_{\boldsymbol{\theta}}}(s_t, a_t) \nabla_{\boldsymbol{\theta}} \log \boldsymbol{\pi}_{\boldsymbol{\theta}}(a_t \mid s_t) \right] + \underset{\tau \sim \mathbb{P}^{\boldsymbol{\pi}_{\boldsymbol{\theta}}}_{\boldsymbol{\pi}_{\boldsymbol{\theta}}}}{\mathbb{E}} \left[ \sum_{t=0}^{\infty} \gamma^t \tilde{A}^{\boldsymbol{\pi}_{\boldsymbol{\theta}}}_{\boldsymbol{\pi}_{\boldsymbol{\theta}}}(s_t, a_t) \nabla_{\boldsymbol{\theta}} \log \mathbf{P}_{\boldsymbol{\pi}_{\boldsymbol{\theta}}}(s_{t+1}|s_t, a_t) \right]
$$

$$
+ \underset{\tau \sim \mathbb{P}^{\boldsymbol{\pi}_{\boldsymbol{\theta}}}_{\boldsymbol{\pi}_{\boldsymbol{\theta}}}}{\mathbb{E}} \left[ \sum_{t=0}^{\infty} \gamma^t \nabla_{\boldsymbol{\theta}} \tilde{r}_{\boldsymbol{\pi}_{\boldsymbol{\theta}}}(s_t, a_t) \right].
$$

$$
= \underset{\tau \sim \mathbb{P}^{\boldsymbol{\pi}_{\boldsymbol{\theta}}}_{\boldsymbol{\pi}_{\boldsymbol{\theta}}}}{\mathbb{E}} \left[ \sum_{t=0}^{\infty} \gamma^t \tilde{A}^{\boldsymbol{\pi}_{\boldsymbol{\theta}}}_{\boldsymbol{\pi}_{\boldsymbol{\theta}}}(s_t, a_t) \nabla_{\boldsymbol{\theta}} \log \boldsymbol{\pi}_{\boldsymbol{\theta}}(a_t \mid s_t) \right] + \underset{\tau \sim \mathbb{P}^{\boldsymbol{\pi}_{\boldsymbol{\theta}}}_{\boldsymbol{\pi}_{\boldsymbol{\theta}}}}{\mathbb{E}} \left[ \sum_{t=0}^{\infty} \gamma^t \tilde{A}^{\boldsymbol{\pi}_{\boldsymbol{\theta}}}_{\boldsymbol{\pi}_{\boldsymbol{\theta}}}(s_t, a_t) \nabla_{\boldsymbol{\theta}} \log \mathbf{P}_{\boldsymbol{\pi}_{\boldsymbol{\theta}}}(s_{t+1}|s_t, a_t) \right]
$$

$$
+ \underset{\tau \sim \mathbb{P}^{\boldsymbol{\pi}_{\boldsymbol{\theta}}}_{\boldsymbol{\pi}_{\boldsymbol{\theta}}}}{\mathbb{E}} \left[ \sum_{t=0}^{\infty} \gamma^t \nabla_{\boldsymbol{\theta}} r_{\boldsymbol{\pi}_{\boldsymbol{\theta}}}(s_t, a_t) \right] - \lambda \underset{\tau \sim \mathbb{P}^{\boldsymbol{\pi}_{\boldsymbol{\theta}}}_{\boldsymbol{\pi}_{\boldsymbol{\theta}}}}{\mathbb{E}} \left[ \sum_{t=0}^{\infty} \gamma^t \nabla_{\boldsymbol{\theta}} \log \boldsymbol{\pi}_{\boldsymbol{\theta}}(a_t|s_t) \right]
$$

Here,

$$
\tilde{A}^{\boldsymbol{\pi}_{\boldsymbol{\theta}}}_{\boldsymbol{\pi}_{\boldsymbol{\theta}}}(s_t, a_t) \triangleq \sum_{i=t+1}^{\infty} \gamma^{t-i} \tilde{r}_{\boldsymbol{\pi}_{\boldsymbol{\theta}}}(s_i, \boldsymbol{\pi}_{\boldsymbol{\theta}}(s_i)) - \underset{\substack{s_{t'+1} \sim \mathbf{P}^{\boldsymbol{\pi}_{\boldsymbol{\theta}}}_{\boldsymbol{\pi}_{\boldsymbol{\theta}}}(\cdot|s_{t'}, a_{t'}) \\ \forall t' \in [t, \infty)}}{\mathbb{E}} \left[ \sum_{i=t}^{\infty} \gamma^{t-i} \tilde{r}_{\boldsymbol{\pi}_{\boldsymbol{\theta}}}(s_i, \boldsymbol{\pi}_{\boldsymbol{\theta}}(s_i)) | (s_t, a_t) \right]
$$

$$
\triangleq \tilde{Q}^{\boldsymbol{\pi}_{\boldsymbol{\theta}}}_{\boldsymbol{\pi}_{\boldsymbol{\theta}}}(s_t, a_t) - \tilde{V}^{\boldsymbol{\pi}_{\boldsymbol{\theta}}}_{\boldsymbol{\pi}_{\boldsymbol{\theta}}}(s_t)
$$

denotes the advantage function with soft rewards, or in brief, the soft advantage function. Hence, we conclude proof of part (b). $\qquad\square$

## G. Convergence results of Smooth PeMDPs

**Definition 6.** *The discounted state occupancy measure $d_{\pi'}^{\pi}(s|s_0)$ induced by a policy $\pi$ and an MDP environment defined by $\pi'$ is defined as*

$$d_{\pi'}^{\pi}(s|\rho) \triangleq \sum_{a \in \mathcal{A}} d_{\pi'}^{\pi}(s, a|\rho) = (1-\gamma) \sum_{a \in \mathcal{A}} \mathop{\mathbb{E}}_{\tau \sim \mathbb{P}_{\pi'}^{\pi}} \Big[ \sum_{t=0}^{\infty} \gamma^t \mathbb{1}\{s_t = s, a_t = a\} \Big].$$

**Lemma 3.** *We define* $\mathsf{Cov} \triangleq \max_{\theta, \nu} \Big\| \frac{d_{\pi_\theta, \rho}^{\pi_o^{\star}}}{d_{\pi_\theta, \nu}^{\pi_\theta}} \Big\|_{\infty}$, *then (a) for unregularised value function,* $\mathrm{SubOpt}(\pi_\theta) \leq \sqrt{|\mathcal{S}||\mathcal{A}|}\mathsf{Cov}\|\nabla_\theta V_{\pi_\theta}^{\pi_\theta}(\nu)\|_2 + \frac{1+\mathsf{Cov}}{(1-\gamma)^2}\Big(L_r + L_\mathbf{P} R_{\max}\Big)$, *(b) for cross-entropy regularised value function* $\mathrm{SubOpt}(\pi_\theta \mid \lambda) \leq \sqrt{|\mathcal{S}||\mathcal{A}|}\mathsf{Cov}\|\nabla_\theta \tilde{V}_{\pi_\theta}^{\pi_\theta}(\nu)\|_2 + \frac{2+\mathsf{Cov}}{(1-\gamma)^2}\Big(L_r + L_\mathbf{P}(R_{\max} + \lambda \log|\mathcal{A}|)\Big)$

*Proof of part - (a).* This proof is divided into two parts. In the first part we bound the expected advantage term from Lemma 2 with the norm of the gradient of value function. During this step, we need to express the expected advantage as a linear combination of the advantage itself and the occupancy measure over all states and actions. The expectation however is taken w.r.t the occupancy measure $d_{\pi_\theta, \rho}^{\pi_o^{\star}}$, thus we need to perform a change of measure which introduces a coverage term as shown below. In the second step we directly use the bound of rewards and transitions obtained from their Lipchitzness in lemma 2. We know by Lemma 1 that

$$\mathrm{SubOpt}(\pi_\theta) = \frac{1}{1-\gamma} \mathop{\mathbb{E}}_{(s,a) \sim d_{\pi_\theta, \rho}^{\pi_o^{\star}}} [A_{\pi_\theta}^{\pi_\theta}(s, a)] + \frac{1}{1-\gamma} \mathop{\mathbb{E}}_{(s,a) \sim d_{\pi_\theta, \rho}^{\pi_o^{\star}}} \Big[ (r_{\pi_o^{\star}}(s, a) - r_{\pi_\theta}(s, a))$$
$$+ \gamma(\mathbf{P}_{\pi_o^{\star}}(\cdot|s, a) - \mathbf{P}_{\pi_\theta}(\cdot|s, a))^{\top} V_{\pi_o^{\star}}^{\pi_o^{\star}}(\cdot) \Big].$$

**Step 1: Upper bounding Term 1.** Let us define $\mathrm{Term1} \triangleq \frac{1}{1-\gamma} \mathbb{E}_{(s,a) \sim d_{\pi_\theta, \rho}^{\pi_o^{\star}}(\cdot|s_0)}[A_{\pi_\theta}^{\pi_\theta}(s, a)]$

$$\mathop{\mathbb{E}}_{(s,a) \sim d_{\pi_\theta, \rho}^{\pi_o^{\star}}} [A_{\pi_\theta}^{\pi_\theta}(s, a)] = \sum_{s,a} d_{\pi_\theta, \rho}^{\pi_o^{\star}}(s, a|\rho) A_{\pi_\theta}^{\pi_\theta}(s, a) = \sum_{s,a} \frac{d_{\pi_\theta, \rho}^{\pi_o^{\star}}(s, a|\rho)}{d_{\pi_\theta}^{\pi_\theta}(s, a|\nu)} d_{\pi_\theta}^{\pi_\theta}(s, a|\nu) A_{\pi_\theta}^{\pi_\theta}(s, a)$$
$$\leq \Big\| \frac{d_{\pi_\theta, \rho}^{\pi_o^{\star}}}{d_{\pi_\theta, \nu}^{\pi_\theta}} \Big\|_{\infty} \sum_{s,a} d_{\pi_\theta}^{\pi_\theta}(s, a|\nu) A_{\pi_\theta}^{\pi_\theta}(s, a)$$
$$\leq \mathsf{Cov} \sum_{s,a} d_{\pi_\theta}^{\pi_\theta}(s, a|\nu) A_{\pi_\theta}^{\pi_\theta}(s, a) \qquad (23)$$

**Step 2: Generic Gradient Calculation** Now, we leverage the gradient of generic performative MDPs,

$$\frac{\partial}{\partial \theta_{s,a}} V_{\pi_\theta}^{\pi_\theta}(\nu) = \mathop{\mathbb{E}}_{\tau \sim \mathbb{P}_{\pi_\theta, \nu}^{\pi_\theta}} \Big[ \sum_{t=0}^{\infty} \gamma^t \Big( A_{\pi_\theta}^{\pi_\theta}(s_t, a_t) \frac{\partial}{\partial \theta_{s,a}} \log \pi_\theta(a_t \mid s_t) + A_{\pi_\theta}^{\pi_\theta}(s_t, a_t) \frac{\partial}{\partial \theta_{s,a}} \log P_{\pi_\theta}(s_{t+1}|s_t, a_t)$$
$$+ \frac{\partial}{\partial \theta_{s,a}} r_{\pi_\theta}(s_t, a_t) \Big) \Big]$$
$$\mathop{\geq}_{(a)} \mathop{\mathbb{E}}_{\tau \sim \mathbb{P}_{\pi_\theta, \nu}^{\pi_\theta}} \Big[ \sum_{t=0}^{\infty} \gamma^t A_{\pi_\theta}^{\pi_\theta}(s_t, a_t) \mathbb{1}[s_t = s, a_t = a] \Big] - \mathop{\mathbb{E}}_{\tau \sim \mathbb{P}_{\pi_\theta, \nu}^{\pi_\theta}} \Big[ \sum_{t=0}^{\infty} \gamma^t \pi_\theta(a|s) \mathbb{1}[s_t = s] A_{\pi_\theta}^{\pi_\theta}(s_t, a_t) \Big]$$
$$- \mathop{\mathbb{E}}_{\tau \sim \mathbb{P}_{\pi_\theta, \nu}^{\pi_\theta}} \Big[ \sum_{t=0}^{\infty} \gamma^t L_r \mathbb{1}[s_t = s, a_t = a] \Big] - \mathop{\mathbb{E}}_{\tau \sim \mathbb{P}_{\pi_\theta, \nu}^{\pi_\theta}} \Big[ \sum_{t=0}^{\infty} \gamma^t A_{\pi_\theta}^{\pi_\theta}(s_t, a_t) \mathbb{1}[s_t = s, a_t = a] L_\mathbf{P} \Big]$$
$$\mathop{\geq}_{(b)} \frac{1}{1-\gamma} d_{\pi_\theta, \nu}^{\pi_\theta}(s, a) A_{\pi_\theta}^{\pi_\theta}(s, a) - \frac{1}{(1-\gamma)^2} d_{\pi_\theta, \nu}^{\pi_\theta}(s, a) R_{\max} L_\mathbf{P} - \frac{1}{1-\gamma} L_r d_{\pi_\theta, \nu}^{\pi_\theta}(s, a)$$

(a) holds due to Lipchitzness of rewards and transitions and (b) holds since $\mathbb{E}_{\tau\sim\mathbb{P}^{\pi_\theta}_{\pi_\theta}}\left[\sum_{t=0}^{\infty}\gamma^t\pi_\theta(a|s)\mathbb{1}[s_t = s]A^{\pi_\theta}_{\pi_\theta}(s_t, a_t)\right] = 0$. Hence,

$$\frac{1}{1-\gamma}\sum_{s,a}d^{\pi_\theta}_{\pi_\theta}(s,a|\nu)A^{\pi_\theta}_{\pi_\theta}(s,a) \leq \sum_{s,a}\frac{\partial V^{\pi_\theta}_{\pi_\theta}(\nu)}{\partial\theta_{s,a}} + \frac{L_r}{1-\gamma} + \frac{R_{\max}L_{\mathbf{P}}}{(1-\gamma)^2}$$

$$\leq \sqrt{|\mathcal{S}||\mathcal{A}|}\|\nabla_\theta V^{\pi_\theta}_{\pi_\theta}(\nu)\|_2 + \frac{L_r}{1-\gamma} + \frac{R_{\max}L_{\mathbf{P}}}{(1-\gamma)^2}$$

Now, substituting the above result back in Equation (23), we get

$$\frac{1}{1-\gamma}\mathop{\mathbb{E}}_{(s,a)\sim d^{\pi^\star_o}_{\pi_\theta,\rho}}[A^{\pi_\theta}_{\pi_\theta}(s,a)] \leq \sqrt{|\mathcal{S}||\mathcal{A}|}\mathsf{Cov}\|\nabla_\theta V^{\pi_\theta}_{\pi_\theta}(\nu)\|_2 + \left(\frac{L_r}{1-\gamma} + \frac{R_{\max}L_{\mathbf{P}}}{(1-\gamma)^2}\right)\mathsf{Cov} \tag{24}$$

**Step 3: Upper bounding Term 2.** For lipchitz rewards and transitions, we further obtain from Lemma 2,

$$\text{Term 2} \triangleq \frac{1}{1-\gamma}\mathop{\mathbb{E}}_{(s,a)\sim d^{\pi^\star_o}_{\pi_\theta,\rho}}\left[(r_{\pi^\star_o}(s,a) - r_{\pi_\theta}(s,a)) + \gamma(\mathbf{P}_{\pi^\star_o}(\cdot|s,a) - \mathbf{P}_{\pi_\theta}(\cdot|s,a))^\top V^{\pi^\star_o}_{\pi^\star_o}(\cdot)\right]$$

$$\leq \frac{1}{1-\gamma}\left(L_r + \frac{\gamma}{1-\gamma}L_{\mathbf{P}}R_{\max}\right) \tag{25}$$

**Step 4:** Now, if we use Equation (24) and (25) together (and knowing $\gamma \leq 1$), we get

$$\mathrm{SubOpt}(\pi_\theta) \leq \sqrt{|\mathcal{S}||\mathcal{A}|}\mathsf{Cov}\|\nabla_\theta V^{\pi_\theta}_{\pi_\theta}(\nu)\|_2 + \frac{1+\mathsf{Cov}}{(1-\gamma)^2}\left(L_r + L_{\mathbf{P}}R_{\max}\right)$$

$\square$

*Proof of part - (b).* We start by deriving a lower bound on the derivative of $\tilde{V}^{\pi_\theta}_{\pi_\theta}(\nu)$

$$\frac{\partial}{\partial\theta_{s,a}}\tilde{V}^{\pi_\theta}_{\pi_\theta}(\nu)$$

$$= \mathop{\mathbb{E}}_{\tau\sim\mathbb{P}^{\pi_\theta}_{\pi_\theta,\nu}}\left[\sum_{t=0}^{\infty}\gamma^t\left(\tilde{A}^{\pi_\theta}_{\pi_\theta}(s_t,a_t)\frac{\partial}{\partial\theta_{s,a}}\log\pi_\theta(a_t \mid s_t) + \tilde{A}^{\pi_\theta}_{\pi_\theta}(s_t,a_t)\frac{\partial}{\partial\theta_{s,a}}\log P_{\pi_\theta}(s_{t+1}|s_t,a_t)\right.\right.$$

$$\left.\left.+ \frac{\partial}{\partial\theta_{s,a}}\tilde{r}_{\pi_\theta}(s_t,a_t)\right)\right]$$

$$\underset{(a)}{\geq} \mathop{\mathbb{E}}_{\tau\sim\mathbb{P}^{\pi_\theta}_{\pi_\theta,\nu}}\left[\sum_{t=0}^{\infty}\gamma^t\tilde{A}^{\pi_\theta}_{\pi_\theta}(s_t,a_t)\mathbb{1}[s_t = s, a_t = a]\right] - \mathop{\mathbb{E}}_{\tau\sim\mathbb{P}^{\pi_\theta}_{\pi_\theta,\nu}}\left[\sum_{t=0}^{\infty}\gamma^t\pi_\theta(a|s)\mathbb{1}[s_t = s]\tilde{A}^{\pi_\theta}_{\pi_\theta}(s_t,a_t)\right]$$

$$- \mathop{\mathbb{E}}_{\tau\sim\mathbb{P}^{\pi_\theta}_{\pi_\theta,\nu}}\left[\sum_{t=0}^{\infty}\gamma^t L_r\mathbb{1}[s_t = s, a_t = a]\right] - \mathop{\mathbb{E}}_{\tau\sim\mathbb{P}^{\pi_\theta}_{\pi_\theta,\nu}}\left[\sum_{t=0}^{\infty}\gamma^t\tilde{A}^{\pi_\theta}_{\pi_\theta}(s_t,a_t)\mathbb{1}[s_t = s, a_t = a]L_{\mathbf{P}}\right]$$

$$\underset{(b)}{=} \mathop{\mathbb{E}}_{\tau\sim\mathbb{P}^{\pi_\theta}_{\pi_\theta,\nu}}\left[\sum_{t=0}^{\infty}\gamma^t\tilde{A}^{\pi_\theta}_{\pi_\theta}(s_t,a_t)\mathbb{1}[s_t = s, a_t = a]\right] + \pi_\theta(a|s)\mathop{\mathbb{E}}_{\tau\sim\mathbb{P}^{\pi_\theta}_{\pi_\theta,\nu}}\left[\sum_{t=0}^{\infty}\gamma^t\log\pi_\theta(a_t|s_t)\mathbb{1}[s_t = s]\right]$$

$$- \mathop{\mathbb{E}}_{\tau\sim\mathbb{P}^{\pi_\theta}_{\pi_\theta,\nu}}\left[\sum_{t=0}^{\infty}\gamma^t L_r\mathbb{1}[s_t = s, a_t = a]\right] - \mathop{\mathbb{E}}_{\tau\sim\mathbb{P}^{\pi_\theta}_{\pi_\theta,\nu}}\left[\sum_{t=0}^{\infty}\gamma^t\tilde{A}^{\pi_\theta}_{\pi_\theta}(s_t,a_t)\mathbb{1}[s_t = s, a_t = a]L_{\mathbf{P}}\right]$$

$$= \mathop{\mathbb{E}}_{\tau\sim\mathbb{P}^{\pi_\theta}_{\pi_\theta,\nu}}\left[\sum_{t=0}^{\infty}\gamma^t\tilde{A}^{\pi_\theta}_{\pi_\theta}(s_t,a_t)\mathbb{1}[s_t = s, a_t = a]\right] + \pi_\theta(a|s)\mathop{\mathbb{E}}_{\tau\sim\mathbb{P}^{\pi_\theta}_{\pi_\theta,\nu}}\left[\sum_a\log\pi_\theta(a|s)\sum_{t=0}^{\infty}\gamma^t\mathbb{1}[s_t = s, a_t = a]\right]$$

$$- \mathop{\mathbb{E}}_{\tau\sim\mathbb{P}^{\pi_\theta}_{\pi_\theta,\nu}}\left[\sum_{t=0}^{\infty}\gamma^t L_r\mathbb{1}[s_t = s, a_t = a]\right] - \mathop{\mathbb{E}}_{\tau\sim\mathbb{P}^{\pi_\theta}_{\pi_\theta,\nu}}\left[\sum_{t=0}^{\infty}\gamma^t\tilde{A}^{\pi_\theta}_{\pi_\theta}(s_t,a_t)\mathbb{1}[s_t = s, a_t = a]L_{\mathbf{P}}\right]$$

$$= \frac{1}{1-\gamma} d_{\boldsymbol{\pi_\theta},\boldsymbol{\nu}}^{\boldsymbol{\pi_\theta}}(s,a) \tilde{A}_{\boldsymbol{\pi_\theta}}^{\boldsymbol{\pi_\theta}}(s,a) - \frac{1}{(1-\gamma)^2} d_{\boldsymbol{\pi_\theta},\boldsymbol{\nu}}^{\boldsymbol{\pi_\theta}}(s,a)(R_{\max} + \lambda \log|\mathcal{A}|)L_{\mathbf{P}} - \frac{1}{1-\gamma} L_r d_{\boldsymbol{\pi_\theta},\boldsymbol{\nu}}^{\boldsymbol{\pi_\theta}}(s,a)$$

$$+ \frac{\lambda}{1-\gamma} \boldsymbol{\pi_\theta}(a|s) \sum_a d_{\boldsymbol{\pi_\theta},\boldsymbol{\nu}}^{\boldsymbol{\pi_\theta}}(s,a) \log \boldsymbol{\pi_\theta}(a|s)$$

$$\underset{(c)}{\geq} \frac{1}{1-\gamma} d_{\boldsymbol{\pi_\theta},\boldsymbol{\nu}}^{\boldsymbol{\pi_\theta}}(s,a) \tilde{A}_{\boldsymbol{\pi_\theta}}^{\boldsymbol{\pi_\theta}}(s,a) - \frac{1}{(1-\gamma)^2} d_{\boldsymbol{\pi_\theta},\boldsymbol{\nu}}^{\boldsymbol{\pi_\theta}}(s,a)(R_{\max} + \lambda \log|\mathcal{A}|)L_{\mathbf{P}} - \frac{1}{1-\gamma} L_r d_{\boldsymbol{\pi_\theta},\boldsymbol{\nu}}^{\boldsymbol{\pi_\theta}}(s,a)$$

$$- \frac{\lambda}{1-\gamma} d_{\boldsymbol{\pi_\theta},\boldsymbol{\nu}}^{\boldsymbol{\pi_\theta}}(s,a) \log|\mathcal{A}|$$

(a) holds due to Lipchitzness of rewards, transitions and also for the following:

$$\underset{\tau \sim \mathbb{P}_{\boldsymbol{\pi_\theta}}^{\boldsymbol{\pi_\theta}}}{\mathbb{E}} \left[ \sum_{t=0}^{\infty} \gamma^t \frac{\partial}{\partial \theta_{s,a}} \log \boldsymbol{\pi_\theta}(a_t|s_t) \right] = \underset{\tau \sim \mathbb{P}_{\boldsymbol{\pi_\theta}}^{\boldsymbol{\pi_\theta}}}{\mathbb{E}} \left[ \sum_{t=0}^{\infty} \gamma^t \mathbb{1}[s_t = s, a_t = a] \right] - \underset{\tau \sim \mathbb{P}_{\boldsymbol{\pi_\theta}}^{\boldsymbol{\pi_\theta}}}{\mathbb{E}} \left[ \sum_{t=0}^{\infty} \gamma^t \boldsymbol{\pi_\theta}(a|s) \mathbb{1}[s_t = s] \right]$$

$$= d_{\boldsymbol{\pi_\theta},\boldsymbol{\rho}}^{\boldsymbol{\pi_\theta}}(s,a) - d_{\boldsymbol{\pi_\theta},\boldsymbol{\rho}}^{\boldsymbol{\pi_\theta}}(s) \boldsymbol{\pi_\theta}(a|s) = 0$$

(b) holds because:

$$\underset{\tau \sim \mathbb{P}_{\boldsymbol{\pi_\theta},\boldsymbol{\nu}}^{\boldsymbol{\pi_\theta}}}{\mathbb{E}} \left[ \sum_{t=0}^{\infty} \gamma^t \boldsymbol{\pi_\theta}(a|s) \mathbb{1}[s_t = s] \tilde{A}_{\boldsymbol{\pi_\theta}}^{\boldsymbol{\pi_\theta}}(s_t, a_t) \right] = \underset{\tau \sim \mathbb{P}_{\boldsymbol{\pi_\theta},\boldsymbol{\nu}}^{\boldsymbol{\pi_\theta}}}{\mathbb{E}} \left[ \sum_{t=0}^{\infty} \gamma^t \boldsymbol{\pi_\theta}(a|s) \mathbb{1}[s_t = s] A_{\boldsymbol{\pi_\theta}}^{\boldsymbol{\pi_\theta}}(s_t, a_t) \right]$$

$$- \underset{\tau \sim \mathbb{P}_{\boldsymbol{\pi_\theta},\boldsymbol{\nu}}^{\boldsymbol{\pi_\theta}}}{\mathbb{E}} \left[ \sum_{t=0}^{\infty} \gamma^t \boldsymbol{\pi_\theta}(a|s) \log \boldsymbol{\pi_\theta}(a_t|s_t) \mathbb{1}[s_t = s] \right]$$

$$= -\boldsymbol{\pi_\theta}(a|s) \left[ \sum_{t=0}^{\infty} \gamma^t \log \boldsymbol{\pi_\theta}(a_t|s_t) \mathbb{1}[s_t = s] \right]$$

And (c) holds since,

$$-\sum_a d_{\boldsymbol{\pi_\theta}}^{\boldsymbol{\pi_\theta}}(s,a|\boldsymbol{\nu}) \log \boldsymbol{\pi_\theta}(a|s) = d_{\boldsymbol{\pi_\theta}}^{\boldsymbol{\pi_\theta}}(s|\boldsymbol{\nu}) \left( -\sum_a \boldsymbol{\pi_\theta}(a|s) \log \boldsymbol{\pi_\theta}(a|s) \right)$$

$$\underset{(d)}{\leq} d_{\boldsymbol{\pi_\theta}}^{\boldsymbol{\pi_\theta}}(s|\boldsymbol{\nu}) \log|\mathcal{A}|$$

while (d) holds as entropy is upper bounded by $\log|\mathcal{A}|$ (Cover & Thomas, 2006, Theorem 2.6.4).

Hence,

$$\frac{1}{1-\gamma} \sum_{s,a} d_{\boldsymbol{\pi_\theta}}^{\boldsymbol{\pi_\theta}}(s,a|\boldsymbol{\nu}) A_{\boldsymbol{\pi_\theta}}^{\boldsymbol{\pi_\theta}}(s,a) \leq \sum_{s,a} \frac{\partial V_{\boldsymbol{\pi_\theta}}^{\boldsymbol{\pi_\theta}}(\boldsymbol{\nu})}{\partial \theta_{s,a}} + \frac{L_r}{1-\gamma} + \frac{(R_{\max} + \lambda \log|\mathcal{A}|)L_{\mathbf{P}}}{(1-\gamma)^2} + \frac{\lambda}{1-\gamma} \log|\mathcal{A}|$$

$$\leq \sqrt{|\mathcal{S}||\mathcal{A}|} \|\nabla_{\boldsymbol{\theta}} V_{\boldsymbol{\pi_\theta}}^{\boldsymbol{\pi_\theta}}(\boldsymbol{\nu})\|_2 + \frac{L_r}{1-\gamma} + \frac{(R_{\max} + \lambda \log|\mathcal{A}|)L_{\mathbf{P}}}{(1-\gamma)^2}$$

$$+ \frac{\lambda}{1-\gamma} \log|\mathcal{A}| \tag{26}$$

The last inequality holds from Cauchy-Schwartz. Now, upper-bounding term 2 we get from lemma 7,

$$\mathrm{SubOpt}(\boldsymbol{\pi_\theta} \mid \lambda) \leq \frac{1}{1-\gamma} \underset{(s,a) \sim d_{\boldsymbol{\pi_\theta},\boldsymbol{\rho}}^{\boldsymbol{\pi_\theta^\star}}}{\mathbb{E}} [\tilde{A}_{\boldsymbol{\pi_\theta}}^{\boldsymbol{\pi_\theta}}(s,a)]$$

$$+ \frac{1}{1-\gamma} \left( L_r + L_{\mathbf{P}} \frac{\gamma(R_{\max} + \lambda \log|\mathcal{A}|)}{1-\gamma} \right)$$

$$- \frac{\lambda}{1-\gamma} \sum_s \boldsymbol{d}_{\boldsymbol{\pi_\theta},\boldsymbol{\rho}}^{\boldsymbol{\pi}_o^\star}(s) D_{\mathrm{KL}}\left(\boldsymbol{\pi}_o^\star(\cdot|s) \,\|\, \boldsymbol{\pi_\theta}(\cdot|s)\right) \tag{27}$$

And we know,

$$-D_{\mathrm{KL}}\left(\boldsymbol{\pi}_o^\star(\cdot|s) \,\|\, \boldsymbol{\pi_\theta}(\cdot|s)\right) \leq -\sum_{a \in \mathcal{A}} \boldsymbol{\pi}_o^\star(a|s) \log \boldsymbol{\pi}_o^\star(a|s) \leq \log |\mathcal{A}|$$

Hence, we get

$$-\sum_s \boldsymbol{d}_{\boldsymbol{\pi_\theta},\boldsymbol{\rho}}^{\boldsymbol{\pi}_o^\star}(s) D_{\mathrm{KL}}\left(\boldsymbol{\pi}_o^\star(\cdot|s) \,\|\, \boldsymbol{\pi_\theta}(\cdot|s)\right) \leq \log |\mathcal{A}| \tag{28}$$

Now, combining (26), (27), (28) and following the steps in the proof of Lemma 3, we obtain the final gradient domination lemma. □

**Lemma 7** (Entropy-Regularized Performative Policy Difference: Generic Upper Bound). *Under Assumption 1 and 2 (a), the sub-optimality gap of a policy $\boldsymbol{\pi_\theta}$ is*

$$\mathrm{SubOpt}(\boldsymbol{\pi_\theta} \mid \lambda) \leq \frac{1}{1-\gamma} \mathop{\mathbb{E}}_{(s,a)\sim \boldsymbol{d}_{\boldsymbol{\pi_\theta},\boldsymbol{\rho}}^{\boldsymbol{\pi}_o^\star}} [\tilde{A}_{\boldsymbol{\pi_\theta}}^{\boldsymbol{\pi_\theta}}(s,a)] + \frac{1}{1-\gamma}\left(L_r + L_{\mathbf{P}} \frac{\gamma(R_{\max} + \lambda \log |\mathcal{A}|)}{1-\gamma}\right)$$
$$- \frac{\lambda}{1-\gamma} \sum_s \boldsymbol{d}_{\boldsymbol{\pi_\theta},\boldsymbol{\rho}}^{\boldsymbol{\pi}_o^\star}(s) D_{\mathrm{KL}}\left(\boldsymbol{\pi}_o^\star(\cdot|s) \,\|\, \boldsymbol{\pi_\theta}(\cdot|s)\right) \tag{29}$$

*Proof.* This lemma follows the same sketch as Lemma 2 with an exception in the way the soft rewards are handled. The difference in the soft rewards equals the difference of the original rewards with a Lagrange dependent term. This term is the expected KL divergence over the state visitation distribution. Lemma 1 for regularized rewards reduces to,

$$\tilde{V}_{\boldsymbol{\pi}}^{\boldsymbol{\pi}}(\boldsymbol{\rho}) - \tilde{V}_{\boldsymbol{\pi}'}^{\boldsymbol{\pi}'}(\boldsymbol{\rho}) = \frac{1}{1-\gamma} \mathop{\mathbb{E}}_{(s,a)\sim \boldsymbol{d}_{\boldsymbol{\pi}',\boldsymbol{\rho}}^{\boldsymbol{\pi}}} [\tilde{A}_{\boldsymbol{\pi}'}^{\boldsymbol{\pi}'}(s,a)]$$
$$+ \frac{1}{1-\gamma} \mathop{\mathbb{E}}_{(s,a)\sim \boldsymbol{d}_{\boldsymbol{\pi}',\boldsymbol{\rho}}^{\boldsymbol{\pi}}} \left([\tilde{r}_{\boldsymbol{\pi}}(s,a) - \tilde{r}_{\boldsymbol{\pi}'}(s,a)] + \gamma(\mathbf{P}_{\boldsymbol{\pi}} - \mathbf{P}_{\boldsymbol{\pi}'})^\top \tilde{V}_{\boldsymbol{\pi}}^{\boldsymbol{\pi}}(s_0)\right). \tag{30}$$

Therefore,

$$\tilde{r}_{\boldsymbol{\pi}_o^\star}(s,a) - \tilde{r}_{\boldsymbol{\pi_\theta}}(s,a) = r_{\boldsymbol{\pi}_o^\star}(s,a) - r_{\boldsymbol{\pi_\theta}}(s,a) + \lambda\left(\log \boldsymbol{\pi_\theta}(a|s) - \log \boldsymbol{\pi}_o^\star(a|s)\right)$$

Therefore, we can write (30) in the following way,

$$\mathrm{SubOpt}(\boldsymbol{\pi_\theta} \mid \lambda) = \frac{1}{1-\gamma} \mathop{\mathbb{E}}_{(s,a)\sim \boldsymbol{d}_{\boldsymbol{\pi_\theta},\boldsymbol{\rho}}^{\boldsymbol{\pi}_o^\star}} [\tilde{A}_{\boldsymbol{\pi_\theta}}^{\boldsymbol{\pi_\theta}}(s,a)]$$
$$+ \frac{1}{1-\gamma} \mathop{\mathbb{E}}_{(s,a)\sim \boldsymbol{d}_{\boldsymbol{\pi_\theta},\boldsymbol{\rho}}^{\boldsymbol{\pi}_o^\star}} \left([\tilde{r}_{\boldsymbol{\pi}_o^\star}(s,a) - \tilde{r}_{\boldsymbol{\pi_\theta}}(s,a)] + \gamma(\mathbf{P}_{\boldsymbol{\pi}_o^\star} - \mathbf{P}_{\boldsymbol{\pi_\theta}})^\top \tilde{V}_{\boldsymbol{\pi}_o^\star}^{\boldsymbol{\pi}_o^\star}(s_0)\right).$$
$$+ \frac{\lambda}{1-\gamma} \sum_{s,a} [\log \boldsymbol{\pi_\theta}(a|s) - \log \boldsymbol{\pi}_o^\star(a|s)] \boldsymbol{d}_{\boldsymbol{\pi_\theta},\boldsymbol{\rho}}^{\boldsymbol{\pi}_o^\star}(s,a|s_0)$$
$$= \frac{1}{1-\gamma} \mathop{\mathbb{E}}_{(s,a)\sim \boldsymbol{d}_{\boldsymbol{\pi_\theta},\boldsymbol{\rho}}^{\boldsymbol{\pi}_o^\star}} [\tilde{A}_{\boldsymbol{\pi_\theta}}^{\boldsymbol{\pi_\theta}}(s,a)]$$
$$+ \frac{1}{1-\gamma} \mathop{\mathbb{E}}_{(s,a)\sim \boldsymbol{d}_{\boldsymbol{\pi_\theta},\boldsymbol{\rho}}^{\boldsymbol{\pi}_o^\star}} \left([\tilde{r}_{\boldsymbol{\pi}_o^\star}(s,a) - \tilde{r}_{\boldsymbol{\pi_\theta}}(s,a)] + \gamma(\mathbf{P}_{\boldsymbol{\pi}_o^\star} - \mathbf{P}_{\boldsymbol{\pi_\theta}})^\top \tilde{V}_{\boldsymbol{\pi}_o^\star}^{\boldsymbol{\pi}_o^\star}(s_0)\right)$$
$$+ \frac{\lambda}{1-\gamma} \sum_{s,a} \boldsymbol{d}_{\boldsymbol{\pi_\theta},\boldsymbol{\rho}}^{\boldsymbol{\pi}_o^\star}(s) \boldsymbol{\pi}_o^\star(a|s) [\log \boldsymbol{\pi_\theta}(a|s) - \log \boldsymbol{\pi}_o^\star(a|s)]$$

$$\underset{(a)}{=} \frac{1}{1-\gamma} \underset{(s,a)\sim d_{\pi_\theta,\rho}^{\pi_o^\star}}{\mathbb{E}} [\tilde{A}_{\pi_\theta}^{\pi_\theta}(s,a)]$$

$$+ \frac{1}{1-\gamma} \underset{(s,a)\sim d_{\pi_\theta,\rho}^{\pi_o^\star}}{\mathbb{E}} \left( [\tilde{r}_{\pi_o^\star}(s,a) - \tilde{r}_{\pi_\theta}(s,a)] + \gamma(\mathbf{P}_{\pi_o^\star} - \mathbf{P}_{\pi_\theta})^\top \tilde{V}_{\pi_o^\star}^{\pi_o^\star}(s_0) \right)$$

$$- \frac{\lambda}{1-\gamma} \sum_s d_{\pi_\theta,\rho}^{\pi_o^\star}(s) D_{\mathrm{KL}}\left( \pi_o^\star(\cdot|s) \,\|\, \pi_\theta(\cdot|s) \right)$$

$$\underset{\substack{\leq\\ \text{Holder's ineq.}}}{} \frac{1}{1-\gamma} \underset{(s,a)\sim d_{\pi_\theta,\rho}^{\pi_o^\star}}{\mathbb{E}} [\tilde{A}_{\pi_\theta}^{\pi_\theta}(s,a)]$$

$$+ \frac{1}{1-\gamma} \underset{(s,a)\sim d_{\pi_\theta,\rho}^{\pi_o^\star}}{\mathbb{E}} \left( [\tilde{r}_{\pi_o^\star}(s,a) - \tilde{r}_{\pi_\theta}(s,a)] + \gamma\|\mathbf{P}_{\pi_o^\star} - \mathbf{P}_{\pi_\theta}\|_1 \|\tilde{V}_{\pi_o^\star}^{\pi_o^\star}(s_0)\|_\infty \right)$$

$$- \frac{\lambda}{1-\gamma} \sum_s d_{\pi_\theta,\rho}^{\pi_o^\star}(s) D_{\mathrm{KL}}\left( \pi_o^\star(\cdot|s) \,\|\, \pi_\theta(\cdot|s) \right)$$

$$\underset{(b)}{\leq} \frac{1}{1-\gamma} \underset{(s,a)\sim d_{\pi_\theta,\rho}^{\pi_o^\star}}{\mathbb{E}} [\tilde{A}_{\pi_\theta}^{\pi_\theta}(s,a)]$$

$$+ \frac{1}{1-\gamma} \underset{(s,a)\sim d_{\pi_\theta,\rho}^{\pi_o^\star}}{\mathbb{E}} \left( [\tilde{r}_{\pi_o^\star}(s,a) - \tilde{r}_{\pi_\theta}(s,a)] + \gamma\|\mathbf{P}_{\pi_o^\star} - \mathbf{P}_{\pi_\theta}\|_1 \frac{R_{\max} + \lambda \log|\mathcal{A}|}{1-\gamma} \right)$$

$$- \frac{\lambda}{1-\gamma} \sum_s d_{\pi_\theta,\rho}^{\pi_o^\star}(s) D_{\mathrm{KL}}\left( \pi_o^\star(\cdot|s) \,\|\, \pi_\theta(\cdot|s) \right)$$

$$\underset{\substack{\leq\\ \text{Lipschitz } r \,\&\, \mathbf{P}}}{} \frac{1}{1-\gamma} \underset{(s,a)\sim d_{\pi_\theta,\rho}^{\pi_o^\star}}{\mathbb{E}} [\tilde{A}_{\pi_\theta}^{\pi_\theta}(s,a)] + \frac{1}{1-\gamma} \underset{(s,a)\sim d_{\pi_\theta,\rho}^{\pi_o^\star}}{\mathbb{E}} \left( L_r + L_{\mathbf{P}} \frac{\gamma(R_{\max} + \lambda \log|\mathcal{A}|)}{1-\gamma} \right) \|\pi_o^\star - \pi_\theta\|_\infty$$

$$- \frac{\lambda}{1-\gamma} \sum_s d_{\pi_\theta,\rho}^{\pi_o^\star}(s) D_{\mathrm{KL}}\left( \pi_o^\star(\cdot|s) \,\|\, \pi_\theta(\cdot|s) \right)$$

$$\underset{(c)}{\leq} \frac{1}{1-\gamma} \underset{(s,a)\sim d_{\pi_\theta,\rho}^{\pi_o^\star}}{\mathbb{E}} [\tilde{A}_{\pi_\theta}^{\pi_\theta}(s,a)] + \frac{1}{1-\gamma} \left( L_r + L_{\mathbf{P}} \frac{\gamma(R_{\max} + \lambda \log|\mathcal{A}|)}{1-\gamma} \right)$$

$$- \frac{\lambda}{1-\gamma} \sum_s d_{\pi_\theta,\rho}^{\pi_o^\star}(s) D_{\mathrm{KL}}\left( \pi_o^\star(\cdot|s) \,\|\, \pi_\theta(\cdot|s) \right)$$

The equality (a) holds since,

$$\underset{a\sim\pi_o^\star(\cdot|s)}{\mathbb{E}} [\log \pi_\theta(a|s) - \log \pi_o^\star(a|s)] = -D_{\mathrm{KL}}\left( \pi_o^\star(\cdot|s) \,\|\, \pi_\theta(\cdot|s) \right)$$

The inequality (b) holds due to the result of (Mei et al., 2020), i.e.

$$\|\tilde{V}_{\pi_o^\star}^{\pi_o^\star}\|_\infty \leq \frac{R_{\max} + \lambda \log|\mathcal{A}|}{1-\gamma} \tag{31}$$

Finally, (c) is due to the fact that $\|\pi_o^\star - \pi_\theta\|_\infty \leq 1$.

$\square$

Thus, as a consequence, we obtain convergence of PePG.

**Theorem 3** (Convergence of smooth PeMDPs). *We set learning rate* $\eta = \mathcal{O}\left( \frac{(1-\gamma)^2}{|\mathcal{A}|} \right)$. *Then, (a) For unregularised objective , we get* $\min_{t<T} \mathrm{SubOpt}(\pi_{\theta_t}) \leq \epsilon + \mathcal{O}\left( \frac{\mathsf{Cov}}{(1-\gamma)^2} \right)$ *when* $T = \Omega\left( \frac{|\mathcal{S}||\mathcal{A}|^2 R_{\max}^2 \mathsf{Cov}^2}{\epsilon^2(1-\gamma)^3} \right)$. *(b) For entropy-regularised objective with* $\lambda = \frac{(1-\gamma)R_{\max}}{1+2\log|\mathcal{A}|}$, *we get* $\min_{t<T} \mathrm{SubOpt}(\pi_{\theta_t}|\lambda) \leq \epsilon + \mathcal{O}\left( \frac{\mathsf{Cov}}{(1-\gamma)^2} \right)$ *when* $T = \Omega\left( \frac{|\mathcal{S}||\mathcal{A}|^2 R_{\max}^2 \mathsf{Cov}^2}{\epsilon^2(1-\gamma)^3} \right)$.

*Proof of Theorem 3– Part (a).* We proceed with this proof by dividing it in four steps. In the first step, we use the smoothness of the value function to prove an upper bound for the minimum squared gradient norm of the value over time which is a

constant times $1/T$. In the second step, we derive a lower bound on the norm of gradient of value function using Lemma 4. In the final two steps, we combine the bounds obtained from the first two steps to derive lower bounds for $T$ and $\epsilon$, i.e. the error threshold.

**Step 1:** As $V_{\boldsymbol{\pi_\theta}}^{\boldsymbol{\pi_\theta}}$ is $L$-smooth (Lemma 5), it satisfies

$$\left| V_{\boldsymbol{\pi_\theta}}^{\boldsymbol{\pi_\theta}}(\boldsymbol{\rho}) - V_{\boldsymbol{\pi_\theta'}}^{\boldsymbol{\pi_\theta'}}(\boldsymbol{\rho}) - \langle \nabla_{\boldsymbol{\theta}} V_{\boldsymbol{\pi_\theta}}^{\boldsymbol{\pi_\theta}}(\boldsymbol{\rho}), \boldsymbol{\theta} - \boldsymbol{\theta'} \rangle \right| \leq \frac{L}{2} \|\boldsymbol{\theta} - \boldsymbol{\theta'}\|^2$$

Thus, taking $\boldsymbol{\theta}$ as $\boldsymbol{\theta}_{t+1}$ and $\boldsymbol{\theta'}$ as $\boldsymbol{\theta}_t$ and using the gradient ascent expression (Equation (5)) yields

$$\left| V_{\boldsymbol{\pi_{\theta_{t+1}}}}^{\boldsymbol{\pi_{\theta_{t+1}}}}(\boldsymbol{\rho}) - V_{\boldsymbol{\pi_{\theta_t}}}^{\boldsymbol{\pi_{\theta_t}}}(\boldsymbol{\rho}) - \eta \|\nabla_{\boldsymbol{\theta}} V_{\boldsymbol{\pi_{\theta_t}}}^{\boldsymbol{\pi_{\theta_t}}}(\boldsymbol{\rho})\|^2 \right| \leq \frac{L}{2} \|\boldsymbol{\theta}_{t+1} - \boldsymbol{\theta}_t\|^2$$

$$\implies \qquad V_{\boldsymbol{\pi_{\theta_{t+1}}}}^{\boldsymbol{\pi_{\theta_{t+1}}}}(\boldsymbol{\rho}) - V_{\boldsymbol{\pi_{\theta_t}}}^{\boldsymbol{\pi_{\theta_t}}}(\boldsymbol{\rho}) \geq \eta \|\nabla_{\boldsymbol{\theta}} V_{\boldsymbol{\pi_{\theta_t}}}^{\boldsymbol{\pi_{\theta_t}}}(\boldsymbol{\rho})\|^2 - \frac{L}{2} \|\boldsymbol{\theta}_{t+1} - \boldsymbol{\theta}_t\|^2$$

This further implies that

$$V_{\boldsymbol{\pi_{\theta_{t+1}}}}^{\boldsymbol{\pi_{\theta_{t+1}}}}(\boldsymbol{\rho}) - V_{\boldsymbol{\pi_o^\star}}^{\boldsymbol{\pi_o^\star}}(\boldsymbol{\rho}) \geq V_{\boldsymbol{\pi_{\theta_t}}}^{\boldsymbol{\pi_{\theta_t}}}(\boldsymbol{\rho}) - V_{\boldsymbol{\pi_o^\star}}^{\boldsymbol{\pi_o^\star}}(\boldsymbol{\rho}) + \eta \|\nabla_{\boldsymbol{\theta}} V_{\boldsymbol{\pi_{\theta_t}}}^{\boldsymbol{\pi_{\theta_t}}}(\boldsymbol{\rho})\|^2 - \frac{L}{2} \|\boldsymbol{\theta}_{t+1} - \boldsymbol{\theta}_t\|^2$$

$$= V_{\boldsymbol{\pi_{\theta_t}}}^{\boldsymbol{\pi_{\theta_t}}}(\boldsymbol{\rho}) - V_{\boldsymbol{\pi_o^\star}}^{\boldsymbol{\pi_o^\star}}(\boldsymbol{\rho}) + \eta(1 - \frac{L\eta}{2}) \|\nabla V_{\boldsymbol{\pi_{\theta_t}}}^{\boldsymbol{\pi_{\theta_t}}}(\boldsymbol{\rho})\|^2 \qquad (32)$$

The last equality is due to Equation (5).

Now, telescoping Equation (32) leads to

$$\eta(1 - \frac{L\eta}{2}) \sum_{t=0}^{T-1} \|\nabla V_{\boldsymbol{\pi_{\theta_t}}}^{\boldsymbol{\pi_{\theta_t}}}(\boldsymbol{\rho})\|^2 \leq (\text{SubOpt}(\boldsymbol{\pi_{\theta_0}})) - \left( V_{\boldsymbol{\pi_o^\star}}^{\boldsymbol{\pi_o^\star}}(\boldsymbol{\rho}) - V_{\boldsymbol{\pi_{\theta_T}}}^{\boldsymbol{\pi_{\theta_T}}}(\boldsymbol{\rho}) \right)$$

$$\leq (\text{SubOpt}(\boldsymbol{\pi_{\theta_0}})) \qquad (33)$$

Since $\sum_{t=0}^{T-1} \|\nabla V_{\boldsymbol{\pi_{\theta_t}}}^{\boldsymbol{\pi_{\theta_t}}}(\boldsymbol{\rho})\|^2 \geq T \min_{t \in [T-1]} \|\nabla V_{\boldsymbol{\pi_{\theta_t}}}^{\boldsymbol{\pi_{\theta_t}}}(\boldsymbol{\rho})\|^2$, we obtain

$$\min_{t \in [T-1]} \|\nabla V_{\boldsymbol{\pi_{\theta_t}}}^{\boldsymbol{\pi_{\theta_t}}}(\boldsymbol{\rho})\|^2 \leq \frac{1}{T\eta \left(1 - \frac{L\eta}{2}\right)} (\text{SubOpt}(\boldsymbol{\pi_{\theta_0}})) \leq \frac{R_{\max}}{T\eta \left(1 - \frac{L\eta}{2}\right)(1 - \gamma)}.$$

The last inequality comes from $V_{\boldsymbol{\pi_o^\star}}^{\boldsymbol{\pi_o^\star}}(\boldsymbol{\rho}) \leq \frac{R_{\max}}{1-\gamma}$ (Assumption 1).

**Step 2:** We derive from lemma 3 - part (a) that

$$(\text{SubOpt}(\boldsymbol{\pi_\theta}))^2 \leq \left( \sqrt{|\mathcal{S}||\mathcal{A}|} \text{Cov} \|\nabla_{\boldsymbol{\theta}} V_{\boldsymbol{\pi_\theta}}^{\boldsymbol{\pi_\theta}}(\boldsymbol{\nu})\|_2 + \frac{1 + \text{Cov}}{(1-\gamma)^2} \left( L_r + L_{\mathbf{P}} R_{\max} \right) \right)^2$$

$$\leq 2|\mathcal{S}||\mathcal{A}| \text{Cov}^2 \|\nabla_{\boldsymbol{\theta}} V_{\boldsymbol{\pi_\theta}}^{\boldsymbol{\pi_\theta}}(\boldsymbol{\nu})\|_2^2 + 2 \left( \frac{1 + \text{Cov}}{(1-\gamma)^2} \left( L_r + L_{\mathbf{P}} R_{\max} \right) \right)^2.$$

Thus, we further get

$$\min_{t \in [T-1]} (\text{SubOpt}(\boldsymbol{\pi_{\theta_t}}))^2 \leq 2|\mathcal{S}||\mathcal{A}| \text{Cov}^2 \min_{t \in [T-1]} \|\nabla_{\boldsymbol{\theta}} V_{\boldsymbol{\pi_{\theta_t}}}^{\boldsymbol{\pi_{\theta_t}}}(\boldsymbol{\nu})\|_2^2 + 2 \left( \frac{1 + \text{Cov}}{(1-\gamma)^2} \left( L_r + L_{\mathbf{P}} R_{\max} \right) \right)^2$$

$$\leq 2|\mathcal{S}||\mathcal{A}| \text{Cov}^2 \frac{R_{\max}}{T\eta \left(1 - \frac{L\eta}{2}\right)(1 - \gamma)} + 2 \left( \frac{1 + \text{Cov}}{(1-\gamma)^2} \left( L_r + L_{\mathbf{P}} R_{\max} \right) \right)^2.$$

**Step 3:** Now, we set

$$\min_{t\in[T-1]}(\mathrm{SubOpt}(\boldsymbol{\pi}_{\boldsymbol{\theta}_t}))^2 \le 2|\mathcal{S}||\mathcal{A}|\mathsf{Cov}^2 \frac{R_{\max}}{T\eta\left(1-\frac{L\eta}{2}\right)(1-\gamma)} + 2\left(\frac{1+\mathsf{Cov}}{(1-\gamma)^2}\left(L_r + L_{\mathbf{P}}R_{\max}\right)\right)^2$$

$$\le \left(\sqrt{2|\mathcal{S}||\mathcal{A}|\frac{R_{\max}}{T\eta\left(1-\frac{L\eta}{2}\right)(1-\gamma)}}\mathsf{Cov} + \frac{\sqrt{2}+\sqrt{2}\mathsf{Cov}}{(1-\gamma)^2}\left(L_r + L_{\mathbf{P}}R_{\max}\right)\right)^2$$

$$\le \left(\epsilon + \frac{\sqrt{2}+\sqrt{2}\mathsf{Cov}}{(1-\gamma)^2}\left(L_r + L_{\mathbf{P}}R_{\max}\right)\right)^2,$$

and solve for $T$ to get $T \ge \frac{2|\mathcal{S}||\mathcal{A}|\mathsf{Cov}^2 R_{\max}}{\eta(1-\frac{L\eta}{2})(1-\gamma)\epsilon^2}$

Choosing $\eta = \frac{1}{L}$, we get the final expression $T \ge \frac{4L|\mathcal{S}||\mathcal{A}|\mathsf{Cov}^2 R_{\max}}{\epsilon^2(1-\gamma)}$, for any $\epsilon > 0$ and the smoothness constant $L$ is same as in lemma 5

Hence, we conclude that for $T = \Omega\left(\frac{|\mathcal{S}||\mathcal{A}|L\,\mathsf{Cov}^2}{\epsilon^2(1-\gamma)}\right)$,

$$\min_{t\in[T-1]}\mathrm{SubOpt}(\boldsymbol{\pi}_{\boldsymbol{\theta}_t}) \le \epsilon + \mathcal{O}\left(\frac{\mathsf{Cov}}{(1-\gamma)^2}\right).$$

$\square$

*Proof of Theorem 3 - part (b).* This proof follows similar steps as part (a) of Theorem 3 with two additional changes: (i) We have a $\lambda$, i.e. regularisation coefficient, dependent term due to the entropy regulariser. (ii) The maximum value of the soft value function is $\frac{R_{\max}+\lambda\log|\mathcal{A}|}{1-\gamma}$ instead of $\frac{R_{\max}}{1-\gamma}$ for the unregularised value function.

**Step 1:** From Equation (22), we observe that the soft-value function $\tilde{V}_{\boldsymbol{\pi}_{\boldsymbol{\theta}}}^{\boldsymbol{\pi}_{\boldsymbol{\theta}}}$ is $L_\lambda$-smooth.

Thus, following the Step 1 of part (a) of Theorem 3, we get

$$\min_{t\in[T-1]}\|\nabla\tilde{V}_{\boldsymbol{\pi}_{\boldsymbol{\theta}_t}}^{\boldsymbol{\pi}_{\boldsymbol{\theta}_t}}(\boldsymbol{\rho})\|^2 \le \frac{1}{T\eta\left(1-\frac{L_\lambda\eta}{2}\right)}\left(\tilde{V}_{\boldsymbol{\pi}_o^\star}^{\boldsymbol{\pi}_o^\star}(\boldsymbol{\rho}) - \tilde{V}_{\boldsymbol{\pi}_{\boldsymbol{\theta}_0}}^{\boldsymbol{\pi}_{\boldsymbol{\theta}_0}}(\boldsymbol{\rho})\right) \le \frac{R_{\max}+\lambda\log|\mathcal{A}|}{T\eta\left(1-\frac{L_\lambda\eta}{2}\right)(1-\gamma)}. \tag{34}$$

The last inequality is true due to the fact that $\tilde{V}_{\boldsymbol{\pi}_o^\star}^{\boldsymbol{\pi}_o^\star}(\boldsymbol{\rho}) - \tilde{V}_{\boldsymbol{\pi}_{\boldsymbol{\theta}_0}}^{\boldsymbol{\pi}_{\boldsymbol{\theta}_0}}(\boldsymbol{\rho}) \le \tilde{V}_{\boldsymbol{\pi}_o^\star}^{\boldsymbol{\pi}_o^\star}(\boldsymbol{\rho}) \le \frac{R_{\max}+\lambda\log|\mathcal{A}|}{1-\gamma}$.

**Step 2:** Now, from Part (b) of Lemma 3, we obtain that

$$\min_{t\in[T-1]}\left(\mathrm{SubOpt}(\boldsymbol{\theta}_t\mid\lambda)\right)^2$$

$$\le 2|\mathcal{S}||\mathcal{A}|\mathsf{Cov}^2\frac{R_{\max}}{T\eta\left(1-\frac{L\eta}{2}\right)(1-\gamma)} + 2\left(\frac{2+\mathsf{Cov}}{(1-\gamma)^2}\left(L_r + L_{\mathbf{P}}(R_{\max}+\lambda\log|\mathcal{A}|)\right)\right)^2$$

Thus, we conclude that

$$\min_{t\in[T-1]}\mathrm{SubOpt}(\boldsymbol{\theta}_t\mid\lambda)$$

$$\le \sqrt{\frac{2|\mathcal{S}||\mathcal{A}|\mathsf{Cov}^2\left(R_{\max}+\lambda\log|\mathcal{A}|\right)}{T\eta\left(1-\frac{L_\lambda\eta}{2}\right)(1-\gamma)}}$$

$$+ \sqrt{2}\left(\frac{2+\mathsf{Cov}}{(1-\gamma)^2}\left(L_r + L_{\mathbf{P}}(R_{\max}+\lambda\log|\mathcal{A}|)\right)\right), \tag{35}$$

**Step 4:** Now, by setting the $T$-dependent term in Equation (35) to $\epsilon$, we get $T \geq \frac{2|\mathcal{S}||\mathcal{A}|\mathsf{Cov}^2(R_{\max}+\lambda\log|\mathcal{A}|)}{\eta\left(1-\frac{L_\lambda\eta}{2}\right)(1-\gamma)\epsilon^2}$.

Choosing $\eta = \frac{1}{L_\lambda}, \lambda = \frac{(1-\gamma)R_{\max}}{(1+2\log|\mathcal{A}|)}$, we get the final expression $T \geq \frac{8|\mathcal{S}||\mathcal{A}|\mathsf{Cov}^2 L_\lambda R_{\max}}{(1-\gamma)\epsilon^2}$ , and

$$\min_{t\in[T-1]} \mathrm{SubOpt}(\boldsymbol{\theta}_t \mid \lambda) \leq \epsilon + \mathcal{O}\left(\frac{\mathsf{Cov}}{(1-\gamma)^2}\right) .$$

$\square$

# H. Convergence of PePG for Exponential PeMDPs

## H.1. Proofs for Unregularised Value Function

**Lemma 8** (Performative Policy Gradient for Softmax PeMDPs). *Given exponential family PeMDPs, for all* $(s, a, s') \in (\mathcal{S}, \mathcal{A}, \mathcal{S})$, *derivative of the performative value function w.r.t* $\boldsymbol{\theta}_{s,a}$ *satisfies:*

$$\frac{\partial V_{\boldsymbol{\pi}_{\boldsymbol{\theta}}}^{\boldsymbol{\pi}_{\boldsymbol{\theta}}}(\boldsymbol{\rho})}{\partial \boldsymbol{\theta}_{s,a}} \geq \frac{1}{1-\gamma} d_{\boldsymbol{\pi}_{\boldsymbol{\theta}}}^{\boldsymbol{\pi}_{\boldsymbol{\theta}}}(s, a|\boldsymbol{\rho}) \left(A_{\boldsymbol{\pi}_{\boldsymbol{\theta}}}^{\boldsymbol{\pi}_{\boldsymbol{\theta}}}(s, a) + \xi\right) . \tag{36}$$

*Proof.* First, we note that

$$\frac{\partial}{\partial \boldsymbol{\theta}_{s',a'}} \log \boldsymbol{\pi}_{\boldsymbol{\theta}}(a|s) = \mathbb{1}[s = s', a = a'] - \boldsymbol{\pi}_{\boldsymbol{\theta}}(a'|s)\mathbb{1}[s = s']$$

$$\frac{\partial}{\partial \boldsymbol{\theta}_{s',a'}} \log \mathbf{P}_{\boldsymbol{\pi}_{\boldsymbol{\theta}}}(s''|s,a) = \psi(s'')\mathbb{1}[s = s', a = a'] \left(1 - \mathbf{P}_{\boldsymbol{\pi}_{\boldsymbol{\theta}}}(s''|s,a)\right)$$

$$\frac{\partial}{\partial \boldsymbol{\theta}_{s',a'}} r_{\boldsymbol{\pi}_{\boldsymbol{\theta}}}(s,a) = \xi\mathbb{1}[s = s', a = a'] . \tag{37}$$

In this proof, we further substitute the expressions of individual gradients in Equation (37) into Equation (6).

Therefore, for a given initial state distribution $\boldsymbol{\rho}$, we get

$$\frac{\partial}{\partial \boldsymbol{\theta}_{s,a}} V_{\boldsymbol{\pi}_{\boldsymbol{\theta}}}^{\boldsymbol{\pi}_{\boldsymbol{\theta}}}(\boldsymbol{\rho})$$

$$= \mathbb{E}_{\tau\sim\mathbb{P}_{\boldsymbol{\pi}_{\boldsymbol{\theta}}}^{\boldsymbol{\pi}_{\boldsymbol{\theta}}}} \left[\sum_{t=0}^{\infty} \gamma^t \left(A_{\boldsymbol{\pi}_{\boldsymbol{\theta}}}^{\boldsymbol{\pi}_{\boldsymbol{\theta}}}(s_t, a_t)\frac{\partial}{\partial \boldsymbol{\theta}_{s,a}} \log \boldsymbol{\pi}_{\boldsymbol{\theta}}(a_t \mid s_t) + A_{\boldsymbol{\pi}_{\boldsymbol{\theta}}}^{\boldsymbol{\pi}_{\boldsymbol{\theta}}}(s_t, a_t)\frac{\partial}{\partial \boldsymbol{\theta}_{s,a}} \log P_{\boldsymbol{\pi}_{\boldsymbol{\theta}}}(s_{t+1}|s_t, a_t) + \frac{\partial}{\partial \boldsymbol{\theta}_{s,a}} r_{\boldsymbol{\pi}_{\boldsymbol{\theta}}}(s_t, a_t)\right)\right]$$

$$= \mathbb{E}_{\tau\sim\mathbb{P}_{\boldsymbol{\pi}_{\boldsymbol{\theta}}}^{\boldsymbol{\pi}_{\boldsymbol{\theta}}}} \left[\sum_{t=0}^{\infty} \gamma^t \left(A_{\boldsymbol{\pi}_{\boldsymbol{\theta}}}^{\boldsymbol{\pi}_{\boldsymbol{\theta}}}(s_t, a_t)(\mathbb{1}[s_t = s, a_t = a]\right.\right.$$

$$\left.\left. - \boldsymbol{\pi}_{\boldsymbol{\theta}}(a|s)\mathbb{1}[s_t = s])A_{\boldsymbol{\pi}_{\boldsymbol{\theta}}}^{\boldsymbol{\pi}_{\boldsymbol{\theta}}}(s_t, a_t)\psi(s_{t+1})\mathbb{1}[s_t = s, a_t = a]\left(1 - \mathbf{P}_{\boldsymbol{\pi}_{\boldsymbol{\theta}}}(s_{t+1}|s, a)\right) + \xi\mathbb{1}[s_t = s, a_t = a]\right)\right]$$

$$\underset{(a)}{\geq} \mathbb{E}_{\tau\sim\mathbb{P}_{\boldsymbol{\pi}_{\boldsymbol{\theta}}}^{\boldsymbol{\pi}_{\boldsymbol{\theta}}}} \left[\sum_{t=0}^{\infty} \gamma^t A_{\boldsymbol{\pi}_{\boldsymbol{\theta}}}^{\boldsymbol{\pi}_{\boldsymbol{\theta}}}(s_t, a_t)\mathbb{1}[s_t = s, a_t = a]\right] - \mathbb{E}_{\tau\sim\mathbb{P}_{\boldsymbol{\pi}_{\boldsymbol{\theta}}}^{\boldsymbol{\pi}_{\boldsymbol{\theta}}}} \left[\sum_{t=0}^{\infty} \gamma^t \boldsymbol{\pi}_{\boldsymbol{\theta}}(a|s)\mathbb{1}[s_t = s]A_{\boldsymbol{\pi}_{\boldsymbol{\theta}}}^{\boldsymbol{\pi}_{\boldsymbol{\theta}}}(s_t, a_t)\right]$$

$$+ \mathbb{E}_{\tau\sim\mathbb{P}_{\boldsymbol{\pi}_{\boldsymbol{\theta}}}^{\boldsymbol{\pi}_{\boldsymbol{\theta}}}} \left[\sum_{t=0}^{\infty} \gamma^t \xi\mathbb{1}[s_t = s, a_t = a]\right]$$

$$\underset{(b)}{=} \frac{1}{1-\gamma} d_{\boldsymbol{\pi}_{\boldsymbol{\theta}},\boldsymbol{\rho}}^{\boldsymbol{\pi}_{\boldsymbol{\theta}}}(s, a)A_{\boldsymbol{\pi}_{\boldsymbol{\theta}}}^{\boldsymbol{\pi}_{\boldsymbol{\theta}}}(s, a) + \frac{1}{1-\gamma} \xi d_{\boldsymbol{\pi}_{\boldsymbol{\theta}},\boldsymbol{\rho}}^{\boldsymbol{\pi}_{\boldsymbol{\theta}}}(s, a)$$

(a) is due to the fact that $1 - \mathbf{P}_{\boldsymbol{\pi}_{\boldsymbol{\theta}}}(s, a) \geq 0$ for all $s, a$. (b) is due to $\mathbb{E}_{\tau\sim\mathbb{P}_{\boldsymbol{\pi}_{\boldsymbol{\theta}}}^{\boldsymbol{\pi}_{\boldsymbol{\theta}}}}\left[\sum_{t=0}^{\infty} \gamma^t \boldsymbol{\pi}_{\boldsymbol{\theta}}(a|s)\mathbb{1}[s_t = s]A_{\boldsymbol{\pi}_{\boldsymbol{\theta}}}^{\boldsymbol{\pi}_{\boldsymbol{\theta}}}(s_t, a_t)\right] = 0$.

$\square$

**Lemma 4** (Performative Gradient Domination for Exponential PeMDPs). *Let us consider PeMDPs with softmax policies, linear rewards and exponential transitions.*

*(a) For unregularised value function,*

$$\text{SubOpt}(\boldsymbol{\pi_\theta}) \leq \sqrt{|\mathcal{S}||\mathcal{A}|}\text{Cov}\|\nabla_{\boldsymbol{\theta}}V_{\boldsymbol{\pi_\theta}}^{\boldsymbol{\pi_\theta}}(\boldsymbol{\nu})\|_2 + \frac{\gamma}{(1-\gamma)^2}R_{\max}\psi_{\max}. \tag{38}$$

*Proof of Lemma 4– Part (a).* This proof is divided into two parts. In the first part we bound the expected advantage term from Lemma 2 with the norm of the gradient of value function. During this step, we need to express the expected advantage as a linear combination of the advantage itself and the occupancy measure over all states and actions like in equation (36). The expectation however is taken w.r.t the occupancy measure $\boldsymbol{d}_{\boldsymbol{\pi_\theta^\star}}^{\boldsymbol{\pi_\theta^\star}}$, thus we need to perform a change of measure which introduces a coverage term as shown below. In the second step we directly use the bound of rewards and transitions obtained from their Lipchitzness in lemma 2. We know by Lemma 1 that

$$\text{SubOpt}(\boldsymbol{\pi_\theta}) = \frac{1}{1-\gamma}\underset{(s,a)\sim\boldsymbol{d}_{\boldsymbol{\pi_\theta}^\star,\rho}^{\boldsymbol{\pi_\theta^\star}}(\cdot|\rho)}{\mathbb{E}}[A_{\boldsymbol{\pi_\theta}}^{\boldsymbol{\pi_\theta}}(s,a)]$$

$$+ \frac{1}{1-\gamma}\underset{(s,a)\sim\boldsymbol{d}_{\boldsymbol{\pi_\theta},\rho}^{\boldsymbol{\pi_\theta^\star}}}{\mathbb{E}}\Big[(r_{\boldsymbol{\pi_o^\star}}(s,a) - r_{\boldsymbol{\pi_\theta}}(s,a)) + \gamma(\mathbf{P}_{\boldsymbol{\pi_o^\star}}(\cdot|s,a) - \mathbf{P}_{\boldsymbol{\pi_\theta}}(\cdot|s,a))^\top V_{\boldsymbol{\pi_o^\star}}^{\boldsymbol{\pi_o^\star}}(\cdot)\Big].$$

**Step 1: Upper bounding Term 1.**

$$\text{Term 1} \triangleq \underset{(s,a)\sim\boldsymbol{d}_{\boldsymbol{\pi_\theta},\rho}^{\boldsymbol{\pi_\theta^\star}}}{\mathbb{E}}[A_{\boldsymbol{\pi_\theta}}^{\boldsymbol{\pi_\theta}}(s,a)] = \sum_{s,a}\boldsymbol{d}_{\boldsymbol{\pi_\theta}}^{\boldsymbol{\pi_o^\star}}(s,a|\rho)A_{\boldsymbol{\pi_\theta}}^{\boldsymbol{\pi_\theta}}(s,a) = \sum_{s,a}\frac{\boldsymbol{d}_{\boldsymbol{\pi_\theta}}^{\boldsymbol{\pi_o^\star}}(s,a|\rho)}{\boldsymbol{d}_{\boldsymbol{\pi_\theta}}^{\boldsymbol{\pi_\theta}}(s,a|\nu)}\boldsymbol{d}_{\boldsymbol{\pi_\theta}}^{\boldsymbol{\pi_\theta}}(s,a|\nu)A_{\boldsymbol{\pi_\theta}}^{\boldsymbol{\pi_\theta}}(s,a)$$

$$\leq \left\|\frac{\boldsymbol{d}_{\boldsymbol{\pi_\theta},\rho}^{\boldsymbol{\pi_o^\star}}}{\boldsymbol{d}_{\boldsymbol{\pi_\theta},\nu}^{\boldsymbol{\pi_\theta}}}\right\|_\infty \sum_{s,a}\boldsymbol{d}_{\boldsymbol{\pi_\theta}}^{\boldsymbol{\pi_\theta}}(s,a|\nu)A_{\boldsymbol{\pi_\theta}}^{\boldsymbol{\pi_\theta}}(s,a) \tag{39}$$

Now, we leverage the gradient of softmax performative MDPs to obtain

$$\sum_{s,a}\boldsymbol{d}_{\boldsymbol{\pi_\theta}}^{\boldsymbol{\pi_\theta}}(s,a|\nu)A_{\boldsymbol{\pi_\theta}}^{\boldsymbol{\pi_\theta}}(s,a) \leq (1-\gamma)\sum_{s,a}\frac{\partial V_{\boldsymbol{\pi_\theta}}^{\boldsymbol{\pi_\theta}}(\nu)}{\partial\boldsymbol{\theta}_{s,a}} - \xi$$

$$= (1-\gamma)\mathbf{1}^\top\nabla_{\boldsymbol{\theta}}V_{\boldsymbol{\pi_\theta}}^{\boldsymbol{\pi_\theta}}(\nu) - \xi$$

$$\leq (1-\gamma)\sqrt{|\mathcal{S}||\mathcal{A}|}\|\nabla_{\boldsymbol{\theta}}V_{\boldsymbol{\pi_\theta}}^{\boldsymbol{\pi_\theta}}(\nu)\|_2 - \xi$$

The last inequality is obtained by applying Cauchy-Schwarz inequality.

Now, substituting the above result back in Equation (39), we get

$$\frac{1}{1-\gamma}\underset{(s,a)\sim\boldsymbol{d}_{\boldsymbol{\pi_\theta},\rho}^{\boldsymbol{\pi_o^\star}}(\cdot|s_0)}{\mathbb{E}}[A_{\boldsymbol{\pi_\theta}}^{\boldsymbol{\pi_\theta}}(s,a)] \leq \sqrt{|\mathcal{S}||\mathcal{A}|}\left\|\frac{\boldsymbol{d}_{\boldsymbol{\pi_\theta},\rho}^{\boldsymbol{\pi_o^\star}}}{\boldsymbol{d}_{\boldsymbol{\pi_\theta},\nu}^{\boldsymbol{\pi_\theta}}}\right\|_\infty\|\nabla_{\boldsymbol{\theta}}V_{\boldsymbol{\pi_\theta}}^{\boldsymbol{\pi_\theta}}(\nu)\|_2 - \left\|\frac{\boldsymbol{d}_{\boldsymbol{\pi_\theta},\rho}^{\boldsymbol{\pi_o^\star}}}{\boldsymbol{d}_{\boldsymbol{\pi_\theta},\nu}^{\boldsymbol{\pi_\theta}}}\right\|_\infty\frac{\xi}{1-\gamma} \tag{40}$$

**Step 2: Upper bounding Term 2.** For softmax rewards and transitions, we further obtain from Lemma 2,

$$\text{Term 2} \triangleq \frac{1}{1-\gamma}\underset{(s,a)\sim\boldsymbol{d}_{\boldsymbol{\pi_\theta},\rho}^{\boldsymbol{\pi_o^\star}}}{\mathbb{E}}\Big[(r_{\boldsymbol{\pi_o^\star}}(s,a) - r_{\boldsymbol{\pi_\theta}}(s,a)) + \gamma(\mathbf{P}_{\boldsymbol{\pi_o^\star}}(\cdot|s,a) - \mathbf{P}_{\boldsymbol{\pi_\theta}}(\cdot|s,a))^\top V_{\boldsymbol{\pi_o^\star}}^{\boldsymbol{\pi_o^\star}}(\cdot)\Big]$$

$$\leq \frac{1}{1-\gamma}(\xi + \frac{\gamma}{1-\gamma}R_{\max}\psi_{\max})\|\boldsymbol{\pi_o^\star}(\cdot|\rho) - \boldsymbol{\pi_\theta}(\cdot|\rho)\|_\infty$$

$$\leq \frac{1}{1-\gamma}(\xi + \frac{\gamma}{1-\gamma}R_{\max}\psi_{\max}). \tag{41}$$

**Step 3:** Now, if we use Equation (40) and (41) together, we get

$$\text{SubOpt}(\boldsymbol{\pi_\theta}) \leq \sqrt{|\mathcal{S}||\mathcal{A}|}\left\|\frac{\boldsymbol{d}_{\boldsymbol{\pi_\theta},\rho}^{\boldsymbol{\pi_o^\star}}}{\boldsymbol{d}_{\boldsymbol{\pi_\theta},\nu}^{\boldsymbol{\pi_\theta}}}\right\|_\infty\|\nabla_{\boldsymbol{\theta}}V_{\boldsymbol{\pi_\theta}}^{\boldsymbol{\pi_\theta}}(\nu)\|_2 + \left(1 - \left\|\frac{\boldsymbol{d}_{\boldsymbol{\pi_\theta},\rho}^{\boldsymbol{\pi_o^\star}}}{\boldsymbol{d}_{\boldsymbol{\pi_\theta},\nu}^{\boldsymbol{\pi_\theta}}}\right\|_\infty\right)\frac{\xi}{1-\gamma}$$

$$+ \frac{\gamma}{(1-\gamma)^2} R_{\max} \psi_{\max}$$

$$\leq \sqrt{|\mathcal{S}||\mathcal{A}|} \left\| \frac{d_{\pi_\theta,\rho}^{\pi_\theta^\star}}{d_{\pi_\theta,\nu}^{\pi_\theta}} \right\|_\infty \|\nabla_\theta V_{\pi_\theta}^{\pi_\theta}(\nu)\|_2 + \frac{\gamma}{(1-\gamma)^2} R_{\max} \psi_{\max}$$

The last inequality is true since $\left\| \frac{d_{\pi_\theta,\rho}^{\pi_o^\star}}{d_{\pi_\theta,\nu}^{\pi_\theta}} \right\|_\infty \geq 1$ (Lemma 11). For the final step, we use $\left\| \frac{d_{\pi_\theta,\rho}^{\pi_o^\star}}{d_{\pi_\theta,\nu}^{\pi_\theta}} \right\|_\infty \leq \mathsf{Cov}$. $\qquad \square$

**Note:** Since we are working with exponential family transitions, we can substitute $\psi_{\max} = \mathcal{O}\left(\frac{1-\gamma}{\gamma}\right)$ in (38) to get the final version of the unregularised gradient domination reported in the main paper.

**Theorem 4** (Convergence of PePG for Exponential family PeMDPs). *We set learning rate $\eta = \mathcal{O}\left(\frac{(1-\gamma)^2}{|\mathcal{A}|}\right)$, and $\psi_{\max} = \mathcal{O}(\frac{1-\gamma}{\gamma})$. Then,*

$$\min_{t<T} \mathrm{SubOpt}(\pi_{\theta_t}) \leq \epsilon + \mathcal{O}\left(\frac{R_{\max}}{1-\gamma}\right),$$

*when (a) $T = \Omega\left(\frac{|\mathcal{S}||\mathcal{A}|R_{\max}\mathsf{Cov}^2}{\epsilon^2(1-\gamma)^3} \max\left\{R_{\max} \mid \mathcal{A} \mid, \frac{\gamma}{(1-\gamma)}\right\}\right)$ and $\psi_{\max} = \mathcal{O}(\frac{1-\gamma}{\gamma})$ for unregularised objective.*

*Proof of Part (a).* We proceed with this proof by dividing it in four steps. In the first step, we use the smoothness of the value function to prove an upper bound for the minimum squared gradient norm of the value over time which is a constant times $1/T$. In the second step, we derive a lower bound on the norm of gradient of value function using Lemma 4. In the final two steps, we combine the bounds obtained from the first two steps to derive lower bounds for $T$ and $\epsilon$, i.e. the error threshold.

**Step 1:** As $V_{\pi_\theta}^{\pi_\theta}$ is $L$-smooth (Lemma 5), it satisfies

$$\left| V_{\pi_\theta}^{\pi_\theta}(\rho) - V_{\pi_\theta'}^{\pi_\theta'}(\rho) - \langle \nabla_\theta V_{\pi_\theta}^{\pi_\theta}(\rho), \theta - \theta' \rangle \right| \leq \frac{L}{2}\|\theta - \theta'\|^2$$

Thus, taking $\theta$ as $\theta_{t+1}$ and $\theta'$ as $\theta_t$ and using the gradient ascent expression (Equation (5)) yields

$$\left| V_{\pi_{\theta_{t+1}}}^{\pi_{\theta_{t+1}}}(\rho) - V_{\pi_{\theta_t}}^{\pi_{\theta_t}}(\rho) - \eta\|\nabla_\theta V_{\pi_{\theta_t}}^{\pi_{\theta_t}}(\rho)\|^2 \right| \leq \frac{L}{2}\|\theta_{t+1} - \theta_t\|^2$$

$$\implies \quad V_{\pi_{\theta_{t+1}}}^{\pi_{\theta_{t+1}}}(\rho) - V_{\pi_{\theta_t}}^{\pi_{\theta_t}}(\rho) \geq \eta\|\nabla_\theta V_{\pi_{\theta_t}}^{\pi_{\theta_t}}(\rho)\|^2 - \frac{L}{2}\|\theta_{t+1} - \theta_t\|^2$$

This further implies that

$$V_{\pi_{\theta_{t+1}}}^{\pi_{\theta_{t+1}}}(\rho) - V_{\pi_o^\star}^{\pi_o^\star}(\rho) \geq V_{\pi_{\theta_t}}^{\pi_{\theta_t}}(\rho) - V_{\pi_o^\star}^{\pi_o^\star}(\rho) + \eta\|\nabla_\theta V_{\pi_{\theta_t}}^{\pi_{\theta_t}}(\rho)\|^2 - \frac{L}{2}\|\theta_{t+1} - \theta_t\|^2$$

$$= V_{\pi_{\theta_t}}^{\pi_{\theta_t}}(\rho) - V_{\pi_o^\star}^{\pi_o^\star}(\rho) + \eta(1 - \frac{L\eta}{2})\|\nabla V_{\pi_{\theta_t}}^{\pi_{\theta_t}}(\rho)\|^2 \tag{42}$$

The last equality is due to Equation (5).

Now, telescoping Equation (42) leads to

$$\eta(1 - \frac{L\eta}{2}) \sum_{t=0}^{T-1} \|\nabla V_{\pi_{\theta_t}}^{\pi_{\theta_t}}(\rho)\|^2 \leq \left(V_{\pi_o^\star}^{\pi_o^\star}(\rho) - V_{\pi_{\theta_0}}^{\pi_{\theta_0}}(\rho)\right) - \left(V_{\pi_o^\star}^{\pi_o^\star}(\rho) - V_{\pi_{\theta_T}}^{\pi_{\theta_T}}(\rho)\right)$$

$$\leq \left(V_{\pi_o^\star}^{\pi_o^\star}(\rho) - V_{\pi_{\theta_0}}^{\pi_{\theta_0}}(\rho)\right) \tag{43}$$

Since $\sum_{t=0}^{T-1} \|\nabla V_{\pi_{\theta_t}}^{\pi_{\theta_t}}(\rho)\|^2 \geq T \min_{t\in[T-1]} \|\nabla V_{\pi_{\theta_t}}^{\pi_{\theta_t}}(\rho)\|^2$, we obtain

$$\min_{t\in[T-1]} \|\nabla V_{\pi_{\theta_t}}^{\pi_{\theta_t}}(\rho)\|^2 \leq \frac{1}{T\eta\left(1-\frac{L\eta}{2}\right)}\left(V_{\pi_o^\star}^{\pi_o^\star}(\rho) - V_{\pi_{\theta_0}}^{\pi_{\theta_0}}(\rho)\right) \leq \frac{R_{\max}}{T\eta\left(1-\frac{L\eta}{2}\right)(1-\gamma)} .$$

The last inequality comes from $V_{\pi_o^\star}^{\pi_o^\star}(\rho) \leq \frac{R_{\max}}{1-\gamma}$ (Assumption 1).

**Step 2:** We derive from Equation (38) that

$$(\mathrm{SubOpt}(\pi_\theta))^2 \leq \left(\sqrt{|\mathcal{S}||\mathcal{A}|}\left\|\frac{d_{\pi_\theta,\rho}^{\pi_o^\star}}{d_{\pi_\theta,\nu}^{\pi_\theta}}\right\|_\infty \|\nabla_\theta V_{\pi_\theta}^{\pi_\theta}(\nu)\|_2 + \frac{\gamma R_{\max}}{(1-\gamma)^2}\psi_{\max}\right)^2$$

$$\leq 2|\mathcal{S}||\mathcal{A}|\left\|\frac{d_{\pi_\theta,\rho}^{\pi_o^\star}}{d_{\pi_\theta,\nu}^{\pi_\theta}}\right\|_\infty^2 \|\nabla_\theta V_{\pi_\theta}^{\pi_\theta}(\nu)\|_2^2 + \frac{2\gamma^2 R_{\max}^2}{(1-\gamma)^4}\psi_{\max}^2$$

Thus, we further get

$$\min_{t\in[T-1]}(\mathrm{SubOpt}(\pi_{\theta_t}))^2 \leq 2|\mathcal{S}||\mathcal{A}| \min_{t\in[T-1]}\left\|\frac{d_{\pi_{\theta_t},\rho}^{\pi_o^\star}}{d_{\pi_{\theta_t},\nu}^{\pi_\theta}}\right\|_\infty^2 \|\nabla_\theta V_{\pi_{\theta_t}}^{\pi_{\theta_t}}(\nu)\|_2^2 + \frac{2\gamma^2 R_{\max}^2}{(1-\gamma)^4}\psi_{\max}^2$$

$$\leq 2|\mathcal{S}||\mathcal{A}|\mathsf{Cov}^2 \min_{t\in[T-1]}\|\nabla_\theta V_{\pi_{\theta_t}}^{\pi_{\theta_t}}(\nu)\|_2^2 + \frac{2\gamma^2 R_{\max}^2}{(1-\gamma)^4}\psi_{\max}^2$$

$$\leq 2|\mathcal{S}||\mathcal{A}|\mathsf{Cov}^2 \frac{R_{\max}}{T\eta\left(1-\frac{L\eta}{2}\right)(1-\gamma)} + \frac{2\gamma^2 R_{\max}^2}{(1-\gamma)^4}\psi_{\max}^2 .$$

**Step 3:** Now, we set

$$\min_{t\in[T-1]}(\mathrm{SubOpt}(\pi_{\theta_t}))^2 \leq 2|\mathcal{S}||\mathcal{A}|\mathsf{Cov}^2 \frac{R_{\max}}{T\eta\left(1-\frac{L\eta}{2}\right)(1-\gamma)} + \frac{2\gamma^2 R_{\max}^2}{(1-\gamma)^4}\psi_{\max}^2$$

$$\leq \left(\sqrt{2|\mathcal{S}||\mathcal{A}|\frac{R_{\max}}{T\eta\left(1-\frac{L\eta}{2}\right)(1-\gamma)}}\mathsf{Cov} + \frac{\sqrt{2}\gamma R_{\max}}{(1-\gamma)^2}\psi_{\max}\right)^2$$

$$\implies \min_{t\in[T-1]}\mathrm{SubOpt}(\pi_{\theta_t}) \leq \left(\epsilon + \frac{\sqrt{2}\gamma R_{\max}}{(1-\gamma)^2}\psi_{\max}\right),$$

and solve for $T$ to get

$$T \geq \frac{2|\mathcal{S}||\mathcal{A}|\mathsf{Cov}^2 R_{\max}}{\eta(1-\frac{L\eta}{2})(1-\gamma)\epsilon^2} \tag{44}$$

Choosing $\eta = \frac{1}{L}$, we get the final expression

$$T \geq \frac{4L|\mathcal{S}||\mathcal{A}|\mathsf{Cov}^2 R_{\max}}{\epsilon^2(1-\gamma)} . \tag{45}$$

for any $\epsilon > 0$ and the smoothness constant $L = \mathcal{O}\left(\max\left\{\frac{\gamma R_{\max}|\mathcal{A}|}{(1-\gamma)^2}, \frac{\gamma^2}{(1-\gamma)^3}\right\}\right)$.

Hence, we conclude that for $T = \Omega\left(\frac{|\mathcal{S}||\mathcal{A}|R_{\max}\mathsf{Cov}^2}{\epsilon^2(1-\gamma)^3}\max\left\{R_{\max}\mid\mathcal{A}\mid, \frac{\gamma}{(1-\gamma)}\right\}\right)$ and $\psi_{\max} = \mathcal{O}(\frac{1-\gamma}{\gamma})$,

$$\min_{t\in[T-1]}\mathrm{SubOpt}(\pi_{\theta_t}) \leq \epsilon + \mathcal{O}\left(\frac{R_{\max}}{1-\gamma}\right) .$$

$\square$

## H.2. Proofs for Entropy-regularised or Soft Value Function

**Lemma 9** (Regularized Performative Policy Difference: Upper Bound for PeMDPs with linear rewards, exponential transition and softmax policy class). *Under Assumption 1, the sub-optimality gap of a policy $\pi_{\boldsymbol{\theta}}$ is*

$$
\begin{aligned}
\mathrm{SubOpt}(\boldsymbol{\pi_\theta}|\lambda) \leq{} & \frac{1}{1-\gamma} \underset{(s,a)\sim \boldsymbol{d}^{\pi_\theta^\star}_{\pi_\theta,\boldsymbol{\rho}}}{\mathbb{E}} [\tilde{A}^{\boldsymbol{\pi_\theta}}_{\boldsymbol{\pi_\theta}}(s,a)] \\
& + \frac{1}{1-\gamma}\Big(\xi + \frac{\gamma}{1-\gamma}\psi_{\max}(R_{\max} + \lambda\log|\mathcal{A}|)\Big) \\
& - \frac{\lambda}{1-\gamma}\sum_s \boldsymbol{d}^{\pi_o^\star}_{\pi_\theta}(s|\boldsymbol{\rho}) D_{\mathrm{KL}}\left(\boldsymbol{\pi}^\star_o(\cdot|s)\,\|\,\boldsymbol{\pi_\theta}(\cdot|s)\right)
\end{aligned}
\tag{46}
$$

*Proof.* This lemma follows the same sketch as Lemma 7 but replacing the constants $L_r$ and $L_{\mathbf{P}}$ with constants $\xi$ and $\psi_{\max}$ specific to the given choice of rewards and transitions. $\qquad\square$

**Lemma 10** (Regularized Performative Policy gradient for softmax policies and softmax MDPs). *For a class of PeMDPs $\mathcal{M} \triangleq (\mathcal{S}, \mathcal{A}, \boldsymbol{\pi}, \mathbf{P}_{\boldsymbol{\pi}}, r_{\boldsymbol{\pi}}, \boldsymbol{\theta}, \boldsymbol{\rho})$ consider softmax parametrization for policy $\boldsymbol{\pi_\theta} \in \Delta(\boldsymbol{\theta} \in \boldsymbol{\Theta})$ and transition dynamics $\mathbf{P}_{\boldsymbol{\pi_\theta}}$ and linear parametrization for reward $r_{\boldsymbol{\pi_\theta}}$. For all $(s,a,s') \in (\mathcal{S}, \mathcal{A}, \mathcal{S})$, derivative of the expected return w.r.t $\boldsymbol{\theta}_{s,a}$ satisfies:*

$$
\frac{\partial \tilde{V}^{\boldsymbol{\pi_\theta}}_{\boldsymbol{\pi_\theta}}(\boldsymbol{\rho})}{\partial \boldsymbol{\theta}_{s,a}} \geq \frac{1}{1-\gamma}\boldsymbol{d}^{\boldsymbol{\pi_\theta}}_{\boldsymbol{\pi_\theta},\boldsymbol{\rho}}(s,a)\left(\tilde{A}^{\boldsymbol{\pi_\theta}}_{\boldsymbol{\pi_\theta}}(s,a) + \xi\right) - \frac{\lambda}{1-\gamma}\boldsymbol{d}^{\boldsymbol{\pi_\theta}}_{\boldsymbol{\pi_\theta},\boldsymbol{\rho}}(s,a)\log|\mathcal{A}| .
\tag{47}
$$

*Proof.* This proof follows the same sketch as the proof of Lemma 4. However, we get two additional $\lambda$-dependent terms– (a) one from the log policy term in the soft advantage, and (b) the other from the log policy term in the soft rewards. We then simplify these terms to obtain the final expression.

First, let us note that

$$
\begin{aligned}
\frac{\partial}{\partial \boldsymbol{\theta}_{s',a'}}\log \boldsymbol{\pi_\theta}(a|s) &= \mathbb{1}[s=s', a=a'] - \boldsymbol{\pi_\theta}(a'|s')\mathbb{1}[s=s'] \\
\frac{\partial}{\partial \boldsymbol{\theta}_{s',a'}}\log \mathbf{P}_{\boldsymbol{\pi_\theta}}(s''|s,a) &= \psi(s'')\mathbb{1}[s=s', a=a']\left(1 - \mathbf{P}_{\boldsymbol{\pi_\theta}}(s''|s,a)\right) \\
\frac{\partial}{\partial \boldsymbol{\theta}_{s',a'}}r_{\boldsymbol{\pi_\theta}}(s,a) &= \xi\mathbb{1}[s=s', a=a'] .
\end{aligned}
\tag{48}
$$

Now, we get from Theorem 2,

$$
\begin{aligned}
&\frac{\partial}{\partial \boldsymbol{\theta}_{s,a}}\tilde{V}^{\boldsymbol{\pi_\theta}}_{\boldsymbol{\pi_\theta}}(\boldsymbol{\rho}) \\
={} & \underset{\tau\sim\mathbb{P}^{\boldsymbol{\pi_\theta}}_{\boldsymbol{\pi_\theta}}}{\mathbb{E}}\Big[\sum_{t=0}^{\infty}\gamma^t\Big(\tilde{A}^{\boldsymbol{\pi_\theta}}_{\boldsymbol{\pi_\theta}}(s_t,a_t)\frac{\partial}{\partial \boldsymbol{\theta}_{s,a}}\log \boldsymbol{\pi_\theta}(a_t\mid s_t) + \tilde{A}^{\boldsymbol{\pi_\theta}}_{\boldsymbol{\pi_\theta}}(s_t,a_t)\frac{\partial}{\partial \boldsymbol{\theta}_{s,a}}\log P_{\boldsymbol{\pi_\theta}}(s_{t+1}|s_t,a_t) \\
&\qquad + \frac{\partial}{\partial \boldsymbol{\theta}_{s,a}}r_{\boldsymbol{\pi_\theta}}(s_t,a_t) - \lambda\frac{\partial}{\partial \boldsymbol{\theta}_{s,a}}\log \boldsymbol{\pi_\theta}(a_t\mid s_t)\Big)\Big] \\
={} & \underset{\tau\sim\mathbb{P}^{\boldsymbol{\pi_\theta}}_{\boldsymbol{\pi_\theta}}}{\mathbb{E}}\Big[\sum_{t=0}^{\infty}\gamma^t\Big(\tilde{A}^{\boldsymbol{\pi_\theta}}_{\boldsymbol{\pi_\theta}}(s_t,a_t)\left(\mathbb{1}[s_t=s, a_t=a] - \boldsymbol{\pi_\theta}(a|s)\mathbb{1}[s_t=s]\right) + \tilde{A}^{\boldsymbol{\pi_\theta}}_{\boldsymbol{\pi_\theta}}(s_t,a_t)\psi(s_{t+1})\mathbb{1}[s_t=s, a_t=a] \\
&\qquad \left(1 - \mathbf{P}_{\boldsymbol{\pi_\theta}}(s_{t+1}|s,a)\right) + \xi\mathbb{1}[s_t=s, a_t=a] - \lambda\mathbb{1}[s_t=s, a_t=a] + \lambda\boldsymbol{\pi_\theta}(a|s)\mathbb{1}[s_t=s]\Big)\Big] \\
\underset{(a)}{\geq}{} & \underset{\tau\sim\mathbb{P}^{\boldsymbol{\pi_\theta}}_{\boldsymbol{\pi_\theta}}}{\mathbb{E}}\Big[\sum_{t=0}^{\infty}\gamma^t\tilde{A}^{\boldsymbol{\pi_\theta}}_{\boldsymbol{\pi_\theta}}(s_t,a_t)\mathbb{1}[s_t=s, a_t=a]\Big] - \underset{\tau\sim\mathbb{P}^{\boldsymbol{\pi_\theta}}_{\boldsymbol{\pi_\theta}}}{\mathbb{E}}\Big[\sum_{t=0}^{\infty}\gamma^t\boldsymbol{\pi_\theta}(a|s)\mathbb{1}[s_t=s]\tilde{A}^{\boldsymbol{\pi_\theta}}_{\boldsymbol{\pi_\theta}}(s_t,a_t)\Big] \\
&\quad + \underset{\tau\sim\mathbb{P}^{\boldsymbol{\pi_\theta}}_{\boldsymbol{\pi_\theta}}}{\mathbb{E}}\Big[\sum_{t=0}^{\infty}\gamma^t\xi\mathbb{1}[s_t=s, a_t=a]\Big] - \lambda\underset{\tau\sim\mathbb{P}^{\boldsymbol{\pi_\theta}}_{\boldsymbol{\pi_\theta}}}{\mathbb{E}}\Big[\sum_{t=0}^{\infty}\gamma^t\mathbb{1}[s_t=s, a_t=a]\Big]
\end{aligned}
$$

$$+ \lambda \mathop{\mathbb{E}}_{\tau \sim \mathbb{P}^{\pi_{\boldsymbol{\theta}}}_{\pi_{\boldsymbol{\theta}}}} \left[ \sum_{t=0}^{\infty} \gamma^t \pi_{\boldsymbol{\theta}}(a|s) \mathbb{1}[s_t = s] \right]$$

$$\underset{(b)}{=} \frac{1}{1-\gamma} d^{\pi_{\boldsymbol{\theta}}}_{\pi_{\boldsymbol{\theta}},\boldsymbol{\rho}}(s,a) \tilde{A}^{\pi_{\boldsymbol{\theta}}}_{\pi_{\boldsymbol{\theta}}}(s,a) + \lambda \mathop{\mathbb{E}}_{\tau \sim \mathbb{P}^{\pi_{\boldsymbol{\theta}}}_{\pi_{\boldsymbol{\theta}}}} \left[ \sum_{t=0}^{\infty} \gamma^t \pi_{\boldsymbol{\theta}}(a|s) \log \pi_{\boldsymbol{\theta}}(a_t|s_t) \mathbb{1}[s_t = s] \right] + \frac{1}{1-\gamma} \xi d^{\pi_{\boldsymbol{\theta}}}_{\pi_{\boldsymbol{\theta}},\boldsymbol{\rho}}(s,a)$$

$$= \frac{1}{1-\gamma} d^{\pi_{\boldsymbol{\theta}}}_{\pi_{\boldsymbol{\theta}},\boldsymbol{\rho}}(s,a) \tilde{A}^{\pi_{\boldsymbol{\theta}}}_{\pi_{\boldsymbol{\theta}}}(s,a) + \lambda \mathop{\mathbb{E}}_{\tau \sim \mathbb{P}^{\pi_{\boldsymbol{\theta}}}_{\pi_{\boldsymbol{\theta}}}} \left[ \sum_{t=0}^{\infty} \gamma^t \pi_{\boldsymbol{\theta}}(a|s) \log \pi_{\boldsymbol{\theta}}(a_t|s_t) \sum_a \mathbb{1}[s_t = s, a_t = a] \right] + \frac{1}{1-\gamma} \xi d^{\pi_{\boldsymbol{\theta}}}_{\pi_{\boldsymbol{\theta}},\boldsymbol{\rho}}(s,a)$$

$$= \frac{1}{1-\gamma} d^{\pi_{\boldsymbol{\theta}}}_{\pi_{\boldsymbol{\theta}},\boldsymbol{\rho}}(s,a) \tilde{A}^{\pi_{\boldsymbol{\theta}}}_{\pi_{\boldsymbol{\theta}}}(s,a) + \lambda \pi_{\boldsymbol{\theta}}(a|s) \mathop{\mathbb{E}}_{\tau \sim \mathbb{P}^{\pi_{\boldsymbol{\theta}}}_{\pi_{\boldsymbol{\theta}}}} \left[ \sum_a \log \pi_{\boldsymbol{\theta}}(a|s) \sum_{t=0}^{\infty} \gamma^t \mathbb{1}[s_t = s, a_t = a] \right] + \frac{1}{1-\gamma} \xi d^{\pi_{\boldsymbol{\theta}}}_{\pi_{\boldsymbol{\theta}},\boldsymbol{\rho}}(s,a)$$

$$= \frac{1}{1-\gamma} d^{\pi_{\boldsymbol{\theta}}}_{\pi_{\boldsymbol{\theta}},\boldsymbol{\rho}}(s,a) \left( \tilde{A}^{\pi_{\boldsymbol{\theta}}}_{\pi_{\boldsymbol{\theta}}}(s,a) + \xi \right) + \frac{\lambda}{1-\gamma} \pi_{\boldsymbol{\theta}}(a|s) \sum_a d^{\pi_{\boldsymbol{\theta}}}_{\pi_{\boldsymbol{\theta}},\boldsymbol{\rho}}(s,a) \log \pi_{\boldsymbol{\theta}}(a|s)$$

$$\underset{(c)}{\geq} \frac{1}{1-\gamma} d^{\pi_{\boldsymbol{\theta}}}_{\pi_{\boldsymbol{\theta}},\boldsymbol{\rho}}(s,a) \left( \tilde{A}^{\pi_{\boldsymbol{\theta}}}_{\pi_{\boldsymbol{\theta}}}(s,a) + \xi \right) - \frac{\lambda}{1-\gamma} d^{\pi_{\boldsymbol{\theta}}}_{\pi_{\boldsymbol{\theta}},\boldsymbol{\rho}}(s,a) \log |\mathcal{A}| .$$

(b) holds as:

$$\mathop{\mathbb{E}}_{\tau \sim \mathbb{P}^{\pi_{\boldsymbol{\theta}}}_{\pi_{\boldsymbol{\theta}}}} \left[ \sum_{t=0}^{\infty} \gamma^t \mathbb{1}[s_t = s, a_t = a] \right] - \mathop{\mathbb{E}}_{\tau \sim \mathbb{P}^{\pi_{\boldsymbol{\theta}}}_{\pi_{\boldsymbol{\theta}}}} \left[ \sum_{t=0}^{\infty} \gamma^t \pi_{\boldsymbol{\theta}}(a|s) \mathbb{1}[s_t = s] \right] = d^{\pi_{\boldsymbol{\theta}}}_{\pi_{\boldsymbol{\theta}},\boldsymbol{\rho}}(s,a) - d^{\pi_{\boldsymbol{\theta}}}_{\pi_{\boldsymbol{\theta}},\boldsymbol{\rho}}(s) \pi_{\boldsymbol{\theta}}(a|s) = 0$$

(c) holds from the following:

$$-\sum_a d^{\pi_{\boldsymbol{\theta}}}_{\pi_{\boldsymbol{\theta}}}(s,a|s_0) \log \pi_{\boldsymbol{\theta}}(a|s) = d^{\pi_{\boldsymbol{\theta}}}_{\pi_{\boldsymbol{\theta}}}(s|s_0) \left( -\sum_a \pi_{\boldsymbol{\theta}}(a|s) \log \pi_{\boldsymbol{\theta}}(a|s) \right)$$

$$\underset{(d)}{\leq} d^{\pi_{\boldsymbol{\theta}}}_{\pi_{\boldsymbol{\theta}}}(s|s_0) \log |\mathcal{A}|$$

and (d) holds as entropy is upper bounded by $\log |\mathcal{A}|$ (Cover & Thomas, 2006, Theorem 2.6.4).

$\square$

**Lemma 4** (Regularized Performative Gradient Domination: Part(b) of Lemma 4). *For regularized PeMDPs the following inequality holds:*

$$\mathrm{SubOpt}(\pi_{\boldsymbol{\theta}} \mid \lambda) \leq \sqrt{|\mathcal{S}||\mathcal{A}|} \mathsf{Cov} \|\nabla_{\boldsymbol{\theta}} \tilde{V}^{\pi_{\boldsymbol{\theta}}}_{\pi_{\boldsymbol{\theta}}}(\boldsymbol{\nu})\|_2 + \frac{\lambda \log |\mathcal{A}|}{1-\gamma} \left( \frac{\gamma \psi_{\max}}{1-\gamma} + 2 \right) + \frac{\gamma \psi_{\max} R_{\max}}{(1-\gamma)^2} . \tag{49}$$

*Proof.* **Step 1.** First, we observe that

$$-D_{\mathrm{KL}} \left( \pi^{\star}_o(\cdot|s) \parallel \pi_{\boldsymbol{\theta}}(\cdot|s) \right) \leq -\sum_{a \in \mathcal{A}} \pi^{\star}_o(a|s) \log \pi^{\star}_o(a|s) \leq \log |\mathcal{A}|$$

Hence, we get

$$-\sum_s d^{\pi^{\star}_o}_{\pi_{\boldsymbol{\theta}}}(s|s_0) D_{\mathrm{KL}} \left( \pi^{\star}_o(\cdot|s) \parallel \pi_{\boldsymbol{\theta}}(\cdot|s) \right) \leq \log |\mathcal{A}| \tag{50}$$

**Step 2.** Using Lemma 10 and applying Cauchy-Schwarz inequality, we get

$$\sum_{s,a} d^{\pi_{\boldsymbol{\theta}}}_{\pi_{\boldsymbol{\theta}}}(s,a) \tilde{A}^{\pi_{\boldsymbol{\theta}}}_{\pi_{\boldsymbol{\theta}}}(s,a) \leq \sqrt{|\mathcal{S}||\mathcal{A}|}(1-\gamma) \|\nabla_{\boldsymbol{\theta}} \tilde{V}^{\pi_{\boldsymbol{\theta}}}_{\pi_{\boldsymbol{\theta}}}(\boldsymbol{\nu})\|_2 - \xi + \lambda \log |\mathcal{A}| \tag{51}$$

**Step 3.** Now, substituting Equation (50) and (51) in Equation (46), we finally get

$$\mathrm{SubOpt}(\pi_{\boldsymbol{\theta}} \mid \lambda) \leq \sqrt{|\mathcal{S}||\mathcal{A}|} \mathsf{Cov} \|\nabla_{\boldsymbol{\theta}} \tilde{V}^{\pi_{\boldsymbol{\theta}}}_{\pi_{\boldsymbol{\theta}}}(\boldsymbol{\nu})\|_2 - \mathsf{Cov} \frac{\xi}{1-\gamma} + \frac{\lambda}{1-\gamma} \log |\mathcal{A}|$$

$$+ \frac{1}{1-\gamma}\left(L_r + \frac{\gamma}{1-\gamma}L_{\mathbf{P}}(R_{\max} + \lambda \log|\mathcal{A}|)\right) + \frac{\lambda}{1-\gamma}\log|\mathcal{A}|$$

$$\underset{(a)}{=} \sqrt{|\mathcal{S}||\mathcal{A}|}\mathsf{Cov}\|\nabla_{\boldsymbol{\theta}}\tilde{V}_{\boldsymbol{\pi_\theta}}^{\boldsymbol{\pi_\theta}}(\boldsymbol{\nu})\|_2 - \mathsf{Cov}\frac{\xi}{1-\gamma} + \frac{\lambda}{1-\gamma}2\log|\mathcal{A}|$$

$$+ \frac{1}{1-\gamma}\left(\xi + \frac{\gamma}{1-\gamma}\psi_{\max}(R_{\max} + \lambda \log|\mathcal{A}|)\right)$$

$$\underset{(b)}{\leq} \sqrt{|\mathcal{S}||\mathcal{A}|}\mathsf{Cov}\|\nabla_{\boldsymbol{\theta}}\tilde{V}_{\boldsymbol{\pi_\theta}}^{\boldsymbol{\pi_\theta}}(\boldsymbol{\nu})\|_2$$

$$+ \frac{\gamma}{(1-\gamma)^2}\psi_{\max}(R_{\max} + \lambda \log|\mathcal{A}|) + \frac{\lambda}{1-\gamma}2\log|\mathcal{A}|$$

$$= \sqrt{|\mathcal{S}||\mathcal{A}|}\mathsf{Cov}\|\nabla_{\boldsymbol{\theta}}\tilde{V}_{\boldsymbol{\pi_\theta}}^{\boldsymbol{\pi_\theta}}(\boldsymbol{\nu})\|_2 + \frac{\lambda \log|\mathcal{A}|}{1-\gamma}\left(\frac{\gamma\psi_{\max}}{1-\gamma} + 2\right) + \frac{\gamma\psi_{\max}R_{\max}}{(1-\gamma)^2}$$

In (a), we substitute the values of $L_r$ and $L_{\mathbf{P}}$ for softmax PeMDPs, and in (b), we use $\mathsf{Cov} \geq 1$ (Lemma 11).

$\square$

**Note:** Analogous to the unregularised case, we can substitute $\psi_{\max}$ with $\mathcal{O}\left(\frac{1-\gamma}{\gamma}\right)$ to get a simpler bound as reported in the main paper.

**Theorem 4** (Convergence of PePG for Exponential family PeMDPs)**.** *We set learning rate* $\eta = \mathcal{O}\left(\frac{(1-\gamma)^2}{|\mathcal{A}|}\right)$*, and* $\psi_{\max} = \mathcal{O}(\frac{1-\gamma}{\gamma})$*. Then,*

$$\min_{t<T} \mathrm{SubOpt}(\boldsymbol{\pi}_{\boldsymbol{\theta}_t}) \leq \epsilon + \mathcal{O}\left(\frac{R_{\max}}{1-\gamma}\right),$$

*(b)* $T = \Omega\left(\frac{|\mathcal{S}||\mathcal{A}|^2 R_{\max}^2 \mathsf{Cov}^2}{\epsilon^2(1-\gamma)^3}\right)$ *for entropy-regularised objective with* $\lambda = \frac{(1-\gamma)R_{\max}}{1+2\log|\mathcal{A}|}$*.*

*Proof of Part (b).* This proof follows similar steps of the proof for Theorem 4 Part (a), with two additional changes: (i) We have a $\lambda$, i.e. regularisation coefficient, dependent term due to the entropy regulariser. (ii) The maximum value of the soft value function is $\frac{R_{\max}+\lambda\log|\mathcal{A}|}{1-\gamma}$ instead of $\frac{R_{\max}}{1-\gamma}$ for the unregularised value function.

**Step 1:** From Equation (22), we observe that the soft-value function $\tilde{V}_{\boldsymbol{\pi_\theta}}^{\boldsymbol{\pi_\theta}}$ is $L_\lambda$-smooth.

Thus, following the Step 1 of Theorem 4, we get

$$\min_{t\in[T-1]} \|\nabla \tilde{V}_{\boldsymbol{\pi}_{\boldsymbol{\theta}_t}}^{\boldsymbol{\pi}_{\boldsymbol{\theta}_t}}(\boldsymbol{\rho})\|^2 \leq \frac{1}{T\eta\left(1-\frac{L_\lambda\eta}{2}\right)}\left(\tilde{V}_{\boldsymbol{\pi}_o^\star}^{\boldsymbol{\pi}_o^\star}(\boldsymbol{\rho}) - \tilde{V}_{\boldsymbol{\pi}_{\boldsymbol{\theta}_0}}^{\boldsymbol{\pi}_{\boldsymbol{\theta}_0}}(\boldsymbol{\rho})\right)$$

$$\leq \frac{R_{\max} + \lambda\log|\mathcal{A}|}{T\eta\left(1-\frac{L_\lambda\eta}{2}\right)(1-\gamma)}. \tag{52}$$

The last inequality is true due to the fact that $\mathrm{SubOpt}(\boldsymbol{\pi}_{\boldsymbol{\theta}_0}) \leq \tilde{V}_{\boldsymbol{\pi}_o^\star}^{\boldsymbol{\pi}_o^\star}(\boldsymbol{\rho}) \leq \frac{R_{\max}+\lambda\log|\mathcal{A}|}{1-\gamma}$.

**Step 2:** Now, from Part (b) of Lemma 4, we obtain that

$$\min_{t\in[T-1]} \left(\mathrm{SubOpt}(\boldsymbol{\pi}_{\boldsymbol{\theta}_t} \mid \lambda)\right)^2$$

$$\leq \min_{t\in[T-1]} \left(\sqrt{|\mathcal{S}||\mathcal{A}|}\left\|\frac{\boldsymbol{d}_{\boldsymbol{\pi}_{\boldsymbol{\theta}_t},\boldsymbol{\rho}}^{\boldsymbol{\pi}_o^\star}}{\boldsymbol{d}_{\boldsymbol{\pi}_{\boldsymbol{\theta}_t},\boldsymbol{\nu}}^{\boldsymbol{\pi}_{\boldsymbol{\theta}_t}}}\right\|_\infty \|\nabla_{\boldsymbol{\theta}}\tilde{V}_{\boldsymbol{\pi}_{\boldsymbol{\theta}_t}}^{\boldsymbol{\pi}_{\boldsymbol{\theta}_t}}(\boldsymbol{\nu})\|_2 + \frac{\lambda\log|\mathcal{A}|}{1-\gamma}\left(\frac{\gamma\psi_{\max}}{1-\gamma} + 2\right) + \frac{\gamma\psi_{\max}R_{\max}}{(1-\gamma)^2}\right)^2$$

$$\leq 2|\mathcal{S}||\mathcal{A}| \min_{t\in[T-1]} \left\|\frac{\boldsymbol{d}_{\boldsymbol{\pi}_{\boldsymbol{\theta}_t},\boldsymbol{\rho}}^{\boldsymbol{\pi}_o^\star}}{\boldsymbol{d}_{\boldsymbol{\pi}_{\boldsymbol{\theta}_t},\boldsymbol{\nu}}^{\boldsymbol{\pi}_{\boldsymbol{\theta}_t}}}\right\|_\infty^2 \|\nabla_{\boldsymbol{\theta}}\tilde{V}_{\boldsymbol{\pi}_{\boldsymbol{\theta}_t}}^{\boldsymbol{\pi}_{\boldsymbol{\theta}_t}}(\boldsymbol{\nu})\|_2^2$$

$$+ 2\left(\frac{\lambda\log|\mathcal{A}|}{1-\gamma}\left(\frac{\gamma\psi_{\max}}{1-\gamma} + 2\right) + \frac{\gamma\psi_{\max}R_{\max}}{(1-\gamma)^2}\right)^2$$

$$\leq \frac{2|\mathcal{S}||\mathcal{A}|\mathsf{Cov}^2\left(R_{\max} + \lambda \log|\mathcal{A}|\right)}{T\eta\left(1 - \frac{L_\lambda \eta}{2}\right)(1 - \gamma)}$$

$$+ 2\left(\frac{\lambda \log|\mathcal{A}|}{1 - \gamma}\left(\frac{\gamma \psi_{\max}}{1 - \gamma} + 2\right) + \frac{\gamma \psi_{\max} R_{\max}}{(1 - \gamma)^2}\right)^2.$$

The last inequality is due to the upper bound on the minimum gradient norm as in Equation (52) and by definition of the coverage parameter Cov.

Thus, we conclude that

$$\min_{t \in [T-1]} \mathrm{SubOpt}(\boldsymbol{\pi}_{\boldsymbol{\theta}_t} \mid \lambda)$$

$$\leq \sqrt{\frac{2|\mathcal{S}||\mathcal{A}|\mathsf{Cov}^2\left(R_{\max} + \lambda \log|\mathcal{A}|\right)}{T\eta\left(1 - \frac{L_\lambda \eta}{2}\right)(1 - \gamma)}} \tag{53}$$

$$+ \sqrt{2}\left(\frac{\lambda \log|\mathcal{A}|}{1 - \gamma}\left(\frac{\gamma \psi_{\max}}{1 - \gamma} + 2\right) + \frac{\gamma \psi_{\max} R_{\max}}{(1 - \gamma)^2}\right). \tag{54}$$

**Step 4:** Now, by setting the $T$-dependent term in Equation (54) to $\epsilon$, we get $T \geq \frac{2|\mathcal{S}||\mathcal{A}|\mathsf{Cov}^2(R_{\max} + \lambda \log|\mathcal{A}|)}{\eta\left(1 - \frac{L_\lambda \eta}{2}\right)(1 - \gamma)\epsilon^2}$.

Choosing $\eta = \frac{1}{L_\lambda}$, $\lambda = \frac{(1-\gamma)R_{\max}}{(1 + 2\log|\mathcal{A}|)}$, and $\psi_{\max} = \mathcal{O}(\frac{1-\gamma}{\gamma})$, we get the final expression $T \geq \frac{8|\mathcal{S}||\mathcal{A}|\mathsf{Cov}^2 L_\lambda R_{\max}}{(1-\gamma)\epsilon^2}$, and

$$\min_{t \in [T-1]} \mathrm{SubOpt}(\boldsymbol{\theta}_t \mid \lambda) \leq \epsilon + \mathcal{O}\left(\frac{R_{\max}}{1 - \gamma}\right).$$

Finally, noting that $L_\lambda = \mathcal{O}\left(\max\left\{\frac{\gamma R_{\max}|\mathcal{A}|}{(1-\gamma)^2}, \frac{R_{\max}}{\gamma^2(1-\gamma)}\right\}\right)$, we get

$$T = \Omega\left(\frac{|\mathcal{S}||\mathcal{A}|^2 R_{\max}^2 \mathsf{Cov}^2}{\epsilon^2(1 - \gamma)^3}\right).$$

$\square$

# I. Technical Lemmas

**Lemma 11** (Lower Bound of Coverage). *For any $\pi, \pi' \in \Pi(\Theta)$, the following non-trivial lower bound holds,*

$$\left\| \frac{d_{\pi'}}{d_{\pi}} \right\|_{\infty} \geq 1$$

*Proof.*

$$\left\| \frac{d_{\pi'}}{d_{\pi}} \right\|_{\infty} = \max_{s,a} \frac{d_{\pi'}(s,a)}{d_{\pi}(s,a)} \geq \frac{1}{\sum_{s,a} w_{s,a}} \sum_{s,a} \frac{d_{\pi'}(s,a)}{d_{\pi}(s,a)} \cdot w_{s,a}$$

Choose $w_{s,a} = d_{\pi}(s,a)$ Hence, we get,

$$\max_{s,a} \frac{d_{\pi'}(s,a)}{d_{\pi}(s,a)} \geq \frac{\sum_{s,a} d_{\pi'}(s,a)}{\sum_{s,a} d_{\pi}(s,a)} = 1$$

The last equality holds from the fact that the state-action occupancy measure is a distribution over $\mathcal{S} \times \mathcal{A}$. Hence, $\sum_{s,a} d_{\pi'}(s,a) = \sum_{s,a} d_{\pi}(s,a)$ $\qquad \square$

**Lemma 12.** *The discounted state occupancy measure*

$$d_{\pi'}^{\pi}(s|s_0) \triangleq (1-\gamma) \, \mathbb{E}_{\tau \sim \mathbb{P}_{\pi'}^{\pi}} \left[ \sum_{t=0}^{\infty} \gamma^t \mathbb{1}\{s_t = s\} \right]$$

*is a probability mass function over the state-space $\mathcal{S}$.*

*Proof.* For each fixed $s$ the integrand $\sum_{t=0}^{\infty} \gamma^t \mathbb{1}\{s_t = s\} \geq 0$, hence $d_{\pi'}^{\pi}(s|s_0) \geq 0$.

To check normalization, we sum over all states and use Tonelli/Fubini (permitted because the summand is non-negative) to exchange sums and expectation:

$$\sum_{s \in \mathcal{S}} d_{\pi'}^{\pi}(s|s_0) = (1-\gamma) \, \mathbb{E}_{\tau \sim \mathbb{P}_{\pi'}^{\pi}(\cdot|s_0)} \left[ \sum_{t=0}^{\infty} \gamma^t \sum_{s \in \mathcal{S}} \mathbb{1}\{s_t = s\} \right]$$

$$= (1-\gamma) \, \mathbb{E}_{\tau \sim \mathbb{P}_{\pi'}^{\pi}(\cdot|s_0)} \left[ \sum_{t=0}^{\infty} \gamma^t \cdot 1 \right] = (1-\gamma) \sum_{t=0}^{\infty} \gamma^t = 1.$$

Therefore $\rho$ is a probability mass function on $\mathcal{S}$. $\qquad \square$

A similar argument holds for the discounted state-action occupancy measure $d_{\pi'}^{\pi}(s, a|s_0)$ as well.

# J. Experiment Details

**Environment.** We evaluate PePG in the Gridworld test-bed (Mandal et al., 2023), which has become a benchmark in performative RL. This environment consists of a grid where two agents $A_1$ (the principal) and $A_2$ (the follower), jointly control an actor navigating from start positions (S) to the goal (G) while avoiding hazards. The environment dynamics are as follows: Agent $A_1$ proposes a control policy for the actor by selecting one of four directional actions. Agent $A_2$ can either accept this action (not intervene) or override it with its own directional choice. *This creates a performative environment for $A_1$, as its effective policy outcomes depend on $A_2$'s responses to its deployed strategy.*

The cost structure follows: visiting blank cells (S) incurs penalty of $-0.01$, goal cells (F) cost $-0.02$, hazard cells (H) impose a severe penalty of $-0.5$, and any intervention by $A_2$ results in an additional cost of $-0.05$ for the intervening agent. The response model also follows that of (Mandal et al., 2023), i.e., the agent $A_2$ responds to $A_1$'s policy using a Boltzmann softmax operator. Given $A_1$'s current policy $\boldsymbol{\pi}_1$, we compute the optimal Q-function $Q^{*|\boldsymbol{\pi}_1}$ for each follower agent $A_j$ relative to a perturbed version of the grid world, where each cell types matches $A_1$'s environment with probability $0.7$. We then define an average Q-function over the follower agents and determine the collective response policy via Boltzmann softmax $Q^{*|\pi_1}(s,a) = \frac{1}{n}\sum_{j=2}^{n+1} Q_j^{*|\pi_1}(s,a), \pi_2(a|s) = \frac{\exp(\beta \cdot Q^{*|\pi_1}(s,a))}{\sum_{a'}\exp(\beta \cdot Q^{*|\pi_1}(s,a'))}$.

Note that our experimental setup deliberately uses the immediate response model from the original performative RL framework, rather than the gradually shifting environment of (Rank et al., 2024) that assumes slow shifts in the environment. Our choice to use the immediate response model presents a more challenging performative setting where the environment responds instantaneously to policy changes. This allows us to demonstrate that unlike MDRR (Rank et al., 2024), PePG can handle the fundamental performative challenge without requiring environmental assumptions that artificially slows down the feedback loop, thereby highlighting the robustness of the proposed PePG approach.

**Experimental Setup.** We evaluate PePG (with and without entropy regularisation) alongside Mixed Delayed Repeated Retraining (MDRR), which represents the current state-of-the-art in performative reinforcement learning under gradually shifting environments (Rank et al., 2024), and Repeated Policy Optimization with Finite Samples (RPO FS). MDRR has demonstrated significant improvements over traditional repeated retraining methods, by leveraging historical data from multiple deployments, while RPO FS is included as the baseline method from (Mandal et al., 2023) for direct comparison with the original performative RL approach.

All experiments use a $8 \times 8$ grid with $\gamma = 0.9$, exploration parameter $\epsilon = 0.5$ for initial policy construction, one follower agent $A_2$, and 100 trajectory samples per iteration. The algorithms share common parameters of $T = 1000$ iterations. For regularization, RPO FS and MDRR use $\lambda = 0.1$ from their original experiments, while entropy-regularized PePG uses $\lambda = 2.0$ (ablation studies for this choice are provided in the appendix). PePG uses learning rate $\eta = 0.1$, MDRR employs memory weight $v = 1.1$ for historical data utilization, delayed round parameter $k = 3$, and FTRL parameters $N = B = 10$, while RPO FS follows the finite-sample optimization from (Mandal et al., 2023).

**Computational Cost.** We report the wall-clock runtime of each algorithm on a single CPU. Table 3 summarizes the per-iteration cost and the total runtime extrapolated to 200 iterations based on measurements over 50 iterations.

*Table 3.* Runtime comparison of algorithms (single CPU).

| Algorithm | ms/iter | Total (200 iter) |
|---|---|---|
| PePG (Regularized) | 10,635 | $\sim$35 min |
| PePG | 11,664 | $\sim$39 min |
| RPO FS (Mandal et al., 2023) | 13,039 | $\sim$43 min |
| MDRR (Rank et al., 2024) | 24,317 | $\sim$81 min |

All algorithms were run on CPU with identical environment configurations. PePG (with and without regularization) is the most efficient, with per-iteration costs roughly half that of MDRR. RPO sits between the two groups.

# K. Ablation Studies

## K.1. Entropy regularisation

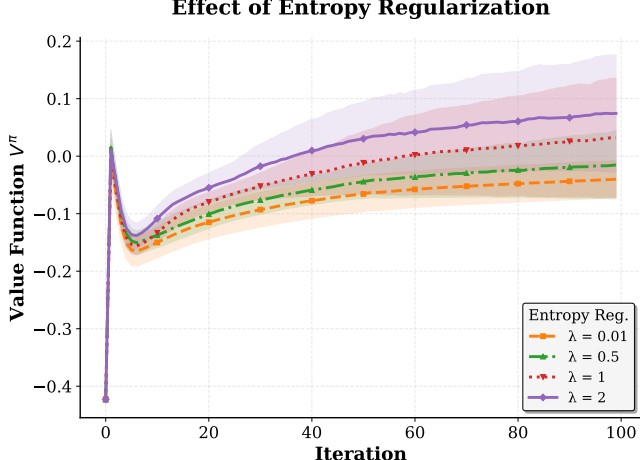

*Figure 4.* Ablation study for PePG for different values of regularised $\lambda$ with 20 random seeds, each for 100 iterations

We conducted an ablation study across four entropy regularization strengths ($\lambda \in \{0.01, 0.5, 1, 2\}$ to determine the optimal balance between exploration and convergence stability in entropy regularised PePG. The results demonstrate that $\lambda = 2$ achieves the highest final performance ( 0.05), while smaller values ($\lambda \leq 1$) converge to similar suboptimal levels around $-0.01$ to $0$, indicating that stronger entropy regularization enables more effective exploration of the policy space in performative settings.

## K.2. Learning rate

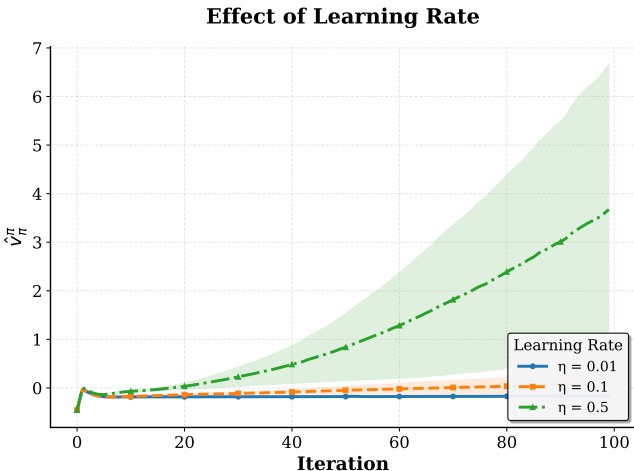

*Figure 5.* Ablation study for PePG for different values of $\eta$ with 20 random seeds across 100 iterations

We additionally performed an ablation study on the learning rate, considering $\eta \in \{0.01, 0.1, 0.5\}$, to examine its effect on convergence behavior and performance stability in PePG. The results indicate that the largest learning rate ($\eta = 0.5$) attains the highest final performance ($\hat{V}_{\pi}^{\pi} = 4$); however, it is also accompanied by substantially higher variance across runs. In contrast, smaller learning rates yield more stable learning dynamics but converge to comparatively lower performance levels. This highlights the classical trade-off between fast convergence and stability in policy optimization.

