# Performative Policy Gradient:
# Optimality in Performative Reinforcement Learning

## Abstract

Post-deployment machine learning algorithms often influence the environments that they act in, and thus, *performatively shift* the underlying dynamics that the standard Reinforcement Learning (RL) ignores. While designing optimal algorithms in this *performative* setting has been studied in supervised learning, the RL counterpart remains under-explored. In this paper, we prove the performative counterparts of the performance difference lemma and the policy gradient theorem in RL, and introduce the **Performative Policy Gradient** algorithm (PePG). PePG is the first policy gradient algorithm designed to account for performativity in RL. Under softmax parametrisation, and also with and without entropy regularisation, we prove that PePG converges to *performatively optimal policies*, i.e. policies that remain optimal under the distribution shifts induced by themselves. Thus, PePG significantly extends the prior works in Performative RL that achieves *performative stability* but not optimality. Our empirical analysis on standard performative RL environments validate that PePG outperforms the existing performative RL algorithms aiming for stability.

## 1. Introduction

Reinforcement Learning (RL) studies the dynamic decision making problems under incomplete information (Sutton & Barto, 1998). Since an RL algorithm tries and optimises a utility function over a sequence of interactions with an unknown environment, RL has emerged as a powerful tool for algorithmic decision making. Specially, in the last decade, RL has underpinned some of the celebrated successes of AI, such as championing Go with AlphaGo (Silver et al., 2014), aligning Large Language Models (LLMs) (Bai et al., 2022), reasoning (Havrilla et al., 2024) etc. The classical paradigm

[1]Anonymous Institution, Anonymous City, Anonymous Region, Anonymous Country. Correspondence to: Anonymous Author <anon.email@domain.com>.

Preliminary work. Under review by the International Conference on Machine Learning (ICML). Do not distribute.

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

^{\boldsymbol{\pi_o^\star}}_{\boldsymbol{\pi_\theta}}(s,a|\boldsymbol{\rho}) \leq 1$, for rewards, we get

$$
\left| \mathop{\mathbb{E}}_{(s,a)\sim d^{\boldsymbol{\pi_o^\star}}_{\boldsymbol{\pi_\theta},\boldsymbol{\rho}}} \left[ r_{\boldsymbol{\pi_o^\star}}(s,a) - r_{\boldsymbol{\pi_\theta}}(s,a) \right] \right| \leq \mathop{\mathbb{E}}_{(s,a)\sim d^{\boldsymbol{\pi_o^\star}}_{\boldsymbol{\pi_\theta},\boldsymbol{\rho}}} \left| r_{\boldsymbol{\pi_o^\star}}(s,a) - r_{\boldsymbol{\pi_\theta}}(s,a) \right| \leq \| r_{\boldsymbol{\pi_o^\star}} - r_{\boldsymbol{\pi_\theta}} \|_1
$$

Similarly for transitions, we get

$$
\left| \mathop{\mathbb{E}}_{(s,a)\sim d^{\boldsymbol{\pi_o^\star}}_{\boldsymbol{\pi_\theta},\boldsymbol{\rho}}} \left[ (\mathbf{P}_{\boldsymbol{\pi_o^\star}} - \mathbf{P}_{\boldsymbol{\pi_\theta}})^\top V^{\boldsymbol{\pi}}_{\boldsymbol{\pi}} \right] \right| \leq \mathop{\mathbb{E}}_{(s,a)\sim d^{\boldsymbol{\pi_o^\star}}_{\boldsymbol{\pi_\theta},\boldsymbol{\rho}}} \left| (\mathbf{P}_{\boldsymbol{\pi_o^\star}} - \mathbf{P}_{\boldsymbol{\pi_\theta}})^\top V^{\boldsymbol{\pi}}_{\boldsymbol{\pi}} \right|
$$

$$
\overset{(a)}{\leq} \mathop{\mathbb{E}}_{(s,a)\sim d^{\boldsymbol{\pi_o^\star}}_{\boldsymbol{\pi_\theta},\boldsymbol{\rho}}} \left[ \| \mathbf{P}_{\boldsymbol{\pi_o^\star}} - \mathbf{P}_{\boldsymbol{\pi_\theta}} \|_1 \cdot \| V^{\boldsymbol{\pi_o^\star}}_{\boldsymbol{\pi_o^\star}} \|_\infty \right]
$$

$$
= \| \mathbf{P}_{\boldsymbol{\pi_o^\star}} - \mathbf{P}_{\boldsymbol{\pi_\theta}} \|_1 \cdot \| V^{\boldsymbol{\pi_o^\star}}_{\boldsymbol{\pi_o^\star}} \|_\infty ,
$$

(a) holds due to Hölder's inequality.

Now, leveraging the triangle inequality and Lipschitzness assumption on reward and transitions, we further get

$$
\left| \mathop{\mathbb{E}}_{(s,a)\sim d^{\boldsymbol{\pi_o^\star}}_{\boldsymbol{\pi_\theta},\boldsymbol{\rho}}} \left[ r_{\boldsymbol{\pi_o^\star}}(s,a) - r_{\boldsymbol{\pi_\theta}}(s,a) + \gamma(\mathbf{P}_{\boldsymbol{\pi_o^\star}} - \mathbf{P}_{\boldsymbol{\pi_\theta}})^\top V^{\boldsymbol{\pi}}_{\boldsymbol{\pi}} \right] \right| \leq L_r \| \boldsymbol{\pi_o^\star} - \boldsymbol{\pi_\theta} \|_\infty + \gamma L_{\mathbf{P}} \left\| V^{\boldsymbol{\pi_o^\star}}_{\boldsymbol{\pi_o^\star}} \right\|_\infty \| \boldsymbol{\pi_o^\star} - \boldsymbol{\pi_\theta} \|_\infty
$$

Finally, due to Assumption 1, we get $\left\| \mathbf{V}^{\boldsymbol{\pi_o^\star}}_{\boldsymbol{\pi_o^\star}} \right\|_\infty \leq \frac{R_{\max}}{1-\gamma}$, and thus,

$$
\left| \mathop{\mathbb{E}}_{(s,a)\sim d^{\boldsymbol{\pi_o^\star}}_{\boldsymbol{\pi_\theta},\boldsymbol{\rho}}} \left[ r_{\boldsymbol{\pi_o^\star}}(s,a) - r_{\boldsymbol{\pi_\theta}}(s,a) + \gamma(\mathbf{P}_{\boldsymbol{\pi_o^\star}} - \mathbf{P}_{\boldsymbol{\pi_\theta}})^\top V^{\boldsymbol{\pi_o^\star}}_{\boldsymbol{\pi_o^\star}} \right] \right| \leq L_r \| \boldsymbol{\pi_o^\star} - \boldsymbol{\pi_\theta} \|_\infty + \frac{\gamma}{1-\gamma} L_{\mathbf{P}} R_{\max} \| \boldsymbol{\pi_o^\star} - \boldsymbol{\pi_\theta} \|_\infty
$$

**Step 3:** We know $\| \boldsymbol{\pi_o^\star} - \boldsymbol{\pi_\theta} \|_\infty \leq \| \boldsymbol{\pi_o^\star} - \boldsymbol{\pi_\theta} \|_1 = 2\text{TV}\,(\boldsymbol{\pi_o^\star} \| \boldsymbol{\pi_\theta}) \leq 2\sqrt{2} D_{\text{H}}\,(\boldsymbol{\pi_o^\star} \| \boldsymbol{\pi_\theta})$. Thus,

$$

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

$$+ \frac{\lambda}{1-\gamma} \pi_\theta(a|s) \sum_a d^{\pi_\theta}_{\pi_\theta, \nu}(s,a) \log \pi_\theta(a|s)$$

$$\underset{(c)}{\geq} \frac{1}{1-\gamma} d^{\pi_\theta}_{\pi_\theta, \nu}(s,a) \tilde{A}^{\pi_\theta}_{\pi_\theta}(s,a) - \frac{1}{(1-\gamma)^2} d^{\pi_\theta}_{\pi_\theta, \nu}(s,a)(R_{\max} + \lambda \log |\mathcal{A}|) L_{\mathbf{P}} - \frac{1}{1-\gamma} L_r d^{\pi_\theta}_{\pi_\theta, \nu}(s,a)$$

$$- \frac{\lambda}{1-\gamma} d^{\pi_\theta}_{\pi_\theta, \nu}(s,a) \log |\mathcal{A}|$$

(a) holds due to Lipchitzness of rewards, transitions and also for the following:

$$\underset{\tau \sim \mathbb{P}^{\pi_\theta}_{\pi_\theta}}{\mathbb{E}} \left[ \sum_{t=0}^{\infty} \gamma^t \frac{\partial}{\partial \theta_{s,a}} \log \pi_\theta(a_t|s_t) \right] = \underset{\tau \sim \mathbb{P}^{\pi_\theta}_{\pi_\theta}}{\mathbb{E}} \left[ \sum_{t=0}^{\infty} \gamma^t \mathbb{1}[s_t = s, a_t = a] \right] - \underset{\tau \sim \mathbb{P}^{\pi_\theta}_{\pi_\theta}}{\mathbb{E}} \left[ \sum_{t=0}^{\infty} \gamma^t \pi_\theta(a|s) \mathbb{1}[s_t = s] \right]$$

$$= d^{\pi_\theta}_{\pi_\theta, \rho}(s,a) - d^{\pi_\theta}_{\pi_\theta, \rho}(s) \pi_\theta(a|s) = 0$$

(b) holds because:

$$\underset{\tau \sim \mathbb{P}^{\pi_\theta}_{\pi_\theta, \nu}}{\mathbb{E}} \left[ \sum_{t=0}^{\infty} \gamma^t \pi_\theta(a|s) \mathbb{1}[s_t = s] \tilde{A}^{\pi_\theta}_{\pi_\theta}(s_t, a_t) \right] = \underset{\tau \sim \mathbb{P}^{\pi_\theta}_{\pi_\theta, \nu}}{\mathbb{E}} \left[ \sum_{t=0}^{\infty} \gamma^t \pi_\theta(a|s) \mathbb{1}[s_t = s] A^{\pi_\theta}_{\pi_\theta}(s_t, a_t) \right]$$

$$- \underset{\tau \sim \mathbb{P}^{\pi_\theta}_{\pi_\theta, \nu}}{\mathbb{E}} \left[ \sum_{t=0}^{\infty} \gamma^t \pi_\theta(a|s) \log \pi_\theta(a_t|s_t) \mathbb{1}[s_t = s] \right]$$

$$= -\pi_\theta(a|s) \left[ \sum_{t=0}^{\infty} \gamma^t \log \pi_\theta(a_t|s_t) \mathbb{1}[s_t = s] \right]$$

And (c) holds since,

$$-\sum_a d^{\pi_\theta}_{\pi_\theta}(s,a|\nu) \log \pi_\theta(a|s) = d^{\pi_\theta}_{\pi_\theta}(s|\nu) \left( -\sum_a \pi_\theta(a|s) \log \

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

*The gradient ascent algorithm on* $V_{\pi_\theta}^{\pi_\theta}(\rho)$ *(Equation (5)) with step size* $\eta = \Omega(\min\{\frac{(1-\gamma)^2}{\gamma|\mathcal{A}|}, \frac{(1-\gamma)^3}{\gamma^2}\})$ *satisfies, for all distributions* $\rho \in \Delta(\mathcal{S})$. *Then, for entropy regularised case, if we set* $\lambda = \frac{(1-\gamma)R_{\max}}{1+2\log|\mathcal{A}|}$, *we get*

$$\min_{t<T}\text{SubOpt}(\pi_{\theta_t} \mid \lambda) \le \epsilon + \mathcal{O}\left(\frac{1}{1-\gamma}\right) \text{ when } T = \Omega\left(\frac{R_{\max}|\mathcal{S}||\mathcal{A}|^2}{\epsilon^2(1-\gamma)^3}\text{Cov}^2\right),$$