# OpenReview forum: "Performative Policy Gradient: Optimality in Performative Reinforcement Learning"
_ICML.cc/2026/Conference — ICML 2026 regular_

### Official Review · Reviewer_M8GZ · 2026-03-10

**Soundness:** 3
**Presentation:** 3
**Significance:** 3
**Originality:** 2
**Overall Recommendation:** 4
**Confidence:** 4

**Summary:**

The authors analyze a policy gradient algorithm in the performative reinforcement learning (PRL) and entropy regularized PRL (ERPRL) settings under the smoothness, sensitivity and bounded reward assumptions. They extend the performance difference lemma, policy gradient theorem and gradient domination results to the performative setting. They prove the best-iterate convergence of their proposed algorithm. They also discuss the exponential family performative MDPs and show an improved gradient domination result. The authors also provide experimental results, comparing their algorithm with the baselines.

**Compliance With Llm Reviewing Policy:**

Affirmed.

**Final Justification:**

The rebuttal resolves my main concerns.

**Key Questions For Authors:**

1. Could you discuss the smoothness assumption? How restrictive is it? Could you provide some cases where this assumption fail to model? Or, if it could model (almost) all realistic use cases, could you elaborate why?
2. I disagree with the claim that "PePG is the first policy gradient algorithm designed to account for performativity in RL" as [1] shows the convergence of the standard projected gradient ascent and natural policy gradient algorithms to performative stability in a more general setting that contains PRL. The main difference between two algorithms appears to be the gradient expression: [1] ignores the performative effects in gradient calculation, whereas the current work takes the performativity into account. As far as I understand, this leads to convergence to a performatively optimal point instead of a performatively stable one. The current paper also improves the bounds in [1] in the narrower scope of PRL. I think authors should discuss the differences between these two works in the paper as it would clarify the points I just mentioned. I would also appreciate the author's correction if my reasoning is flawed here. I think this discussion would improve the paper as it clarifies some subtle but important aspects of the problem setting.
3. I am not sure if the observation "Existence of such irreducible bias at convergence for both stability and optimality seeking algorithms indicate that this might be inherent to performative RL" (line 366) is correct. Theorem 4 in [1] shows that one can avoid this additional term by switching to a natural gradient algorithm. Could you please elaborate on your comment?
4. Could you please elaborate on the gradient estimation? Estimation of the gradient of transition kernel and the reward function w.r.t. the parameters seems challenging. How does one estimate the gradient from trajectories in a general PeMDP? How do you estimate the gradients in your experiments?

I am willing to increase my score if the authors could provide the additional discussion to address these concerns.

[1] Sahitaj, R., Sasnauskas, P., Yalın, Y., Mandal, D. and Radanovic, G., 2025, April. Independent Learning in Performative Markov Potential Games. In _International Conference on Artificial Intelligence and Statistics_ (pp. 3304-3312). PMLR.

**Limitations:**

Yes.

**Strengths And Weaknesses:**

1. The paper is easy-to-follow. A minor suggestion would be to add direct references to the related proofs in the appendix for each result. The discussions after the results are helpful. I don't like the notation "Cov" for distribution mismatch coefficient as covariance is the first thing it brings to my mind. Is there a reason why the authors avoided the standard RL notations for the mismatch coefficient?
2. The bounded reward and sensitivity assumptions are standard in the PRL literature. As far as I know the smoothness assumption is not common in PRL. This could be fine but I think some discussion is required.
3. The algorithm is a standard policy gradient algorithm. The analysis looks sound but does not appear to provide any novelty. Further comments are in the questions section (Q2).
4. The sensitivity and smoothness constants are hidden in the theorem statements. I think the dependence on these constants are important as they quantify the performative effects in the environment. I think it would improve the paper to include them explicitly in the statements.
5. The experimental analysis are sufficient. The authors make interesting observations based on the experimental results.

---

> ### Author Rebuttal · Authors · 2026-03-31
>
> We thank the reviewer for their valuable time and effort to carefully assess our work. We respond to the weaknesses highlighted and the questions raised.
>
> **Q1: Smoothness.** Smoothness of transitions and rewards is needed to prove smoothness of performative value function (Lemma 5-6). Classical RL requires the value function to be smooth in the policy parameterisation to convergence of PG [1]. Similarly in PRL, we require smoothness of log-policy, log-transitions, and rewards to prove convergence of PePG. There are similar assumptions on smoothness of objective functions with respect to policy/decision parameters in performative learning [2,3,4] and decision-dependent optimization [5]. Furthermore, we demonstrate that this assumption is consistent with a large class of PeMDPs by studying the exponential family of transitions and linear rewards as representative of this smooth class of transitions and rewards. Now, we further include a discussion on the PeMDPs where smoothness does not hold (e.g., Bernoulli rewards and piecewise transitions).
>
> **Q2: ''PePG is...'' in abstract.**
>
> (a) [4] studies the effect of performativity for MARL, specifically Markov potential games. They propose gradient-based algorithms showing convergence to approximate performative stability, but not optimality. We concur to reviewer's remark, and we change the phrasing, also add a discussion in the related work elaborating on the connection to [4] though the problem setting and objective are different. Their analysis hinges on additional assumptions (e.g., Assumption 2, $c$) making the reduction not immediate.
>
> (b) Performative Regret of [4] differs from our definition of sub-optimality. Performative Regret is equivalent to our performance difference term in Equation (12) for a single agent system. *This term measures the change in value function for moving from one policy to another when the environment inducing policy for both of them are same.* This choice of sub-optimality or regret is sufficient when optimising for an $\epsilon$-PSE (or $\epsilon$-stable policy in single-agent RL).
>
> In contrast, for optimality, we need to also account for the performative shift term consisting the reward and transition shift terms (consequently $L_P$ and $L_r$). This yields the PL-condition as well as the convergence guarantee (with multiplicative factors of $\gamma$).
>
> Moreover, in the regularised regime, if we tune $\lambda$ with $\epsilon$ like Theorem 5 in [4], we can get an exact $\epsilon$-optimality. (Refer to our response to reviewer 6AgF)
>
> **Q3: Line 366.** We refer to **Q2(b)** that sufficiently address this concern.
>
> **Q4: Gradient estimation in a general PeMDP.** The performative gradient (Eq. (8)) requires two terms: $\nabla_\theta r_{\pi_\theta}(s,a)$ and $\nabla_\theta \log P_{\pi_\theta}(s'|s,a)$ in addition to gradient of log-policy. Computing the first two gradients depend on what we know about the environment's response to $\theta$.
>
> - *1: Known parametric form.*  For exponential family PeMDPs (Section 4), the transition has an explicit form $P_{\pi_\theta}(s'|s,a) = \exp(\theta_{s,a}\psi(s') - \log Z(\theta))$ and rewards are linear in $\theta$. Both gradients can be computed in closed form, and Equation (8) is exact.
>
> - *2: Unknown parametric form (Our method).* When the closed form is unavailable, we learn differentiable approximations: a reward model $f_r: \theta \mapsto \hat r_{\pi_{\theta}}$ and a transition model $f_p: \theta \mapsto \hat P_{\pi_\theta}$ via neural networks. The gradient terms are obtained by back-propagating through $f_r$ and $f_p$. Both the models of rewards and transitions are retrained at every iteration on a sliding window of the last 50 policy-trajectory pairs. As the environment shifts with the policy, this online retraining keeps the gradient estimates accurate. A brief warm-up provides a seed dataset, while accuracy comes from the ongoing updates. The slight overhead reported in Section 5 reflects this cost.
>
> We add this discussion in the experimental analysis (Section 5) of the final draft.
>
> **Notations.** (i) We update the notation of $Cov$ to $D_{\infty}$ to better align with the RL theory literature [1] and avoid confusion. (ii) We avoided constants ($L_P,L_r$, $R_2 , T_2$) in theorem statements of the main paper for brevity and readability, and retain them only in the detailed proofs. As suggested, we include them in the main body.
>
> We hope that we clarify all the raised concerns/questions. We are eager to discuss if there are more questions or comments.
>
> *Reference:*
>
> [1] On the theory of policy gradient methods: Optimality, approximation, and distribution shift. JMLR 2021.
>
> [2] Performative prediction. ICML 2020.
>
> [3] How to learn when data reacts to your model: performative gradient descent. ICML 2021.
>
> [4] Independent Learning in Performative Markov Potential Games. AISTATS 2025.
>
> [5] Decision-dependent stochastic optimization: The role of distribution dynamics, 2025.

---

> > ### Author Rebuttal · Reviewer_M8GZ · 2026-04-03
> >
> > I thank the authors for their response. Most of my concerns are resolved. My main concern is Q3 as I detail below, and I will increase my score if this is resolved.
> >
> > Q1: I think the smoothness assumption is much stronger assumption than smoothness (w.r.t. the policy parameters) in standard RL. In standard RL, the smoothness could be obtained by proper policy parameterization. In performative RL, this differentiability assumption restricts the environments that can be modeled by the framework. That said, I understand the smoothness (or a similar) assumption is necessary for the analysis in the paper. I think it is a reasonable assumption; a discussion on this assumption should suffice.
> >
> > Q2: The rephrasing and discussion should suffice.
> >
> > Q3: The authors refer to the answer Q2 (b), which highlights the difference between optimality- and stability-seeking algorithms. I do not think this would justify the strong claim made in the paper: "Existence of such irreducible bias at convergence for both stability and optimality seeking algorithms indicate that this might be inherent to performative RL." The claim is both for stability- and optimality- seeking algorithms and [4] shows the bias term could be avoided by natural gradient algorithms for stability seeking algorithms. For optimality seeking algorithms, while I am not aware of any published results suggesting otherwise, I do not see a reason why the natural gradient algorithms could not avoid the bias term. I do not think such speculative claims are necessary.
> >
> > Q4: The additional discussion should suffice.

---

> > > ### Author Response · Authors · 2026-04-05
> > >
> > > We thank the reviewer for considering our rebuttal. Here, we address the specific pointers raised.
> > >
> > > - **Q1:** We agree and we add a detailed discussion related to the smoothness assumptions, its relation to existing literature, and the restriction it poses on the environment in our updated manuscript.
> > > - **Q2 and Q4:** We also add discussions regarding both these points as we have stated in our rebuttal.
> > > - **Q3:** We acknowledge the reviewer’s concern regarding the statement “such biases…”. Since using INPG [1] helps to avoid the bias in the case of stability-seeking algorithms for Markov games, we have revised our statement in the updated manuscript and cited [1] for justification. For optimality-seeking algorithms, while we have provided a specific hard instance (ref. W3 in Reviewer JX81’s rebuttal) where the sub-optimality gap is lower bounded by $O(\frac{1}{1-\gamma})$, obtaining a rigorous lower bound remains an open question. We highlight this as a direction for future research. In addition, we also posit extension of Natural PG algorithms for achieving performartive optimality and also possibly reducing the performative-shift induced bias term as a future research direction.
> > >
> > > We hope that this response addresses your concerns. Let us know if you have any further questions or comments.
> > >
> > > *References*
> > >
> > > [1] Independent Learning in Performative Markov Potential Games. AISTATS 2025.

---

### Official Review · Reviewer_6AgF · 2026-03-12

**Soundness:** 3
**Presentation:** 4
**Significance:** 2
**Originality:** 3
**Overall Recommendation:** 4
**Confidence:** 3

**Summary:**

This paper proposes a policy optimization framework in performative RL. Unlike standard RL, performative RL studies the setting in which a deployed policy affects the environment dynamics. In this setting, two types of policies exist: performative stable policy and performative optimal policy. The authors prove the performative policy gradient theorem and show the conditions under which PG algorithms provably converge to the performative optimal policy.

**Compliance With Llm Reviewing Policy:**

Affirmed.

**Key Questions For Authors:**

Can performative RL be applied to more sophisticated problems other than the gridworld in (Mandal 2023)?

**Limitations:**

yes

**Strengths And Weaknesses:**

Strengths:

This is a beautifully written paper with everything from motivation, prior work, background to contributions clearly presented. The authors build a counterpart of policy gradient theorem in performative RL and accompanies that with convergence guarantees. Overall I do not see major problems with the paper.

Weaknesses:

My main concerns are:
-  This paper and the two predecessors (Mandal et al. 2023 and Rank et al., 2024) all use the same gridwork environment. This begs the question of whether performative RL can be applied to meaningful and practical problems. The example in the introduction is nice, but it is not really an RL problem.
- the proofs make use of standard tools. As a result the bounds e.g. lemma 3 depends on terms like $\frac{\text{Cov}}{(1-\gamma)^2} L_P R_\text{max}$. A bound like this seems overly pessimistic and I am not sure it is any different than the existing results.

---

> ### Author Rebuttal · Authors · 2026-03-31
>
> We thank the reviewer for the positive evaluation and appreciating the overall presentation of our work. We respond to the raised concerns below.
>
> **Q1 and W1: Sophisticated test-bed for PRL algorithms.**
> The grid-world setting considered by [1] has been the gold standard of testing PRL algorithms. We believe it is imperative to for the RL community to set proper (new) benchmarks, simulators that extends the scope of empirical evaluation for PRL algorithms. Note, there are sequential decision making simulation environments built on well known datasets in standard performative learning literature ([2,3,4]) that can be modeled through PeMDPs.
>
> Although constructing benchmarks or simulators in this setting should be considered as a separate future work by itself as mentioned in Section 6. Out of many interesting real-life problems from College admission [7] to maintenance of industrial machinery [8], we add one specific instance in details:
>
> - *Bank Lending Problem.* Consider a lending institution that sequentially gives loans to applicants drawn from a population. The state space $\mathcal{S}$ encodes the joint distribution of applicant features such as credit score, income, employment history etc. The action space $\mathcal{A}$ consists of lending decisions: approve or reject. The transition dynamics $P_{\pi}(s' \mid s, a)$ describes how the population's financial profile evolves over time as a function of past decisions. A lending policy $\pi$ designed to maximise profit of the bank, itself reshapes the distribution of future applicants, shifting the population distribution $\mathcal{D}(\pi)$ in a way that depends non-trivially on $\pi$ ([6]). More subtly, widespread loan access changes actual creditworthiness at the population level by enabling wealth accumulation, which feeds back into future states. The reward signal $r_{\pi}(s,a)$ captures the trade-off between portfolio profitability and default risk (the applicant pays back the loan or not). A performative RL agent must account for the fact that optimizing this reward under a fixed distributional assumption will systematically misestimate future default rates, since the population it will face at deployment is not the population it trained on.
>
>
>
> **W2: Theoretical improvements.** The underlying algorithm design chosen for PePG follows standard policy gradient mechanism. Thus, on a skeletal level, the theoretical performance analysis of PePG must follow standard PG. The technical challenge lies in adapting the analysis to performativity. We show that considering shifts in transitions and rewards gives us novel Lemmas on performance difference, gradient domination, and finally convergence guaranties.
>
> We respectfully disagree with the remark of theoretical guaranties being pessimistic and vacuous. As pointed out by other reviewers, in Table 1 we observe PePG gains $\mathcal{O}\left(\frac{|A|}{\epsilon^2 (1-\gamma)^4}\right)$ in suboptimality compared to existing baselines. Moreover, in the regularised regime, if we tune $\lambda,\eta$ with $\epsilon$ like Theorem 5 in [4] and set it as $\lambda = \mathcal{O}\left(\frac{\epsilon(1-\gamma)^2}{ \log |\mathcal{A}|\mathrm{Cov}}\right)$ with $\eta = \frac{1-\gamma}{L_{\lambda}}$, we can get an exact $\epsilon$-optimality. We consciously avoided this choice to keep $\lambda$ independent of $\epsilon$ as practitioners do not tune parameters with a fixed suboptimality gap $\epsilon$ in mind. Additionally, we choose $\lambda = \mathcal{O}\left(\frac{1}{\log A}\right)$, which is significantly smaller and more realistic compared to existing PRL literature for achieving performative stability ([1,5]) that tend to set the value of $\lambda$ unrealistically high and depend on precision $\epsilon$. We would like to refer the reviewer towards discussions below Theorem 3 and Lemma 4 for more details on the improvements achieved in this work.
>
> We hope that we have been able to sufficiently address all the concerns raised in the review. Please let us know if there are any further questions or suggestions.
>
> *References:*
>
> [1] Mandal, D., Triantafyllou, S. and Radanovic, G., 2023, July. Performative reinforcement learning. ICML.
>
> [2] Fair Isaac Corporation. FICO Explainable Machine Learning Challenge Dataset. Fair Isaac Corporation, 2018.
>
> [3] Becker, Barry, and Ronny Kohavi. Adult Dataset. UCI Machine Learning Repository, 1996.
>
> [4] Hofmann, Hans. Statlog (German Credit Data). UCI Machine Learning Repository, 1994.
>
> [5] Rank, Ben, et al. "Performative reinforcement learning in gradually shifting environments." 2024.
>
> [6] Perdomo, J., Zrnic, T., Mendler-Dünner, C. and Hardt, M., 2020, November. Performative prediction. ICML.
>
> [7] Kleine Buening, Thomas, et al. "On meritocracy in optimal set selection." EAAMO, 2022.
>
> [8] Jardine, Andrew KS, Daming Lin, and Dragan Banjevic. "A review on machinery diagnostics and prognostics implementing condition-based maintenance." MSSP (2006).

---

> > ### Author Rebuttal · Reviewer_6AgF · 2026-04-03
> >
> > I am not very sure how the theoretical contribution is different from the standard DP result except on performative PG, for example we can see similar results in [1,2]. Moreover, I am not entirely satisfied with the experiment: a pertinent, medium-scale problem could further strengthen the paper. But overall, I guess these issues do not overshadow the contributions, and I remain positive towards the paper.
> >
> > [1] Finite-Time Bounds for Fitted Value Iteration, JMLR 2008\
> > [2] Approximate Modified Policy Iteration and its Application to the Game of Tetris, JMLR 2015

---

### Official Review · Reviewer_JX81 · 2026-03-15

**Soundness:** 3
**Presentation:** 4
**Significance:** 3
**Originality:** 3
**Overall Recommendation:** 4
**Confidence:** 4

**Summary:**

This paper studies performative reinforcement learning (RL), where the deployed policy induces shifts in the underlying MDP's transition and reward functions. The authors derive performative counterparts of two classical results — the performance difference lemma and the policy gradient theorem — and use these to design the Performative Policy Gradient (PePG) algorithm. PePG is claimed to be the first policy gradient algorithm provably converging to a performatively optimal policy (rather than merely a stable one) in tabular PeMDPs. Convergence is established under smoothness assumptions on the environment, with explicit rates for softmax policies and exponential family PeMDPs. Empirical comparisons against MDRR and RPO-FS on a gridworld environment support the theoretical claims.

**Compliance With Llm Reviewing Policy:**

Affirmed.

**Final Justification:**

I read the author's response and the discussion. The response resolves my questions and I remain positive about this paper.

**Key Questions For Authors:**

- the discussion in after Lemma 2 is unclear to me. "This implies that the
gap between the optimal performative value function and that of any stability-seeking algorithm is O  (1 − γ)−1." How is that so? By Lemma 2, the suboptimality is bounded by 2 terms -- the advantage value and the Hellinger distance. How does that translate into O(1-\gamma)^{-1} bound on suboptimality? The paragraph right after this, that discusses the behaviorial difference between an optimality-seeking algorithm and a stability-seeking algorithm, is also unclear to me. How does Lemma enable such interpretation exactly?
- Why do you consider different initial state distributions \rho and \nu? (e.g., in the definition of Cov in Theorem 3). shouldn't they be the same?

**Limitations:**

Yes

**Strengths And Weaknesses:**

Strengths:

- Mandal et al. (2023) explicitly posed the design of performative policy gradient algorithms as an open problem. This paper provides an affirmative answer for discrete state-action spaces, which is a meaningful contribution to the performative RL literature.
- Novel theoretical machinery. The performative policy gradient theorem (Theorem 2) is a non-trivial extension of the classical result, correctly identifying two additional gradient terms arising from environment shifts — the expected gradient of reward and the expected gradient of log-transition probabilities weighted by their impact on cumulative return. The derivation appears technically sound and the three-step convergence framework (gradient domination → smoothness → iterative improvement) is clean and modular.
-  The paper makes a useful conceptual contribution by formally and empirically distinguishing optimality-seeking from stability-seeking algorithms. The suboptimality gap comparison in Table 1, showing O(|S||A|²/ε²(1−γ)³) for PePG versus O(|A|²|S|³/ε⁴(1−γ)⁶) for prior work, is a concrete and informative comparison.
- Empirical results are consistent with theory. Figure 2 clearly shows that PePG achieves higher value function performance than stability-seeking baselines, and the observation that stability-seeking algorithms can get trapped at suboptimal equilibria is well-illustrated.

Weaknesses:
- The discussion following Lemma 2 is unclear (see Question)
- Restricted to tabular MDPs with discrete state-action spaces
-  The irreducible bias at convergence is not fully explained. Theorem 3 guarantees convergence to an ε + O(Cov/(1−γ)²)-ball rather than exact ε-optimality. The paper attributes this to "relaxed weak gradient domination" (analogous to Yuan et al., 2022), but does not clearly explain whether this bias is inherent to performative RL or an artifact of the proof technique. The claim that "this might be inherent to performative RL" is made informally with a citation but no argument. A lower bound or impossibility result establishing this as a fundamental barrier would significantly strengthen the paper.
-  Gradient estimation requires knowledge of transition gradients. The performative policy gradient in Equation (8) requires computing ∇{θ} log P{π_θ}(s_{t+1}|s_t, a_t), the gradient of the log-transition probability with respect to the policy parameters. This requires knowing the parametric form of the transition function and being able to differentiate through it — an assumption that is significantly stronger than what standard model-free PG algorithms require. This computational and modelling requirement is acknowledged only briefly and deserves more prominent discussion, as it limits the practical applicability of PePG.
-  Experimental evaluation is limited to a single environment. All experiments use a single gridworld benchmark with a specific linear reward structure. The paper itself acknowledges in Section 6 that "the gridworld environment is the only benchmark currently available for testing performative RL algorithms," but this does not diminish the concern — conclusions about PePG's empirical superiority are harder to generalize from a single, somewhat synthetic environment.

---

> ### Author Rebuttal · Authors · 2026-03-30
>
> We thank the reviewer for their positive feedback towards our technical novelty and theoretical contribution. We respond to the concerns raised by the reviewer below.
>
> **Q1 and W1: Discussion on Lemma 2.** In usual non-performative MDPs, the RHS of Lemma 2 is $0$ (Lemma 2 in [5]), i.e. advantage solely characterises the suboptimality gap. In Lemma 2 of our work, the RHS exactly characterises the additional cost than classical RL while adapting to the performative effects and aiming for performative optimality. This shows us two things:
>
> (a) After completing epoch $k$, we learn the dynamics using the collected data and decide on a maximal policy with it. As we deploy this policy in the present epoch, underlying environment shifts. The maximum suboptimality induced is characterised by the RHS of Lemma 2. Specifically, if we set $R_{\max} = \mathcal{O}(1-\gamma)$, RHS becomes $\mathcal{O}(\frac{1}{1-\gamma})$.
>
> (b) A performative stable policy fixes the environment inducing policy while aiming to find out the value maximising policy. Thus, given a stream of environment inducing policies (e.g. [8]), the RHS can be reduced to $0$ for stability-seeking algorithms, while this is not possible for any optimality-seeking algorithm as the environment inducing and value maximising policies have to be the same. This result also indicates that the suboptimality gaps exhibited by some of the existing performative stable algorithms [1] might be reduced.
>
> We add a more detailed discussion as above in the updated version.
>
> **Q2: $\rho$ and $\nu$.** The distribution mismatch coefficient often denoted by $D_{\infty}$ or $\kappa$ ([5, 6, 7]), is the worst-case ratio between the state-action visitation distribution of the optimal policy and an almost uniform initial state distribution, and that induced by the current policy and the deployed initial state distribution. We refer to the discussion below Corollary 4.5 in [6] for more details. We observe the same phenomenon reappearing in PRL [8]. We add a remark in the revised manuscript to make this connection explicit.
>
> **W2: From discrete to continuous state-action spaces.** PePG can function in PeMDPs with both the discrete and continuous state-action spaces. However, for the clarity and simplicity of theoretical analysis, and to explicate the effects of performativity, we chose to work with PeMDPs with discrete state-action spaces.
> Though this analysis can be generalised to continuous state-action spaces, it would require careful treatment of function approximators along with the present PePG analysis.
>
> **W3: Lower bound on suboptimality.** For a specific PeMDP setting where the action space consists two options, and a single state $s$. The optimal policy $\pi* = (0,1)$ achieves $r_{\pi*} = (0,1)$ and another policy $\pi_{\theta} = (p,1-p)$ acquires $r_{\pi_{\theta}} = (\Delta,1-\Delta)$ for some $\Delta = f(\pi) > 0$. Thus, we can prove a lower bound $V_{\pi*}^{\pi*}(s) - V_{\pi_{\theta}}^{\pi_{\theta}}(s) \geq \frac{1}{1-\gamma}\min \lbrace \Delta, 1-\Delta \rbrace$. This proves existence of a irreducible bias that holds even if $p \rightarrow 0$, i.e. the second policy converge towards the optimal policy, but $\Delta>0$, i.e. the difference between the rewards induced by them stay positive.
>
> *Note:* This discussion characterises the hardness of a simple PRL setting, where we want to compute a stationary policy to work on an environment shifted by it. Proving a rigorous lower bound remains as a future direction for research.
>
> **W4: Gradient estimation.** Please refer to our response to reviewer M8GZ that implies *PePG does not require knowing the true parametric form but only needs a differentiable surrogate.* This is analogous in spirit to model-based RL, though the object being modeled is the environment's response to policy parameters rather than a fixed dynamics.
>
> **W5: New test-bed for PRL.** Please refer to our response to reviewer 6AgF.
>
> We hope that we have responded to all the raised concerns/questions. We are eager to discuss if there are more questions or comments.
>
> *References:*
>
> [1] Mandal, D., Triantafyllou, S. and Radanovic, G., 2023. Performative reinforcement learning. ICML.
>
> [2] Fair Isaac Corporation. FICO Explainable ML Challenge Dataset. Fair Isaac Corporation, 2018.
>
> [3] Becker, Barry, and Ronny Kohavi. Adult Dataset. UCI ML Repository, 1996.
>
> [4] Hofmann, Hans. Statlog (German Credit Data). UCI ML Repository, 1994.
>
> [5] Agarwal, Alekh, et al. "On the theory of policy gradient methods: Optimality, approximation, and distribution shift." JMLR(2021).
>
> [6] Kakade, Sham, and John Langford. "Approximately optimal approximate reinforcement learning." PCML 2002.
>
> [7] Schulman, John, et al. "Trust region policy optimization." International conference on machine learning. PMLR, 2015.
>
> [8] Sahitaj, R., Sasnauskas, P., Yalın, Y., Mandal, D. and Radanovic, G., 2025. Independent Learning in Performative Markov Potential Games. AISTATS.

---

> > ### Author Rebuttal · Reviewer_JX81 · 2026-04-01
> >
> > I thank the authors for the response. I remain positive about this work.

---

### Official Review · Reviewer_aEKi · 2026-03-17

**Soundness:** 4
**Presentation:** 3
**Significance:** 3
**Originality:** 3
**Overall Recommendation:** 4
**Confidence:** 3

**Summary:**

The authors propose an adaptive sampling in the episodic continuous-time reinforcement learning setting. The policy at episode n maximizes the return over a maximum likelihood confidence set for the dynamics. The authors show an instance-dependent regret guarantee under measures of complexity of the dynamics (bracketing entropy and eluder dimension) where instance dependence shows up under the form of the variance of the return under the deployed policies.

**Compliance With Llm Reviewing Policy:**

Affirmed.

**Key Questions For Authors:**

What are guarantees for optimism under MLE confidence intervals of dynamics in discrete RL?

Are there any known minimax lower bounds in CTRL? Instance-dependent lower bounds?

**Limitations:**

Contextualisation with discrete-time RL and previous rate results would help.

**Strengths And Weaknesses:**

Strength:

Very clear exposition - it's a pleasure to read.
Results appear sound.
A second-order regret bound in this setting appears novel and significant.

Weaknesses:

- CTRL appear niche to me (it may be subjective, but to the best of my knowledge, CTRL has been much less studied than discrete time RL), and the comparison with known results in the discrete case RL setting would help - even though comparisons may not be 1-to-1.
- A discussion of the regret rate in comparison to existing regret rate results in CTRL would help understand the novelty.

---

### Decision · Program_Chairs · 2026-04-30

**Decision:**

Accept (regular)

**Comment:**

This paper considers RL in the performative setting and derives performative analogs of the performance difference lemma and policy gradient theorem. Furthermore, it introduces PePG, a policy gradient method that provably converges to performative optimality in tabular PeMDPs. The reviewers concensus is for accepting the paper, highlighting the clear exposition, sound theory, and the conceptual/quantitative distinction between optimality- and stability-seeking methods, with improved empirical evidence on a gridworld benchmark. The authors’ rebuttal does not resolve all questions and concerns but adequately clarifies the made smoothness assumptions, notation for the distribution mismatch coefficient, differences from stability-seeking approaches (e.g., INPG), and practical gradient estimation via differentiable surrogate models. Remaining limitations include the tabular/discrete scope, reliance on smoothness and access to (or modeling of) transition/reward gradients, and the narrow empirical evaluation. Nevertheless, the contributions and theoretical significance of the presented results justify acceptance despite limited empirical evaluation.
For the camera-ready, the authors should (i) clarify the discussion after Lemma 2, (ii) remove or soften speculative statements, (iii) expand on where smoothness holds/fails and show explicit dependence on sensitivity/smoothness constants, and (iii) detail gradient-estimation practice and note broader benchmarks/future extensions to continuous spaces.